# Benchmarking Retrieval-Augmented Multimodal Generation for Document Question Answering

**Kuicai Dong**\* **Yujing Chang**\* **Shijie Huang** **Yasheng Wang** **Ruiming Tang** **Yong Liu**

HUAWEI NOAH'S ARK LAB

correspond to {kuicai.dong, liu.yong6}@huawei.com

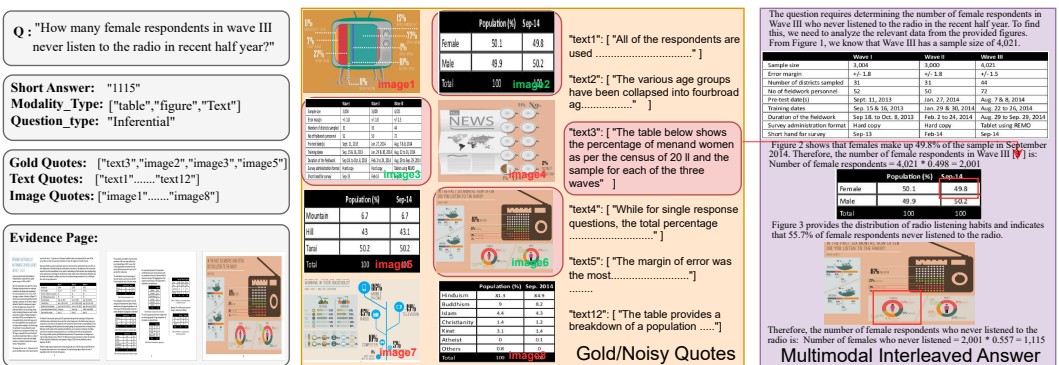

Figure 1: `MMDocRAG` annotations: QA pair, noisy multimodal quotes, and multimodal answer.

## Abstract

Document Visual Question Answering (DocVQA) faces dual challenges in processing lengthy multimodal documents (text, images, tables) and performing cross-modal reasoning. Current document retrieval-augmented generation (DocRAG) methods remain limited by their text-centric approaches, frequently missing critical visual information. The field also lacks robust benchmarks for assessing multimodal evidence selection and integration. We introduce `MMDocRAG`, a comprehensive benchmark featuring 4,055 expert-annotated QA pairs with multi-page, cross-modal evidence chains. Our framework introduces innovative metrics for evaluating multimodal quote selection and enables answers that interleave text with relevant visual elements. Through large-scale experiments with 60 VLM/LLM models and 14 retrieval systems, we identify persistent challenges in multimodal evidence retrieval, selection, and integration. Key findings reveal advanced proprietary LVMs show superior performance than open-sourced alternatives. Also, they show moderate advantages using multimodal inputs over text-only inputs, while open-source alternatives show significant performance degradation. Notably, fine-tuned VLM/LLMs achieve substantial improvements for multimodal generation. `MMDocRAG` establishes a rigorous testing ground and provides actionable insights for developing more robust multimodal DocVQA systems. Our benchmark and code are available at `https://mmdocrag.github.io/MMDocRAG/`.

## 1 Introduction

DocVQA [49] focuses on visual question answering over documents with rich multimodal content. Multimodal documents (*e.g.,* financial reports, technical manuals, and medical records) present significant challenges for DocVQA: (i) they are typically lengthy [18], complicating the identification of key evidence, and (ii) they require complex reasoning across various modalities, including images, tables, charts, and layout structures. Thus, recent studies [9, 10, 20, 30, 67] have adopted document

---

\*These authors contributed equally to this work.

39th Conference on Neural Information Processing Systems (NeurIPS 2025) Track on Datasets and Benchmarks.

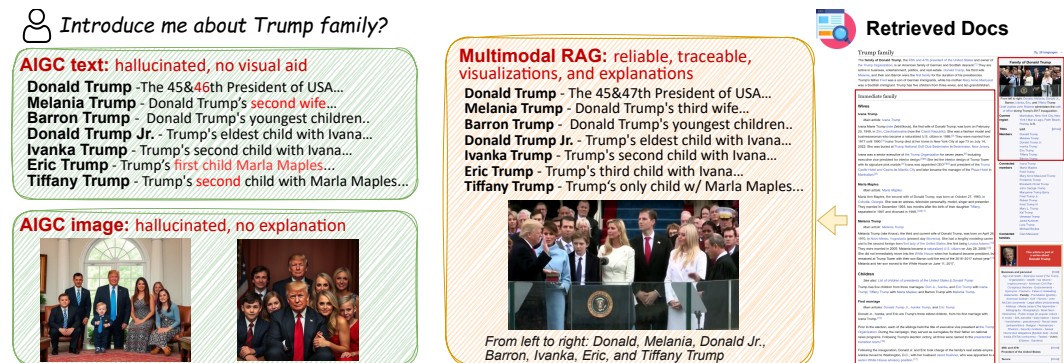

Figure 2: Comparison of different QA paradigms: text-only, image-only, and interleaved text-image.

| Benchmarks | Document | | Question | | Evi. Loc. | | Answer | Evaluation Metric | | |
|---|---|---|---|---|---|---|---|---|---|---|
| | Domain | #Pages | #Num | Expert | Page | Quote | Type | Evi Loc. | Evi Sel. | Ans. |
| MP-DocVQA [76] | Industrial | 8,3 | 46k | ✗ | ✓ | ✗ | TXT | ✗ | ✗ | ✓ |
| DUDE [38] | Multiple | 5.7 | 24k | ✗ | ✓ | ✓ | TXT | ✗ | ✗ | ✓ |
| SlideVQA [72] | Slides | 20.0 | 14.5k | ✗ | ✓ | ✗ | TXT | ✗ | ✗ | ✓ |
| PDF-MVQA [15] | Biomedical | 9.6 | 260k | ✗ | ✓ | ✓ | TXT | ✓ | ✗ | ✓ |
| MMLongBench-Doc [47] | Multiple | 47.5 | 1,082 | ✓ | ✓ | ✗ | TXT | ✗ | ✗ | ✓ |
| DocBench [92] | Multiple | 66.0 | 1,102 | ✓ | ✗ | ✗ | TXT | ✗ | ✗ | ✓ |
| M3DocVQA [10] | Wikipedia | 12.2 | 2,441 | ✓ | ✓ | ✗ | TXT | ✓ | ✗ | ✓ |
| M-Longdoc [9] | Multiplie | 210.8 | 851 | ✓ | ✓ | ✗ | TXT | ✓ | ✗ | ✓ |
| MMDocIR [19] | Multiple | 65.1 | 1,658 | ✓ | ✓ | ✓ | TXT | ✓ | ✗ | ✗ |
| MuRAR [91] | Webpage | - | 300 | ✓ | ✗ | ✗ | TXT/TAB/I/V | ✗ | ✗ | ✓ |
| M²RAG [48] | Webpage | - | 200 | ✓ | ✗ | ✗ | TXT/I | ✗ | ✗ | ✓ |
| MMDocRAG | Multiple | 67.0 | 4,055 | ✓ | ✓ | ✓ | TXT/C/TAB/I | ✓ | ✓ | ✓ |

Table 1: Comparison between `MMDocRAG` and existing DocVQA/DocRAG benchmarks. **TXT/C/TAB/I/V** refers to pure text/chart/table/image/video, respectively. "**Evi. Loc.**" refer to locating which pages and quotes contain evidence in the document. "**Evi. Sel.**" aims to select useful evidence given a list of noisy multimodal pages or quotes (*e.g.,* only 2 out of 20 quotes are relevant).

retrieval-augmented generation (DocRAG), which first retrieves relevant document pages and then generates answers by selecting and composing supporting evidence. However, current DocRAG systems show significant limitations, resulting in perspective narrowing, as highlighted in Table 1: 1. **Unimodal Bias**: Generated answers frequently over-rely on plain text, neglecting valuable visual information such as charts and tables. Prior work [48, 91] has shown that multimodal content greatly enhances user understanding, supporting the notion that "*a single image is worth a thousand words*". However, current AIGC models struggle in generating informative visualizations from scratch. Ideally, the multimodal evidence displayed in answer must be traceable and credible, allowing users to verify supporting information as shown in Figure 2. 2. **Evaluation Flaws**: Existing benchmarks [9, 19, 47] primarily assess the recall of retrieved quotes or the quality of textual answers. There are no benchmarks for evaluating a model to (i) *select relevant multimodal evidence from noisy retrieved quotes* or (ii) *align and integrate multimodal content with text in a coherent and logical manner*. These gaps hinder the evaluation in complex multimodal RAG scenarios.

In response to these challenges, we propose `MMDocRAG`, a comprehensive multimodal document question answering benchmark (§2), with an annotation exemplified in Figure 1. `MMDocRAG` consists of 4,055 expert-annotated question-answer pairs, each accompanied by multimodal evidence chains which may span multiple pages and modalities, including both text and image quotes. Evidence is provided at multiple granularities, ranging from coarse-grained page-level screenshots to fine-grained quotes extracted based on document layout. In addition to these annotations, `MMDocRAG` introduces two novel evaluation features: **(1) Quote Selection**: We propose a practical evaluation metric that measures a model's ability to select and integrate relevant multimodal quotes. To increase task difficulty, we include hard text and image negatives[2] mixed with gold (relevant) quotes. **(2) Multimodal Output Paradigm**: Our benchmark supports multimodal answers, allowing document figures, infographics, charts, and tables to be interleaved within textual responses. This paradigm enhances both the interpretability and cognitive effectiveness of generated answers.

---

[2]Hard negatives refer to quotes retrieved with high textual or visual similarity but irrelevant to the question.

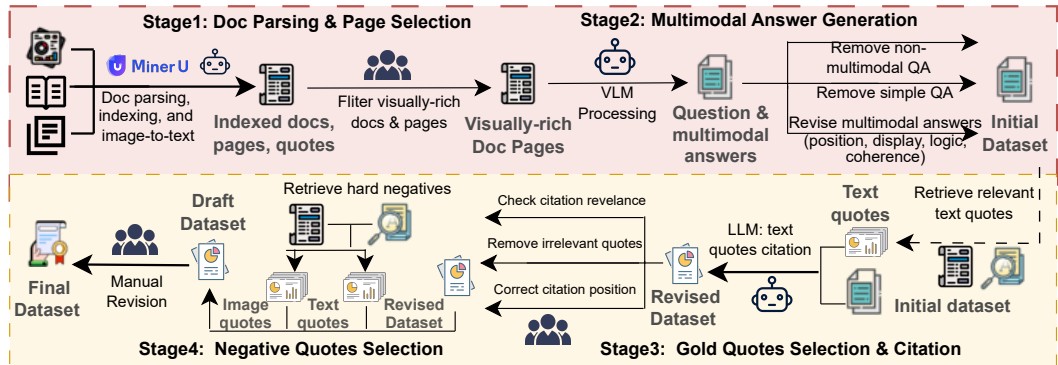

Figure 3: Four-stage Annotation Pipeline for `MMDocRAG`.

Utilizing `MMDocRAG`, we conduct comprehensive experiments on DocVQA/RAG tasks. Our study includes 60 latest large models, among which 33 VLMs can handle multimodal (interleaved text and image) inputs and 27 LLMs can only process text inputs. For multimodal tasks with LLMs, we either extract text from images using OCR [71] tools ("OCR-text") or use VLMs [55, 60] to generate detailed image descriptions ("VLM-text"). We fix the number of input quotes to 15 or 20 for multimodal generation. Experimental results (§4.3) highlight the complexities of multimodal DocRAG: the best model, GPT4.1 [58], achieves an $F_1$ score of only 70.2% for quote selection. For multimodal answer quality, we assess fluency, citation quality, text-image coherence, reasoning, and factual accuracy, with GPT4.1 achieving the highest scores. Overall, proprietary VLMs significantly outperform open-sourced VLMs and LLMs. Meanwhile, fine-tuning Qwen2.5-instruct LLMs [65] (3–72B parameters), Qwen2.5-VL-Instruct VLMs [3] (3&7B), and InternVL-3 VLMs [90] (8&9B) yields substantial performance improvements. It is worthnoting that the advanced proprietary VLMs generally show better performance using multimodal inputs over pure-text inputs, and the performance gap is modest. In contrast, open-source or smaller proprietary VLMs show significant performance boost using pure-text inputs than multimodal inputs (§4.4). Notably, LLMs leveraging VLM-text significantly outperform those using OCR-text (§4.5). Additionally, we evaluate the retrieval performance of 6 text, 4 visual, and 4 hybrid retrievers, in both pure retrieval (§4.7) and end-to-end RAG (§4.8) mode. The results further highlight the challenges of extracting relevant multimodal quotes from long documents. In summary, our contributions are:

- We propose `MMDocRAG` benchmark (§2) for evaluating multimodal generation on DocVQA/RAG tasks. Our dataset include over 4,000 QA pairs, diverse forms of evidence, a mixture of gold and noisy quotes to enable nuanced quote selection, and answers with interleaved multimodal content.

- We conduct extensive evaluations (§4) on multimodal RAG, covering (i) retrieval performance on 6 text, 4 visual,4 hybrid retrievers, (ii) quote selection $F_1$ and (iii) multimodal answer quality across 37 open-source and 23 proprietary models, and 9 models finetuned using `MMDocRAG` dev-set.

- Our results indicate that even state-of-the-art LLMs and VLMs struggle with multimodal integration, while targeted fine-tuning can significantly improve model performance on these tasks.

## 2 `MMDocRAG` Benchmark

As exemplified in Figure 1, `MMDocRAG` contains annotations: QA pair, page and quote evidence, noisy quotes, and multimodal answer. The construction pipeline and statistics are in Figure 3 and Table 2.

### 2.1 Construction

**Document Parsing and Evidence Selection.** We utilize the document corpus from the MMDocIR dataset [19], which consists of 313 documents spanning over 10 diverse domains. These documents are sufficiently long (averaging 65.1 pages) and provide rich multimodal coverage. We process the documents with MinerU [77], which leverages LayoutLMv3 [32] to detect page layouts and classify them as body text, titles, equations, figures, tables, etc. Each identified layout serves as a content-aware chunk [17], or "quote". Text quotes correspond to layouts such as equations or paragraphs, and are stored in text format. For image quotes (*e.g.,* tables or figures), we extract text using OCR [71] ("OCR-text") and generate detailed descriptions using VLMs [55, 60] ("VLM-text"). Consequently, each image quote is stored in three formats: original image, OCR-text, and VLM-text.

| Statistic | Number |
|---|---|
| **Documents** | 222 |
| - Domain Types | 10 |
| - Avg./Med./Max. pages per doc | 67 / 28 / 844 |
| - Avg./Med./Max. words per doc | 33k / 10k / 332k |
| - Avg./Med./Max. images per doc | 63 / 31 / 663 |
| - Avg./Med./Max. texts per doc | 536 / 194 / 5k |
| **Total Questions** | 4,055 |
| - Development / Evaluation split | 2,055 / 2,000 |
| - Derived questions | 820 (20.2%) |
| - Newly-annotated questions | 3,235 (79.8%) |
| - Cross-page questions | 2,107 (52.0%) |
| - Multi-image questions | 1,590 (39.2%) |
| - Cross-modal questions | 2,503 (61.7%) |

**(Question Type)**

| | |
|---|---|
| Comparative: 1,456 (35.9%) | Analytical: 488 (12.0%) |
| Descriptive: 1,256 (31.0%) | Inferential: 75 (1.8%) |
| Interpretative: 697 (17.2%) | Others: 83 (2.0%) |

**(Evidence Modality)**

| | |
|---|---|
| Text - 2,457 (60.1%) | Table - 2,677 (66.0%) |
| Figure - 1,004 (24.8%) | Chart - 636 (15.9%) |

| Statistic | Number |
|---|---|
| **All Selected Quotes (Text/Image)** | 48,618 / 32,071 |
| - Gold Quotes (Text/Image) | 4,640 / 6,349 |
| - Noisy Quotes (Text/Image) | 43,978 / 25,722 |
| Avg./Med./Max words: question | 21.9 / 20 / 73 |
| Avg./Med./Max words: short ans | 23.9 / 22 / 102 |
| Avg./Med./Max words: multimodal ans | 221.0 / 203 / 768 |
| Avg./Med./Max number of gold quotes | 2.7 / 2 / 12 |

Table 2: Overall Dataset Statistics.

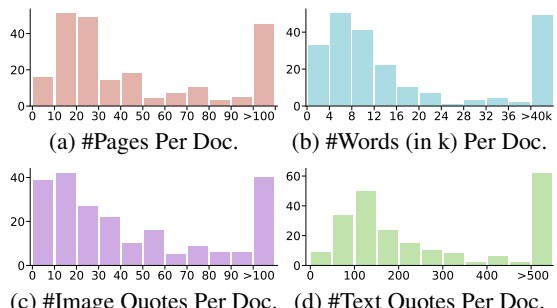

(a) #Pages Per Doc.  (b) #Words (in k) Per Doc.

(c) #Image Quotes Per Doc.  (d) #Text Quotes Per Doc.

Figure 4: Document Distribution.

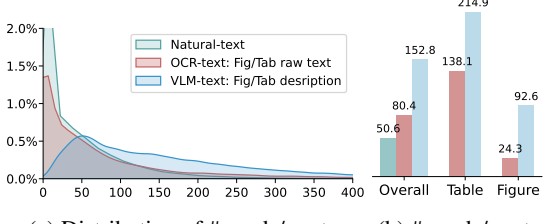

(a) Distribution of #words/quote  (b) #words/quote

Figure 5: Length Distribution: OCR/VLM-text.

After indexing all documents, we carefully select pages with rich multimodal and text information. This process yields 2,373 high-quality pages, forming the basis for subsequent annotation.

**Multimodal Answer Generation: Existing QA Pairs.** We review 1,658 QA pairs from the MMDocIR dataset [19] and select questions suitable for multimodal answer generation. Specifically, we identify 943 questions that can be answered using interleaved text, figures, tables, infographics, or charts as supporting evidence. These questions, along with their textual answers and evidence, are used as input to GPT-4o [55] to generate draft multimodal answers. We further refine the outputs by (i) discarding QA pairs lacking visual content, (ii) removing overly simple questions, and (iii) revising the positioning, formatting, and coherence of the multimodal content. This process results in 821 QA pairs with multimodal answers that effectively interleave text and multimodal information.

**Multimodal Answer Generation: New QA Pairs.** The process for generating multimodal answers for new QA pairs is similar to that of existing QA pairs, with the key distinction that VLMs autonomously generate both the questions and textual answers based on provided evidence. We define eight question types: descriptive, comparative, procedural, interpretative, causal, analytical, inferential, and application-based. To create challenging questions, we use either single or multiple document pages as input during annotation. This results in a new dataset of 1,719 single-page and 1,630 multi-page questions, each paired with corresponding multimodal answers.

**Gold Quotes Citation.** To reduce hallucination and improve answer traceability and credibility, we explicitly cite gold quotes in the generated answers. Image quotes are cited using the format " ", while text quotes are cited as " [i] ". Since images are already explicitly referenced in the multimodal answers, we focus on accurately citing text quotes in this step. For each QA pair, we use a dense retriever to identify the top 20 most relevant text quotes. These candidates are provided to an LLM, which selects the most contextually relevant evidence and inserts the citations at appropriate positions. Expert evaluators assess citation quality by verifying that the selected quotes genuinely support the answer, and ensuring the insertion positions coherently reflect the cited evidence. As a result, we revise 2,457 multimodal answers, with a total of 4,641 text quotes cited.

**Negative Quotes Augmentation.** To increase task difficulty, we augment the context with hard negative text and image quotes mixed with gold (relevant) quotes. Hard negatives are irrelevant quotes that exhibit high textual or visual similarity to the question or answer. This augmentation aims to assess the model's ability to distinguish relevant information from confounding distractors. Specifically, we select hard negatives from the top 20 relevant quotes retrieved, based on either the question or answer. For each question, we generate two versions of the candidate set: (i) 15 quotes (5 images and 10 texts) and (ii) 20 quotes (8 images and 12 texts). Each quote is annotated with its layout and page identifier, allowing precise traceability to its origin within the document corpus.

## 2.2 Dataset Analysis

The main statistics of `MMDocRAG` are summarized in Table 2. In total, our benchmark contains 4,055 questions, each paired with image-text interleaved answers and augmented with supporting evidence. We split the total of 4,055 questions into 2,055 / 2,000 for model development and evaluation. The questions are based on 222 **lengthy documents** spanning 10 different types, with an average length of 67 pages and approximately 33k words per document. Detailed distributions of the documents are shown in Figure 4. For **question characteristics**, there are 2,107 cross-page questions (requiring evidence from 2+ pages), 1,590 multi-image questions (involving 2+ image quotes), and 2,503 cross-modal questions (requiring multiple evidence modalities). All questions are categorized into one of eight predefined types. Regarding **quotes**, the dataset includes 48,618 text quotes (of which 4,640 are gold) and 32,071 image quotes (with 6,349 gold quotes). On average, each question is associated with 2.7 gold quotes out of 15/20 candidates, resulting in only 18.0/13.5% relevant quotes. Figure 5. Notably, VLM-text is significantly longer and more detailed than OCR-text. For **answer** length, the short answer contains an average of 23.9 tokens, whereas the multimodal answer averages 221.0 tokens. Additional annotation examples can be found in Appendix D.

## 2.3 Quality Assurance

To ensure the quality of `MMDocRAG`, we employ a rigorous quality assurance process that combines semi-automated validation of draft annotations with manual cross-validation of final annotations.

**Semi-automated Validation of Draft Annotation.** For document page selection, layout detection models automatically identify pages rich in multimodal content, which are then reviewed by expert annotators; 74.3% of these pages are retained. For quote integration and multimodal answer generation, we leverage (i) VLMs to select and insert relevant visual content coherently, and (ii) LLMs to check the accuracy and coherence of integrated text. Answers that fail validation are regenerated, with a maximum of three attempts. The filtered answers and gold quotes undergo further expert validation, resulting in a retention of 90.2% of answers and 93.5% of gold quotes.

**Manual Cross-validation of Final Annotation.** We divide the draft annotations into two parts of approximately 2,300 QA pairs each, with 500 overlapping pairs serving as validation checkpoints. Two annotation groups are assigned to revise separate parts, while both annotate the overlapping set for quality comparison. Each group's answers are measured against the other's as ground truth, enabling mutual validation. This cross-evaluation allows us to assess consistency in quote selection and answer quality, and to identify discrepancies for further refinement. For quote selection, Groups A and B achieved $F_1$ scores of 89.7 and 91.4, respectively. For answer quality, average scores were 4.23 for Group A and 4.17 for Group B (see Section 4.1 for details on the scoring metric).

# 3 Task Definition

Document retrieval-augmented multimodal generation aims to produce multimodal answer given a user question and targeted document corpus. This task consists of two key stages as follows:

**Multimodal Retrieval.** Let $\mathcal{D}$ denote a document corpus consisting of text quotes $\mathcal{T} = \{t_1, t_2, \ldots, t_m\}$ and image quotes $\mathcal{I} = \{i_1, i_2, \ldots, i_n\}$, as extracted via layout detection (see Section 2.1). On average, documents in `MMDocRAG` contain 63 image quotes and 536 text quotes. The objective is to retrieve a subset of quotes that are most relevant to a query $Q$ from $\mathcal{T}$ and $\mathcal{I}$, by ranking them based on similarity scores, $\text{Sim}(Q, t)$ and $\text{Sim}(Q, i)$. The top-$k$ quotes, where $k \ll n + m$, are selected as candidate evidence.

**Multimodal Answer Generation.** Different document parsing, chunking strategies, or retrieval models may yield varying results, complicating fair evaluation of answer generation due to differences in available context. Therefore, we employ a fixed set of candidate quotes as the input context to isolate the evaluation of LLM/VLM quote selection and answer generation capabilities. Specifically, we consider two settings: using 15 or 20 candidate quotes as context, denoted as $\mathcal{C}_{15}$ and $\mathcal{C}_{20}$, respectively. $\mathcal{C}_{15} = \{t_1, \ldots, t_{10}, i_1, \ldots, i_5\}$ consists of 10 text quotes from $\mathcal{T}$ and 5 image quotes from $\mathcal{I}$. $\mathcal{C}_{20} = \{t_1, \ldots, t_{12}, i_1, \ldots, i_8\}$ consists of 12 text quotes from $\mathcal{T}$ and 8 image quotes from $\mathcal{I}$. Given user question $Q$ and quotes context $\mathcal{C}_{15}$ and $\mathcal{C}_{20}$, the model needs to generate multimodal answer $A$. Irrelevant (noisy) quotes should be excluded from the generated answer.

We highlight that `MMDocRAG` tasks on selecting and integrating multimodal content (from $\mathcal{C}_{15}$ and $\mathcal{C}_{20}$) during multimodal answer generation, rather than generating multimodal content from scratch.

# 4 Experiments

## 4.1 Evaluation Metric

**Multimodal Retrieval.** The retriever scores each quote in the document based on its relevance to the question, and returns the top $k$ candidates with the highest scores. We use recall@$k$ to calculate the proportion of the ground truth quote evidence that is successfully retrieved.

**Multimodal Answer Generation.** To comprehensively evaluate multimodal answer generation, we employ a combination of automatic and LLM-as-judge metrics covering quote selection accuracy, surface-level answer similarity, and qualitative answer quality (See more details in Appendix A.).

- **Quotes Selection.** We explicitly compute precision, recall, and $F_1$ scores for both text and image quotes, which are then averaged to yield an overall quote selection $F_1$.
- **Surface-level Similarity.** We employ BLEU [59] and ROUGE-L [42] as lexical similarity metrics.
- **LLM-as-Judge Criteria.** We evaluate predicted answer from five dimensions: fluency, cite quality, text-image coherence, reasoning logic, and factuality, where each is scaled from 0 to 5.

## 4.2 Baseline Models

**Quotes Retrieval.** We first evaluate **6 text and 4 visual retrievers**. For **hybrid retrieval**, quotes are combined as follows: top 10 (3 images and 7 texts from visual and text retriever, respectively), top 15 (5 images, 10 texts), and top 20 (8 images, 12 texts). See Appendix C.3 for more details.

**Multimodal Answer Generation.** We evaluate **60 latest models** by using quotes as: (i) multimodal inputs for VLM, and (ii) pure-text inputs for VLM and LLM (see Appendix C.1 for implementation details). Then, we evaluate **9 finetuned models** (Qwen2.5 LLMs [65] with 3, 7, 14, 32, and 72B parameters, Qwen2.5-VL VLMs [3] of 3B and 7B parameters, and InternVL-3 VLMs [90] of 8B and 9B parameters) using `MMDocRAG` dev-set. See Appendix C.2 for finetuning details).

## 4.3 Main Results

We present the results of 60 state-of-art LLM and VLM models in Table 8 and Table 3, which use 15 and 20 quotes as context for multimodal generation respectively. The performance distribution of these models is illustrated in Figure 7. Our key findings are summarized below:

- **Quotes Selection with 20 quotes.** GPT-4.1 achieves the highest $F_1$ score of 70.2, while other leading proprietary models range from 60 to 66. In contrast, smaller proprietary and open-source models generally achieve $F_1$ scores between 20 to 60, indicating substantial room for improvement.
- **Answer Quality with 20 quotes.** GPT-4.1 again leads with a best score of 4.14, followed by other proprietary models scoring between 3.6 to 4.0. Most smaller proprietary and open-source models score between 3.0 and 3.6, primarily due to citation, reasoning, and factuality errors.
- **Multimodal vs Pure-text Quotes.** Proprietary VLMs using multimodal inputs generally achieve better or comparable performance compared to pure-text inputs, albeit with significant computational overhead and increased latency. Smaller VLMs struggle with both quote selection and answer generation in the multimodal setting. Additional discussion is provided in Section 4.4.
- **Thinking models do not show advanced performance**, although costing 3 times more output tokens. This indicates the step-by-step reasoning on multimodal quotes selection and integration does not help much on final answer generation. See Appendix B.2 for more results.
- **Fine-tuning** can significantly increase the performance in selecting and generating multimodal information, as clearly displayed in Figure 6. Refer to more qualitatively analysis in Appendix F.3.

Beyond the overall results, we also provide fine-grained analysis on model performance across different document domains (§B.4), question types (§B.5), and evidence configurations (§B.6). Our detailed analysis reveals that model performance varies significantly based on document complexity (with "Workshop" documents being easiest and "Brochure" documents most challenging), question reasoning requirements (with "Descriptive" questions outperforming "Interpretative" ones), and evidence structure (with single-image/page evidence consistently outperforming multi-image/page scenarios). These granular insights demonstrate distinct strengths and limitations across model architectures and provide valuable guidance for practical deployment considerations. Complete breakdowns and detailed findings are presented in Appendix B.4, B.5, and B.6.

Table 3 columns: **Method** | **Tokens** (In, Out) | **Quote Selection** — Image Quotes (Prec, Rec, F₁), Text Quotes (Prec, Rec, F₁), F₁ | **Multimodal Answer Quality** (Bleu, RougeL, Fluency, Cite Qlty, Txt-Im Coher, Reas Logic, Factuality, Avg)

**Use using 20 quotes (8 images & 12 texts) as pure-text input sequence for both LLM and VLM**

*Open-source Models*

| Method | In | Out | Img Prec | Img Rec | Img F₁ | Txt Prec | Txt Rec | Txt F₁ | F₁ | Bleu | RougeL | Fluency | Cite Qlty | Txt-Im Coher | Reas Logic | Factuality | Avg |
|---|---|---|---|---|---|---|---|---|---|---|---|---|---|---|---|---|---|
| Qwen2.5-3B-Inst | 3.6k | 415 | 50.4 | 23.6 | 32.2 | 17.8 | 10.7 | 13.4 | 25.0 | 0.123 | 0.271 | 4.02 | 2.52 | 2.73 | 2.87 | 2.59 | 2.94 |
| *- After Fine-tuning* | 3.6k | 286 | 68.1 | 57.8 | 62.5 | 44.6 | 1.4 | 2.8 | 49.6 | 0.182 | 0.338 | 4.45 | 3.08 | 3.40 | 3.03 | 2.60 | 3.31 |
| Llama3.2-3B-Inst | 3.4k | 418 | 37.9 | 25.7 | 30.6 | 18.5 | 30.4 | 23.0 | 23.0 | 0.089 | 0.243 | 3.35 | 1.87 | 2.17 | 2.30 | 2.17 | 2.37 |
| Qwen3-4B (think) | 3.6k | 1072 | 68.5 | 64.4 | 66.4 | 36.1 | 46.7 | 40.7 | 58.2 | 0.139 | 0.301 | 4.25 | 3.13 | 3.57 | 3.55 | 3.40 | 3.58 |
| Mistral-7B-Inst | 4.0k | 451 | 53.4 | 45.2 | 49.0 | 23.1 | 44.2 | 30.4 | 38.6 | 0.109 | 0.251 | 3.53 | 2.38 | 2.82 | 2.67 | 2.50 | 2.78 |
| Qwen2.5-7B-Inst | 3.6k | 302 | 66.5 | 45.5 | 54.0 | 36.2 | 28.2 | 31.7 | 45.8 | 0.159 | 0.313 | 4.27 | 2.93 | 3.21 | 3.22 | 3.07 | 3.34 |
| *- After Fine-tuning* | 3.6k | 223 | 71.2 | 66.8 | 69.0 | 38.5 | 2.6 | 4.9 | 56.0 | 0.199 | 0.353 | 4.59 | 3.38 | 3.70 | 3.36 | 2.98 | 3.60 |
| Llama3.1-8B-Inst | 3.4k | 435 | 54.1 | 51.8 | 52.9 | 24.1 | 33.1 | 29.5 | 41.0 | 0.112 | 0.254 | 3.61 | 2.40 | 2.82 | 2.75 | 2.70 | 2.86 |
| Qwen3-8B (think) | 3.6k | 1018 | 71.3 | 67.5 | 69.4 | 34.4 | 60.1 | 43.8 | 59.7 | 0.138 | 0.302 | 4.15 | 3.13 | 3.57 | 3.40 | 3.32 | 3.51 |
| InternVL3-8B | 3.6k | 385 | 60.4 | 54.7 | 57.4 | 30.7 | 34.9 | 32.7 | 48.1 | 0.147 | 0.290 | 3.90 | 2.68 | 3.11 | 3.07 | 2.93 | 3.14 |
| InternVL3-9B | 4.0k | 395 | 72.7 | 43.3 | 54.3 | 30.5 | 29.2 | 29.8 | 45.4 | 0.157 | 0.300 | 4.09 | 2.87 | 3.28 | 3.23 | 3.03 | 3.30 |
| Qwen2.5-14B-Inst | 3.6k | 362 | 71.5 | 56.0 | 62.8 | 34.8 | 43.9 | 38.8 | 54.7 | 0.148 | 0.295 | 4.26 | 3.15 | 3.48 | 3.33 | 3.24 | 3.49 |
| *- After Fine-tuning* | 3.6k | 282 | 74.1 | 70.6 | 72.3 | 53.0 | 6.4 | 11.5 | 59.4 | 0.212 | 0.366 | 4.69 | 3.62 | 3.93 | 3.44 | 3.34 | 3.84 |
| Qwen3-14B (think) | 3.6k | 920 | 73.0 | 64.9 | 68.7 | 36.4 | 57.3 | 44.5 | 59.9 | 0.142 | 0.305 | 4.29 | 3.25 | 3.66 | 3.59 | 3.47 | 3.65 |
| InternVL3-14B | 3.6k | 385 | 73.3 | 45.6 | 56.2 | 30.5 | 56.4 | 39.6 | 49.9 | 0.159 | 0.301 | 4.22 | 3.04 | 3.44 | 3.42 | 3.29 | 3.48 |
| Mistral-Small-24B-Inst | 3.7k | 391 | 49.3 | 46.7 | 48.0 | 22.7 | 46.0 | 30.4 | 39.0 | 0.091 | 0.236 | 2.34 | 1.77 | 2.12 | 1.88 | 1.90 | 2.00 |
| Qwen3-30B-A3B | 3.6k | 969 | 72.5 | 68.2 | 70.3 | 36.7 | 61.1 | 45.9 | 61.4 | 0.147 | 0.305 | 4.22 | 3.23 | 3.68 | 3.49 | 3.40 | 3.60 |
| Qwen2.5-32B-Inst | 3.6k | 320 | 69.4 | 66.8 | 68.1 | 40.7 | 33.0 | 36.5 | 58.9 | 0.159 | 0.307 | 4.39 | 3.27 | 3.59 | 3.48 | 3.41 | 3.63 |
| *- After Fine-tuning* | 3.6k | 282 | **77.5** | 74.2 | **75.8** | **62.1** | 22.9 | 33.4 | **65.1** | **0.224** | **0.377** | 4.73 | 3.71 | 4.06 | 3.73 | 3.41 | 3.93 |
| Qwen3-32B (think) | 3.6k | 917 | 72.8 | 57.2 | 64.0 | 34.5 | 64.3 | 44.9 | 54.5 | 0.137 | 0.300 | 4.28 | 3.22 | 3.60 | 3.53 | 3.44 | 3.61 |
| InternVL-38B | 3.6k | 338 | 68.4 | 52.6 | 59.5 | 33.5 | 64.8 | 44.1 | 55.0 | 0.160 | 0.307 | 4.30 | 3.24 | 3.61 | 3.52 | 3.36 | 3.61 |
| Mistral-8x7B-Inst | 4.0k | 259 | 57.2 | 32.6 | 41.5 | 28.5 | 24.2 | 26.1 | 30.7 | 0.098 | 0.248 | 3.22 | 2.09 | 2.38 | 2.37 | 2.23 | 2.46 |
| Llama3.3-70B-Inst | 3.4k | 430 | 54.3 | **82.5** | 65.5 | 30.6 | 64.3 | 41.5 | 55.6 | 0.120 | 0.264 | 3.93 | 2.72 | 3.17 | 3.11 | 3.26 | 3.24 |
| Qwen2.5-72B-Inst | 3.6k | 380 | 76.5 | 62.1 | 68.5 | 38.8 | 49.2 | 43.4 | 59.1 | 0.173 | 0.324 | 4.48 | 3.41 | 3.71 | 3.64 | **3.53** | 3.75 |
| *- After Fine-tuning* | 3.6k | 286 | 76.6 | 74.8 | 75.7 | 56.9 | 23.4 | 33.1 | 64.9 | **0.224** | **0.377** | **4.76** | **3.74** | **4.11** | **3.78** | 3.48 | **3.97** |
| InternVL-78B | 3.6k | 375 | 66.0 | 69.0 | 67.4 | 32.1 | 65.3 | 43.1 | 56.4 | 0.157 | 0.302 | 4.26 | 3.13 | 3.55 | 3.46 | 3.39 | 3.56 |
| Qwen3-235B-A22B | 3.6k | 1052 | 71.2 | 67.4 | 69.2 | 35.3 | 62.8 | 45.2 | 59.5 | 0.138 | 0.296 | 4.34 | 3.38 | 3.77 | 3.72 | 3.63 | 3.77 |
| Deepseek-V3 | 3.4k | 234 | 70.8 | 73.4 | 72.1 | 37.3 | 59.8 | **45.9** | 61.1 | 0.171 | 0.338 | 4.57 | 3.31 | 3.74 | 3.62 | 3.47 | 3.74 |
| Deepseek-R1 | 3.4k | 930 | 66.5 | 77.0 | 71.4 | 31.5 | **68.6** | 43.2 | 59.4 | 0.113 | 0.268 | 4.13 | 3.17 | 3.56 | 3.30 | 3.25 | 3.48 |
| - Distill-Qwen-32B | 3.6k | 731 | 65.5 | 47.4 | 55.0 | 38.4 | 30.1 | 33.8 | 44.8 | 0.137 | 0.305 | 4.29 | 2.75 | 3.15 | 3.31 | 3.20 | 3.34 |
| - Distill-Llama-70B | 3.3k | 680 | 69.0 | 52.7 | 59.8 | 38.4 | 42.6 | 40.4 | 51.0 | 0.144 | 0.311 | 4.36 | 2.99 | 3.39 | 3.42 | 3.33 | 3.50 |
| Llama4-Scout-17Bx16E | 3.3k | 418 | 60.4 | 59.4 | 59.9 | 27.6 | 55.3 | 36.8 | 48.2 | 0.132 | 0.271 | 3.77 | 2.69 | 3.09 | 3.03 | 3.03 | 3.12 |
| Llama4-Mave-17Bx128E | 3.3k | 366 | 69.2 | 75.0 | 72.0 | 36.6 | 50.7 | 42.5 | 58.3 | 0.153 | 0.301 | 4.09 | 3.17 | 3.58 | 3.52 | 3.60 | 3.59 |

*Proprietary Models*

| Method | In | Out | Img Prec | Img Rec | Img F₁ | Txt Prec | Txt Rec | Txt F₁ | F₁ | Bleu | RougeL | Fluency | Cite Qlty | Txt-Im Coher | Reas Logic | Factuality | Avg |
|---|---|---|---|---|---|---|---|---|---|---|---|---|---|---|---|---|---|
| Qwen-Plus | 3.6k | 316 | 70.2 | 62.5 | 66.1 | 36.2 | 53.1 | 43.1 | 55.4 | **0.169** | **0.318** | 4.35 | 3.28 | 3.57 | 3.51 | 3.44 | 3.63 |
| Qwen-Max | 3.6k | 426 | 71.7 | 66.9 | 69.3 | 39.7 | 51.5 | 44.8 | 58.9 | 0.165 | 0.315 | 4.42 | 3.47 | 3.71 | 3.64 | 3.59 | 3.77 |
| Qwen-QwQ-Plus | 3.6k | 1266 | 67.4 | 66.1 | 66.7 | 35.7 | 62.6 | 45.5 | 59.6 | 0.126 | 0.284 | 4.17 | 3.29 | 3.63 | 3.54 | 3.51 | 3.63 |
| Gemini-1.5-Pro | 3.6k | 290 | 66.8 | 72.9 | 69.7 | 32.1 | 63.0 | 41.9 | 56.2 | 0.126 | 0.262 | 3.59 | 2.62 | 3.13 | 2.82 | 3.01 | 3.03 |
| Gemini-2.0-Pro | 3.6k | 307 | 71.7 | 81.4 | 76.3 | 36.7 | 61.3 | 45.9 | 62.8 | 0.164 | 0.308 | 4.13 | 3.08 | 3.56 | 3.34 | 3.46 | 3.51 |
| Gemini-2.0-Flash | 3.6k | 283 | 66.0 | 71.3 | 68.5 | 30.6 | 65.1 | 41.6 | 54.4 | 0.134 | 0.277 | 3.84 | 2.75 | 3.21 | 3.00 | 3.13 | 3.19 |
| Gemini-2.0-Flash-Think | 3.6k | 275 | 72.0 | 73.6 | 72.8 | 37.4 | 60.5 | 46.2 | 61.0 | 0.133 | 0.272 | 4.14 | 3.04 | 3.54 | 3.27 | 3.35 | 3.47 |
| Gemini-2.5-Flash | 3.6k | 385 | 67.4 | 81.7 | 73.8 | 29.9 | **79.9** | 43.5 | 59.5 | 0.131 | 0.268 | 4.02 | 3.09 | 3.68 | 3.39 | 3.57 | 3.55 |
| Gemini-2.5-Pro | 3.6k | 387 | 71.3 | **87.5** | **78.6** | 35.7 | 78.5 | 49.1 | **65.1** | 0.142 | 0.281 | 4.25 | 3.35 | 3.94 | 3.64 | 3.77 | 3.79 |
| Claude-3.5-Sonnet | 3.8k | 348 | 65.2 | 77.5 | 70.8 | 33.7 | 76.6 | 46.8 | 57.4 | 0.122 | 0.276 | 4.30 | 3.11 | 3.60 | 3.50 | 3.50 | 3.60 |
| Grok-3-mini-beta | 3.3k | 315 | 75.2 | 77.8 | 76.5 | 38.4 | 71.5 | **49.9** | 64.6 | 0.127 | 0.261 | 4.21 | 3.24 | 3.73 | 3.40 | 3.57 | 3.63 |
| Grok-3-beta | 3.3k | 434 | 72.8 | 69.0 | 70.9 | 34.7 | 73.7 | 47.2 | 57.9 | 0.119 | 0.255 | 4.55 | 3.38 | 3.77 | 3.70 | 3.76 | 3.83 |
| GPT-4-turbo | 3.4k | 353 | 69.9 | 63.6 | 66.6 | 36.8 | 51.4 | 42.9 | 57.7 | 0.148 | 0.304 | 4.28 | 3.15 | 3.44 | 3.46 | 3.47 | 3.56 |
| GPT-4o-mini | 3.4k | 394 | 61.9 | 71.3 | 66.3 | 31.9 | 49.7 | 38.9 | 56.6 | 0.145 | 0.291 | 4.31 | 3.15 | 3.66 | 3.65 | 3.49 | 3.70 |
| GPT-4o | 3.4k | 353 | 66.9 | 67.1 | 67.0 | 37.0 | 57.2 | 44.9 | 57.2 | 0.160 | 0.313 | 4.29 | 3.37 | 3.65 | 3.56 | 3.59 | 3.69 |
| GPT-o3-mini | 3.4k | 623 | 71.2 | 66.0 | 68.5 | 33.9 | 49.1 | 40.1 | 57.0 | 0.146 | 0.304 | 3.46 | 2.73 | 3.23 | 2.93 | 3.13 | 3.10 |
| GPT-4.1-nano | 3.3k | 320 | 62.1 | 40.0 | 48.7 | 27.2 | 46.6 | 34.4 | 40.8 | 0.129 | 0.283 | 4.22 | 2.93 | 3.33 | 3.35 | 3.13 | 3.39 |
| GPT-4.1-mini | 3.4k | 411 | 66.8 | 80.6 | 73.0 | 30.6 | 68.8 | 42.3 | 61.0 | 0.137 | 0.283 | 4.46 | 3.45 | 3.98 | 3.81 | 3.78 | 3.90 |
| GPT-4.1 | 3.4k | 324 | **77.8** | 80.9 | 79.3 | **42.2** | 59.4 | 49.4 | **68.3** | 0.148 | 0.294 | **4.56** | 3.45 | **4.15** | **3.98** | **3.92** | **4.07** |

**Use using 20 quotes (8 images & 12 texts) as multimodal input sequence for VLM**

*Open-source Models*

| Method | In | Out | Img Prec | Img Rec | Img F₁ | Txt Prec | Txt Rec | Txt F₁ | F₁ | Bleu | RougeL | Fluency | Cite Qlty | Txt-Im Coher | Reas Logic | Factuality | Avg |
|---|---|---|---|---|---|---|---|---|---|---|---|---|---|---|---|---|---|
| Janus-Pro-7B | - | 154 | 0.0 | 0.0 | 0.0 | 0.0 | 0.0 | 0.0 | 0.0 | 0.000 | 0.110 | 0.00 | 0.10 | 0.00 | 0.00 | 0.00 | 0.02 |
| Qwen2.5-VL-3B-Inst | 8.7k | 265 | 42.5 | 0.5 | 1.0 | 22.8 | 3.0 | 5.3 | 1.2 | 0.105 | 0.283 | 4.07 | 1.07 | 1.49 | 2.45 | 2.17 | 2.25 |
| *- After Fine-tuning* | 8.7k | 243 | 74.1 | 64.0 | 68.6 | 52.2 | 5.4 | 9.8 | 55.5 | 0.139 | 0.341 | 4.40 | 3.03 | 3.44 | 3.15 | 2.75 | 3.36 |
| Qwen2.5-VL-7B-Inst | 8.7k | 128 | 58.0 | 14.5 | 23.2 | 31.3 | 11.0 | 16.3 | 16.6 | 0.069 | 0.273 | 4.05 | 1.75 | 1.89 | 2.36 | 2.29 | 2.47 |
| *- After Fine-tuning* | 8.7k | 249 | 76.6 | 68.7 | 72.4 | 44.0 | 3.3 | 6.5 | 58.6 | 0.199 | 0.355 | 4.57 | 3.26 | 3.70 | 3.48 | 3.19 | 3.64 |
| MiniCPM-o-2.6-8B | - | 1346 | 13.0 | 11.5 | 12.2 | 13.9 | 19.4 | 16.2 | 9.3 | 0.062 | 0.184 | 2.13 | 1.74 | 1.33 | 2.27 | 1.29 | 1.75 |
| InternVL2.5-8B | 17.1k | 182 | 38.1 | 38.9 | 38.5 | 16.8 | 2.3 | 4.1 | 33.0 | 0.085 | 0.269 | 3.41 | 1.74 | 2.17 | 2.18 | 1.95 | 2.29 |
| InternVL3-8B | 17.1k | 419 | 61.8 | 30.3 | 40.7 | 27.3 | 46.5 | 34.4 | 37.0 | 0.119 | 0.303 | 3.75 | 2.58 | 2.92 | 2.91 | 2.72 | 2.98 |
| *- After Fine-tuning* | 17.1k | 268 | 75.4 | 68.3 | 71.7 | 57.3 | 8.2 | 14.4 | 58.8 | 0.205 | 0.356 | 4.05 | 3.68 | 3.75 | 3.51 | 3.58 | 3.71 |
| InternVL3-9B | 17.2k | 287 | 72.4 | 52.4 | 60.8 | 33.9 | 25.9 | 29.3 | 50.9 | 0.146 | 0.303 | 3.97 | 2.75 | 3.17 | 2.99 | 2.69 | 3.12 |
| *- After Fine-tuning* | 17.2k | 283 | **77.7** | 68.7 | **72.9** | **60.6** | 10.8 | 18.4 | **60.3** | **0.210** | **0.362** | 4.22 | **3.81** | **3.91** | 3.72 | 3.70 | **3.87** |
| InternVL3-14B | 17.1k | 369 | 66.5 | 51.7 | 58.1 | 27.4 | 56.9 | 37.0 | 49.9 | 0.149 | 0.292 | 4.01 | 2.89 | 3.32 | 3.20 | 3.03 | 3.29 |
| InternVL2.5-26B | 17.1k | 198 | 56.8 | 26.6 | 36.3 | 21.9 | 5.4 | 8.6 | 25.8 | 0.094 | 0.291 | 3.76 | 1.65 | 2.01 | 2.41 | 2.18 | 2.40 |
| Qwen2.5-VL-32B-Inst | 7.0k | 755 | 57.4 | 32.2 | 41.2 | 26.8 | **73.2** | 39.3 | 36.2 | 0.086 | 0.227 | 4.20 | 3.32 | 3.70 | **3.69** | **3.71** | 3.73 |
| InternVL2.5-38B | 17.1k | 470 | 25.2 | 40.1 | 31.0 | 24.5 | 11.5 | 15.7 | 31.3 | 0.098 | 0.257 | 3.16 | 1.46 | 1.82 | 2.49 | 2.60 | 2.30 |
| InternVL3-38B | 17.1k | 359 | 67.7 | 51.0 | 58.2 | 33.1 | 64.7 | **43.8** | 53.9 | 0.155 | 0.301 | 4.08 | 3.07 | 3.47 | 3.36 | 3.27 | 3.45 |
| Qwen2.5-VL-72B-Inst | 7.1k | 320 | 68.9 | 72.1 | 70.5 | 36.0 | 52.9 | 42.8 | 57.5 | 0.151 | 0.298 | 4.15 | 3.08 | 3.43 | 3.35 | 3.33 | 3.47 |
| InternVL2.5-78B | 17.1k | 229 | 66.7 | 30.7 | 42.0 | 39.6 | 31.3 | 34.9 | 34.2 | 0.122 | 0.313 | 4.23 | 2.65 | 2.79 | 2.98 | 2.89 | 3.11 |
| InternVL3-78B | 17.1k | 292 | 69.6 | 68.6 | 69.1 | 35.1 | 54.5 | 42.7 | 59.8 | 0.165 | 0.312 | 4.08 | 3.05 | 3.47 | 3.31 | 3.19 | 3.42 |
| Llama4-Scout-17Bx16E | 11.6k | 339 | 60.0 | 44.0 | 50.8 | 29.1 | 40.9 | 34.0 | 38.9 | 0.128 | 0.288 | 3.91 | 2.57 | 2.96 | 3.07 | 3.01 | 3.10 |
| Llama4-Mave-17Bx128E | 11.6k | 320 | 69.6 | **74.2** | 71.8 | 41.8 | 30.8 | 35.5 | 58.6 | 0.151 | 0.308 | **4.25** | 3.29 | 3.63 | 3.55 | 3.61 | 3.67 |

*Proprietary Models*

| Method | In | Out | Img Prec | Img Rec | Img F₁ | Txt Prec | Txt Rec | Txt F₁ | F₁ | Bleu | RougeL | Fluency | Cite Qlty | Txt-Im Coher | Reas Logic | Factuality | Avg |
|---|---|---|---|---|---|---|---|---|---|---|---|---|---|---|---|---|---|
| Qwen-VL-Plus | 7.1k | 257 | 57.3 | 20.9 | 30.6 | 27.1 | 21.5 | 21.6 | 25.2 | 0.096 | 0.269 | 4.32 | 2.03 | 2.34 | 2.17 | 2.05 | 2.36 |
| Qwen-VL-Max | 7.1k | 206 | **78.4** | 45.9 | 57.9 | 33.5 | 39.3 | 36.2 | 46.8 | 0.124 | 0.308 | 4.17 | 3.01 | 3.32 | 3.14 | 3.13 | 3.35 |
| Qwen-QVQ-Max | 6.8k | 1137 | 63.5 | 6.8 | 12.2 | 34.0 | 13.2 | 19.1 | 12.3 | 0.106 | 0.290 | 4.53 | 2.44 | 2.77 | 3.61 | 3.45 | 3.36 |
| Gemini-1.5-Pro | 3.8k | 202 | 68.0 | 72.5 | 70.2 | 36.8 | 45.6 | 40.7 | 59.3 | 0.098 | 0.261 | 3.27 | 2.50 | 2.90 | 2.48 | 2.68 | 2.77 |
| Gemini-2.0-Pro | 3.8k | 265 | 69.1 | 82.4 | 75.1 | 36.0 | 61.3 | 45.3 | 62.0 | 0.148 | 0.292 | 3.91 | 2.87 | 3.33 | 3.12 | 3.30 | 3.31 |
| Gemini-2.0-Flash | 3.8k | 226 | 72.8 | 69.7 | 71.2 | 37.8 | 63.4 | 47.4 | 60.0 | 0.130 | 0.292 | 4.23 | 2.79 | 3.17 | 2.86 | 3.05 | 3.11 |
| Gemini-2.0-Flash-Think | 3.8k | 290 | 72.6 | 80.6 | 76.4 | 41.2 | 61.2 | 49.2 | **66.2** | 0.144 | 0.297 | 4.21 | 3.24 | 3.69 | 3.41 | 3.48 | 3.61 |
| Gemini-2.5-Flash | 3.7k | 362 | 72.2 | 80.7 | 76.2 | 34.3 | 70.4 | 46.1 | 62.4 | 0.139 | 0.284 | 4.24 | 3.28 | 3.82 | 3.66 | 3.79 | 3.76 |
| Gemini-2.5-Pro | 3.7k | 371 | 68.8 | **89.9** | 78.0 | 35.0 | **72.8** | 47.3 | 65.4 | 0.139 | 0.283 | 4.33 | 3.40 | 3.97 | 3.78 | 3.94 | 3.88 |
| Claude-3.5-Sonnet | 7.8k | 313 | 68.9 | 82.7 | 75.2 | 35.6 | 68.9 | 46.9 | 62.5 | 0.120 | 0.279 | 4.25 | 3.22 | 3.71 | 3.54 | 3.53 | 3.65 |
| GPT-4o-mini | 8.5k | 355 | 63.0 | 71.8 | 67.1 | 32.1 | 47.4 | 38.3 | 56.3 | 0.145 | 0.295 | 4.54 | 3.13 | 3.59 | 3.53 | 3.23 | 3.60 |
| GPT-4o | 6.4k | 347 | 60.2 | 83.4 | 70.0 | 35.2 | 58.1 | 43.8 | 62.6 | **0.157** | **0.315** | 4.39 | 3.42 | 3.74 | 3.58 | 3.58 | 3.74 |
| GPT-4.1-nano | 14.2k | 301 | 54.3 | 20.7 | 30.0 | 30.0 | 49.3 | 36.3 | 29.0 | 0.129 | 0.299 | 4.19 | 2.61 | 2.93 | 3.09 | 2.76 | 3.12 |
| GPT-4.1-mini | 9.8k | 474 | 62.0 | 85.1 | 71.7 | 30.6 | 72.0 | 43.0 | 61.2 | 0.132 | 0.285 | 4.41 | 3.48 | 3.98 | 3.87 | 3.88 | 3.92 |
| GPT-4.1 | 6.6k | 306 | 77.2 | 84.5 | **80.7** | **42.9** | 66.0 | **52.0** | **70.2** | 0.157 | 0.313 | **4.61** | **3.75** | **4.20** | **4.10** | **4.04** | **4.14** |

Table 3: Main results (**using 20 quotes as context**) for quote selection and multimodal answer generation. The best and second best scores are in **boldface** and underlined. Two most important columns: (i) Overall F₁ of both image/text quotes selection, and (ii) Average Scores of fluency, cite quality, text-image coherence, reasoning logic, and factuality for answer generation, are highlighted.

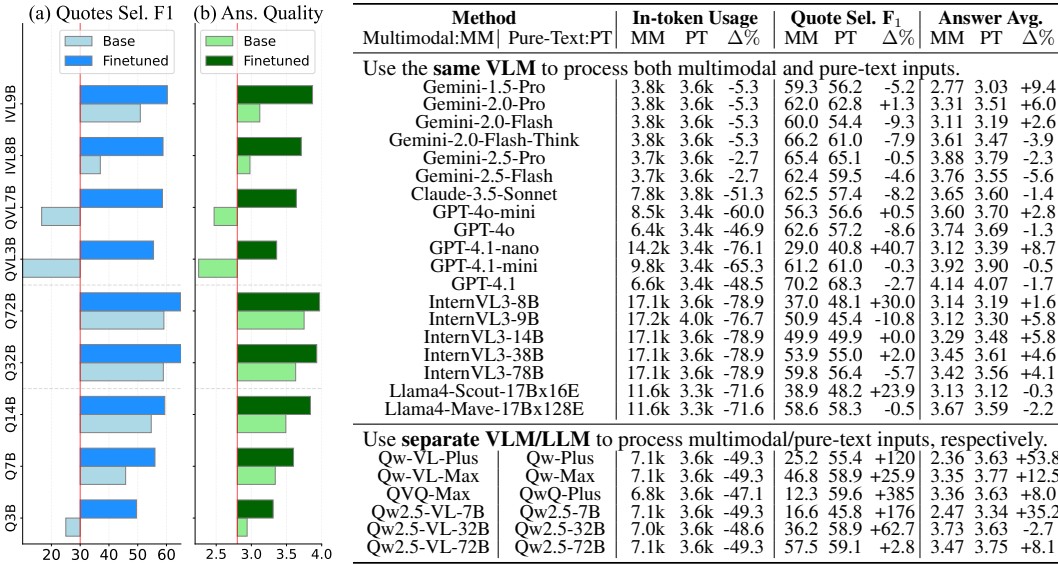

Figure 6: Performance difference: base/finetuned models.

Table 4: Using **20 quotes** for multimodal generation. $\Delta\%$ is calculated by values (PT-MM)/MM in percentage.

| Method Multimodal:MM \| Pure-Text:PT | In-token Usage MM | PT | $\Delta\%$ | Quote Sel. $F_1$ MM | PT | $\Delta\%$ | Answer Avg. MM | PT | $\Delta\%$ |
|---|---|---|---|---|---|---|---|---|---|
| **Use the same VLM to process both multimodal and pure-text inputs.** | | | | | | | | | |
| Gemini-1.5-Pro | 3.8k | 3.6k | -5.3 | 59.3 | 56.2 | -5.2 | 2.77 | 3.03 | +9.4 |
| Gemini-2.0-Pro | 3.8k | 3.6k | -5.3 | 62.0 | 62.8 | +1.3 | 3.31 | 3.51 | +6.0 |
| Gemini-2.0-Flash | 3.8k | 3.6k | -5.3 | 60.0 | 54.4 | -9.3 | 3.11 | 3.19 | +2.6 |
| Gemini-2.0-Flash-Think | 3.8k | 3.6k | -5.3 | 66.2 | 61.0 | -7.9 | 3.61 | 3.47 | -3.9 |
| Gemini-2.5-Pro | 3.7k | 3.6k | -2.7 | 65.4 | 65.1 | -0.5 | 3.88 | 3.79 | -2.3 |
| Gemini-2.5-Flash | 3.7k | 3.6k | -2.7 | 62.4 | 59.5 | -4.6 | 3.76 | 3.55 | -5.6 |
| Claude-3.5-Sonnet | 7.8k | 3.8k | -51.3 | 62.5 | 57.4 | -8.2 | 3.65 | 3.60 | -1.4 |
| GPT-4o-mini | 8.5k | 3.4k | -60.0 | 56.3 | 56.6 | +0.5 | 3.60 | 3.70 | +2.8 |
| GPT-4o | 6.4k | 3.4k | -46.9 | 62.6 | 57.2 | -8.6 | 3.74 | 3.69 | -1.3 |
| GPT-4.1-nano | 14.2k | 3.4k | -76.1 | 29.0 | 40.8 | +40.7 | 3.12 | 3.39 | +8.7 |
| GPT-4.1-mini | 9.8k | 3.4k | -65.3 | 61.2 | 61.0 | -0.3 | 3.92 | 3.90 | -0.5 |
| GPT-4.1 | 6.6k | 3.4k | -48.5 | 70.2 | 68.3 | -2.7 | 4.14 | 4.07 | -1.7 |
| InternVL3-8B | 17.1k | 3.6k | -78.9 | 37.0 | 48.1 | +30.0 | 3.14 | 3.19 | +1.6 |
| InternVL3-9B | 17.2k | 4.0k | -76.7 | 50.9 | 45.4 | -10.8 | 3.12 | 3.30 | +5.8 |
| InternVL3-14B | 17.1k | 3.6k | -78.9 | 49.9 | 49.9 | +0.0 | 3.29 | 3.48 | +5.8 |
| InternVL3-38B | 17.1k | 3.6k | -78.9 | 53.9 | 55.0 | +2.0 | 3.45 | 3.61 | +4.6 |
| InternVL3-78B | 17.1k | 3.6k | -78.9 | 59.8 | 56.4 | -5.7 | 3.42 | 3.56 | +4.1 |
| Llama4-Scout-17Bx16E | 11.6k | 3.3k | -71.6 | 38.9 | 48.2 | +23.9 | 3.13 | 3.12 | -0.3 |
| Llama4-Mave-17Bx128E | 11.6k | 3.3k | -71.6 | 58.6 | 58.3 | -0.5 | 3.67 | 3.59 | -2.2 |
| **Use separate VLM/LLM to process multimodal/pure-text inputs, respectively.** | | | | | | | | | |
| Qw-VL-Plus  Qw-Plus | 7.1k | 3.6k | -49.3 | 25.2 | 55.4 | +120 | 2.36 | 3.63 | +53.8 |
| Qw-VL-Max  Qw-Max | 7.1k | 3.6k | -49.3 | 46.8 | 58.9 | +25.9 | 3.35 | 3.77 | +12.5 |
| QVQ-Max  QwQ-Plus | 6.8k | 3.6k | -47.1 | 12.3 | 59.6 | +385 | 3.36 | 3.63 | +8.0 |
| Qw2.5-VL-7B  Qw2.5-7B | 7.1k | 3.6k | -49.3 | 16.6 | 45.8 | +176 | 2.47 | 3.34 | +35.2 |
| Qw2.5-VL-32B  Qw2.5-32B | 7.0k | 3.6k | -48.6 | 36.2 | 58.9 | +62.7 | 3.73 | 3.63 | -2.7 |
| Qw2.5-VL-72B  Qw2.5-72B | 7.1k | 3.6k | -49.3 | 57.5 | 59.1 | +2.8 | 3.47 | 3.75 | +8.1 |

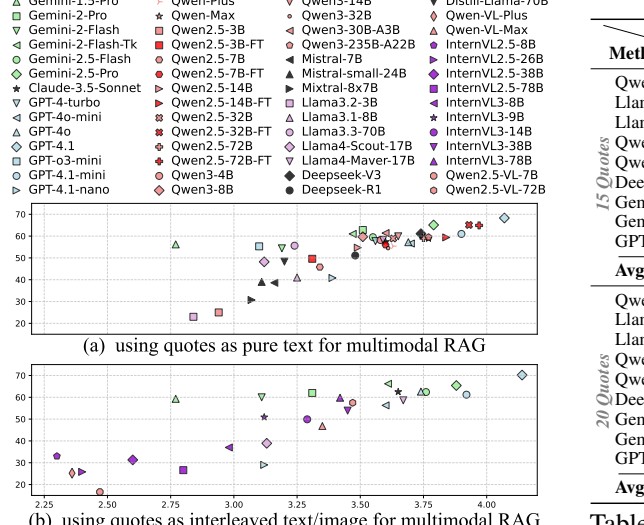

(a) using quotes as pure text for multimodal RAG

(b) using quotes as interleaved text/image for multimodal RAG

Figure 7: Scatter plots of models' answer quality and quote selection scores using 20 quotes as context.

| Metric Method | Image Quote $F_1$ VLM | OCR | $\Delta$ | Answer Avg. VLM | OCR | $\Delta$ |
|---|---|---|---|---|---|---|
| *15 Quotes* | | | | | | |
| Qwen2.5-7B-Inst | 59.8 | 49.6 | -10.2 | 3.37 | 3.15 | -0.22 |
| Llama3.1-8B-Inst | 61.1 | 52.4 | -8.7 | 3.33 | 3.19 | -0.14 |
| Llama3.3-70B-Inst | 71.8 | 64.1 | -7.7 | 3.14 | 3.08 | -0.06 |
| Qwen2.5-72B-Inst | 73.3 | 65.7 | -7.6 | 3.76 | 3.55 | -0.21 |
| Qwen-Max | 74.4 | 65.7 | -8.7 | 3.77 | 3.63 | -0.14 |
| Deepseek-V3 | 76.5 | 70.2 | -6.3 | 3.75 | 3.68 | -0.07 |
| Gemini-2.0-Pro | 77.4 | 74.9 | -2.5 | 3.50 | 3.41 | -0.09 |
| Gemini-2.0-Fl-Tk | 74.9 | 73.0 | -1.9 | 3.51 | 3.46 | -0.05 |
| GPT-4o | 71.6 | 69.4 | -5.9 | 3.73 | 3.65 | -0.08 |
| **Avg. results** | 71.7 | 65.0 | -6.7 | 3.54 | 3.42 | -0.12 |
| *20 Quotes* | | | | | | |
| Qwen2.5-7B-Inst | 53.5 | 43.5 | -10.0 | 3.34 | 3.15 | -0.19 |
| Llama3.1-8B-Inst | 52.2 | 45.8 | -6.4 | 3.25 | 3.16 | -0.09 |
| Llama3.3-70B-Inst | 65.1 | 60.6 | -4.5 | 3.24 | 3.13 | -0.11 |
| Qwen2.5-72B-Inst | 68.0 | 59.7 | -8.3 | 3.75 | 3.50 | -0.25 |
| Qwen-Max | 69.3 | 59.9 | -9.4 | 3.77 | 3.62 | -0.15 |
| Deepseek-V3 | 71.8 | 65.8 | -6.0 | 3.74 | 3.59 | -0.15 |
| Gemini-2.0-Pro | 77.0 | 70.9 | -6.1 | 3.51 | 3.32 | -0.18 |
| Gemini-2.0-Fl-Tk | 73.5 | 68.3 | -5.2 | 3.47 | 3.41 | -0.06 |
| GPT-4o | 66.4 | 63.8 | -2.6 | 3.69 | 3.59 | -0.10 |
| **Avg. results** | 66.3 | 59.8 | -6.5 | 3.53 | 3.39 | -0.14 |

Table 5: Quotes as Text: performance difference using VLM-text and OCR-text.

### 4.4 Multimodal vs Pure-text Quotes: Comparison and Analysis

In Table 4 and 9, we compare model performance when quotes are provided as either pure-text or multimodal inputs. Multimodal quotes significantly increase token usage, as images are typically encoded with more tokens. Interestingly, Gemini models maintain similar token usage across both modes, indicating efficient image encoding. Gemini, Claude, and GPT models demonstrate superior quote selection performance in the multimodal setting and comparable answer quality across both input types. In contrast, Qwen models perform significantly better in both quote selection and answer generation when using pure-text inputs. Smaller VLMs, compared to their LLM counterparts, struggle to effectively process long multimodal input sequences. For instance, the Qwen-7B and 32B LLMs achieve 175.9% and 62.7% higher $F_1$ scores for quote selection, respectively, compared to their equivalent VLMs. Further qualitative analysis is provided in Appendix F.2.

### 4.5 Multimodal Quotes as text: VLM-text vs OCR-text

We compare model performance using OCR-extracted text versus VLM-generated text, as shown in Table 5 (complete results in Table 12). Models utilizing VLM-text significantly outperform those

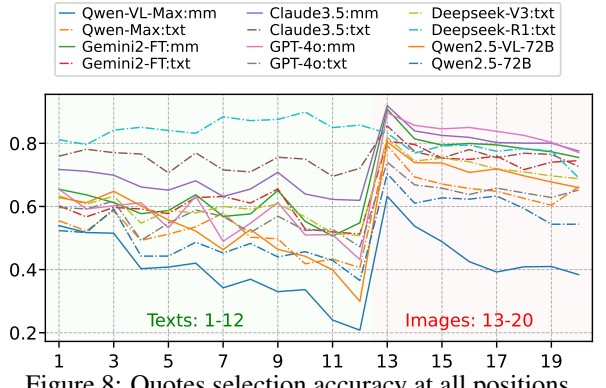

Figure 8: Quotes selection accuracy at all positions.

| Method | | Recall@10 | | Recall@15 | | Recall@20 | |
|---|---|---|---|---|---|---|---|
| | | Txt | Img | Txt | Img | Txt | Img |
| *Text* | DPR | 25.5 | 53.5 | 31.4 | 59.9 | 35.9 | 63.9 |
| | ColBERT | 37.4 | 64.1 | 42.8 | 69.1 | 46.0 | 72.8 |
| | BGE | 38.8 | 64.9 | 43.6 | 70.2 | 47.0 | 74.2 |
| | E5 | 41.7 | 63.5 | 46.4 | 69.1 | 49.5 | 73.7 |
| | Contriever | 37.7 | 64.1 | 42.8 | 69.8 | 46.8 | 73.4 |
| | GTE | 38.2 | 63.0 | 43.3 | 69.1 | 47.3 | 72.8 |
| *Visual* | DSE$_{wiki-ss}$ | 24.7 | 67.1 | 29.6 | 75.3 | 33.4 | 79.9 |
| | DSE$_{docmatix}$ | 25.6 | 65.7 | 30.1 | 75.0 | 33.7 | 78.2 |
| | ColPali | 27.3 | 68.2 | 32.6 | 77.5 | 35.2 | 81.2 |
| | ColQwen | 28.5 | 70.8 | 33.7 | 79.2 | 36.0 | 84.3 |
| *Hybrid* | ColP+ColB | 38.2 | 67.3 | 42.6 | 79.2 | 46.8 | 83.4 |
| | ColP+BGE | 39.2 | 67.3 | 43.5 | 79.2 | 47.7 | 83.4 |
| | ColQ+ColB | 38.2 | 68.5 | 42.6 | 81.0 | 46.8 | 85.2 |
| | ColQ+BGE | 39.2 | 68.5 | 43.5 | 81.0 | 47.7 | 85.2 |

Table 6: Retrieval Results.

| Model | Retriever | Query | Quote Retrieval Rec. | | | Quote Selection F$_1$ | | | Multimodal Answer Quality | | |
|---|---|---|---|---|---|---|---|---|---|---|---|
| | | | Text | Image | All | Text | Image | All | Bleu | RougeL | LLM-Judge |
| GPT-4.1 | perfect | - | - | - | - | 52.0 | 80.7 | 70.2 | 0.157 | 0.313 | 4.14 |
| GPT-4.1 | BGE | original | 34.6 | 77.8 | 71.0 | 34.2 | 60.8 | 54.4 | 0.137 | 0.299 | 3.53 |
| GPT-4.1 | BGE | clauses | 42.1 | 83.6 | 78.9 | 37.9 | 64.0 | 57.5 | 0.141 | 0.302 | 3.71 |
| GPT-4.1 | multiple | clauses | 49.5 | 86.8 | 84.9 | 41.4 | 65.6 | 59.9 | 0.141 | 0.303 | 3.79 |
| Gemini2.5-Flash | perfect | - | - | - | - | 46.1 | 76.2 | 62.4 | 0.139 | 0.284 | 3.76 |
| Gemini2.5-Flash | BGE | original | 34.6 | 77.8 | 71.0 | 27.5 | 55.6 | 47.7 | 0.124 | 0.280 | 3.21 |
| Gemini2.5-Flash | BGE | clauses | 42.1 | 83.6 | 78.9 | 30.9 | 59.2 | 50.4 | 0.125 | 0.281 | 3.39 |
| Gemini2.5-Flash | multiple | clauses | 49.5 | 86.8 | 84.9 | 34.3 | 60.3 | 51.8 | 0.124 | 0.281 | 3.42 |

Table 7: End-to-end RAG Results.

using OCR-text in both image quote selection and multimodal answer generation. This suggests that VLM-text preserves richer multimodal information compared to raw text extracted by OCR tools. As shown in Figure 5, the length of VLM-text is 0.5 times longer for tables and 2.8 times longer for figures, compared with OCR-text. While tables often contain structured text that are adequately captured by OCR, figures present more graphical and visual cues, causing OCR tools to struggle. Although VLM-text captures better multimodal information, it incurs additional overhead and latency.

## 4.6 Quotes Selection Analysis

In MMDocRAG, gold and noisy quotes are randomly mixed, resulting in an even distribution of gold quotes across all positions. Previous work [45] shows that LLMs tend to favor information at the start and end positions, often neglecting middle content. We therefore analyze quote selection accuracy by breaking it down into 20 positions, with indices 1–12 for text quotes and 13–20 for image quotes. As shown in Figure 8, gold quotes (especially image-based) placed in the first position have the highest likelihood of selection. Selection accuracy declines as the quote appears later in the sequence.

## 4.7 Quotes Retrieval Results

Our primary focus is multimodal generation, with fixed quotes used in previous experiments for reproducibility. Here, we assess whether current SOTA retrievers can accurately retrieve the correct gold quotes from long documents. As shown in Table 6, visual retrievers outperform text retrievers in image retrieval, while lagging behind text retrievers in text retrieval. The hybrid retrieval can leverage the strength of both text and visual retrievers. We show retrieval in long document is challenging.

## 4.8 End-to-end Multimodal RAG Analysis

While previous experiments use fixed quotes, real-world RAG systems must contend with imperfect retrieval. To bridge this gap and assess the robustness of multimodal generation under realistic conditions, we conduct end-to-end experiments that jointly evaluate retrieval and generation performance.

**Experimental Setup.** We evaluate four retrieval configurations: **(1) Perfect retriever** (upper bound): All gold quotes provided alongside noisy quotes, maintaining the 20-quote setting (8 images, 12 texts). **Single retriever**: BGE [83] with either **(2)** original questions or **(3)** expanded multi-clause queries. **(4) Multi-retriever ensemble**: Combination of BGE, Qwen3-0.6B [87], BM25 [68], and E5 [78] retrievers with query expansion. Note that for multi-clause queries and multi-retriever methods, we

consolidate top quotes from multi-retriever/clause via reranking using Qwen3-0.6B-reranker [87]. We focus on single-vector embedding models for compatibility with production vector databases (*e.g.,* Milvus), excluding multi-vector approaches like ColBERT [37] and ColQwen [21].

**Results and Analysis.** Table 7 presents performance across retrieval configurations. Three key observations emerge: *(i) Retrieval-generation correlation:* A clear positive correlation exists between retrieval recall and downstream performance. When retrieval recall drops from perfect to 71.0% using single BGE, GPT-4.1's quote selection $F1_1$ degrades from 70.2 to 54.4 (-22.5%), while answer quality drops from 4.14 to 3.53 (-14.7%). *(ii) Query expansion benefits:* Expanding queries into multi-clause formulations consistently improves retrieval recall (+7.9% absolute for BGE), which cascades into better generation performance. This suggests that comprehensive query understanding remains crucial for document-grounded multimodal generation. *(iii) Multi-retriever robustness:* Ensemble approaches achieve substantially higher recall (84.9%) compared to single retrievers (71.0-78.9%), narrowing the performance gap with perfect retrieval. Even leading models like GPT-4.1 and Gemini-2.5-Flash experience approximately 10% performance degradation under realistic retrieval conditions, highlighting the continued challenge of end-to-end multimodal RAG.

## 5   Related Work

**Interleaved Text-Image Generation** aims to produce coherent content mixing multiple images and text segments. This task is inherently challenging due to fundamental differences between modalities. Recent works [22, 73, 75] address this by combining diffusion models with LLMs for interleaved generation. Newer approaches treat images as part of the next-token prediction within autoregressive frameworks. Methods such as [8, 22, 74, 84] demonstrate end-to-end interleaved text-image generation via autoregressive training. However, these models mainly generate images from scratch, making them prone to hallucinations and noise, reflected in recent benchmarks [44, 81, 88].

**Multimodal RAG and Benchmarks.** Retrieval-Augmented Generation (RAG) retrieves relevant quotations as context for answer generation [39, 86]. Multimodal RAG (MRAG) extends RAG by retrieving and leveraging multimodal knowledge (*e.g.,* image-text pairs) for VQA [5, 43]. MuRAR [91] tackles source attribution by retrieving multimodal elements from webpage. $M^2$RAG [48] builds upon MuRAR by proposing a multi-stage image insertion framework that uses model multiple times during answer generation. Although MuRAR and $M^2$RAG enable multimodal answer generation, their benchmarks are limited to webpage domain and lack annotations of supporting evidence.

**DocVQA and DocRAG Benchmarks.** Early DocVQA benchmarks focus on single-page VQA, such as DocVQA [49], InfoVQA [50], and TAT-DQA [89]. To mitigate the limitation of single-page input, DUDE [38], MP-DocVQA [76], SildeVQA [72] extend context lengths to 5-20 pages. However, these tasks are yet to be document-level [16] understanding. Two most recent MMLongBench-Doc [47] and DocBench [92], formulate DocVQA as long-context tasks by inputting entire documents (averaging 50-70 pages). To address increasing document length, M3DocVQA [10], M-Longdoc [9], and MMDocIR [19] propose DocRAG tasks, incorporating evidence retrieval followed by answer generation over the retrieved multimodal evidence. To our best knowledge, no existing DocVQA or DocRAG benchmarks focus on multimodal interleaved generation.

## 6   Conclusion

In this paper, we presented `MMDocRAG`, a comprehensive benchmark for multimodal document question answering and retrieval-augmented generation (RAG). `MMDocRAG` features over 4,000 expert-annotated QA pairs with multimodal evidence chains, as well as novel evaluation metrics for both quote selection and interleaved multimodal answer generation. Through extensive benchmarking of 58 leading LLMs and VLMs along with multiple retrieval methods, we reveal that current models struggle with effective multimodal evidence selection and interleaved image-text answer generation, especially in noisy and diverse document scenarios. Our results indicate that while proprietary models show a significant lead over open-source models, fine-tuning and the use of high-quality visual descriptions can drive substantial improvements. Despite these advances, a significant performance gap remains between current systems and the requirements of comprehensive multimodal DocVQA/DocRAG tasks. We hope that `MMDocRAG` will inspire future research toward more effective and interpretable multimodal reasoning in document understanding and RAG.

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

## Appendix Overview

The appendix includes the following sections:

- **Appendix A**: Details the evaluation metrics for multimodal RAG, including (A.1) related work on multimodal generation, and (A.2) implementation details of the evaluation metrics.

- **Appendix B**: Provides supplementary experimental results, with (B.1) related results by using 15 quotes for multimodal generation, (B.2) comprehensive results comparing thinking and non-thinking modes, (B.3) comprehensive results comparing different models using OCR and LLM text for image quote representation, and fine-grained results by document type (B.4), question type (B.5), and evidence type (B.6).

- **Appendix C**: Presents implementation details, including (C.1) the deployment and inference of large models, (C.2) data preparation and model training procedures, and (C.3) deployment of text, visual, and hybrid retrievers.

- **Appendix D**: Shows six annotated examples that illustrate typical multimodal reasoning and integration patterns, facilitating understanding of `MMDocRAG`.

- **Appendix E**: Lists prompt instructions used in this work, including (E.1) prompts for constructing `MMDocRAG`, and (E.3) prompt messages for inference and evaluation of large models.

- **Appendix F**: Presents a qualitative study on the quality of multimodal answer generation based on existing and finetuned large models, comprising (F.1) error analysis for four typical errors, (F.2) performance comparison of VLM by using multimodal and pure-text quotes for multimodal generation, and (F.3) assessment of finetuning effectiveness.

- **Appendix G**: Discusses the license agreements for `MMDocRAG` and artifacts used to construct `MMDocRAG`.

## A  Evaluation Metric of Multimodal Answer Generation

This section provides more details about the evaluation metrics used for multimodal answer generation (see Section 4.1).

### A.1  Related Work of multimodal generation

Multimodal generation, particularly interleaved image-text sequence generation, involves generating outputs that integrate visual and textual information in a cohesive manner (see Section 5). This capability facilitate applications such as storytelling, question answering, and document comprehension. Recent benchmark, MM-Interleaved [75], MMIE [81], GATE Opening [88], and $M^2$RAG [48] provide comprehensive evaluations for multimodal generation. Commonly adopted metrics include fluency, relevance, image-text coherence, and content quality. These are evaluated through human annotation or automated scoring using large language models such as GPT-4. Specifically, fluency assesses the grammatical correctness and readability of text, relevance measures the alignment of generated content with the prompt, image-text coherence evaluates the logical connection between images and text, and content quality addresses the completeness and richness of the output. Our benchmark, `MMDocRAG`, adopts established metrics such as fluency, image-text coherence, and content quality. Additionally, we incorporate BLEU [59] and ROUGE-L [42] scores to quantitatively assess the lexical similarity between generated and gold answers.

However, existing benchmarks largely focus on end-to-end multimodal generation, and often overlook evaluation settings specific to the Multimodal RAG (see Section 5) paradigm, which requires models to read, select, and integrate multimodal evidence. To address this gap, our work extends multimodal generation evaluation to the RAG setting by: (i) introducing quantitative $F_1$-based metrics for image and text quote selection, and (ii) incorporating RAG-specific criteria such as citation quality, reasoning logic, and factuality. As a result, `MMDocRAG` offers a more balanced and reliable framework for evaluating multimodal RAG, ensuring thorough assessment of both generative and retrieval-augmented capabilities.

## A.2 Evaluation Metrics: details and implementations

To comprehensively evaluate model performance in multimodal Retrieval-Augmented Generation (RAG), we employ a combination of automatic and LLM-as-judge metrics covering quote selection accuracy, surface-level answer similarity, and qualitative answer quality.

**1. Quote Selection Metrics.** We explicitly measure the model's ability to select appropriate evidence by computing precision, recall, and $F_1$ scores for both text and image quotes. Formally, given a predicted set of quotes $\mathcal{P}$ (either image or text) and the ground truth set $\mathcal{G}$, we define:

$$\text{Precision} = \frac{|\mathcal{P} \cap \mathcal{G}|}{|\mathcal{P}|}, \qquad \text{Recall} = \frac{|\mathcal{P} \cap \mathcal{G}|}{|\mathcal{G}|}, \qquad F_1 = 2 \cdot \frac{\text{Precision} \times \text{Recall}}{\text{Precision} + \text{Recall}} \qquad (1)$$

We extract quotes from the model's answer using regular expressions (e.g., text quotes indicated by " [i] " and image quotes by " " patterns). $F_1$ is calculated separately for text and image quotes, then averaged to yield an overall quote selection $F_1$. This directly benchmarks the model's capability to differentiate gold evidence from noisy quotes.

**2. Surface-level Similarity Metrics.** To assess how closely model-generated answers match the reference answers in content, we employ BLEU and ROUGE-L, two widely-used surface-level (lexical) similarity metrics: (i) **BLEU** (Bilingual Evaluation Understudy) computes $n$-gram overlap between the generated text $C$ and reference text $R$. For a maximum $n$-gram length $N$, BLEU is computed by:

$$\text{BLEU} = BP \cdot \exp\left(\sum_{n=1}^{N} w_n \log p_n\right), \qquad \text{where } BP = \begin{cases} 1, & \text{if } c > r \\ \exp\left(1 - \frac{r}{c}\right), & \text{if } c \leq r \end{cases} \qquad (2)$$

where $p_n$ is the modified precision for $n$-grams, $w_n$ is the weight for each $n$ (often $\frac{1}{N}$), and $BP$ is a brevity penalty accounting for length mismatch. (ii) **ROUGE-L** (Recall-Oriented Understudy for Gisting Evaluation) focuses on the Longest Common Subsequence (LCS) between the generated and reference answers. ROUGE-L combines recall and precision using:

$$\text{ROUGE-L} = \frac{(1 + \beta^2) \cdot R_{LCS} \cdot P_{LCS}}{R_{LCS} + \beta^2 P_{LCS}}, \quad P_{LCS} = \frac{LCS(\text{Gen}, \text{Ref})}{|\text{Gen}|}, \; R_{LCS} = \frac{LCS(\text{Gen}, \text{Ref})}{|\text{Ref}|}$$
$$(3)$$

where $R_{LCS}$ and $P_{LCS}$ are the recall and precision based on LCS length, and $|\cdot|$ refers to the length of generated or reference answer. $\beta$ is typically set to favor recall ($\beta = 1.2$ by default).

While effective for surface-level comparison, both BLEU and ROUGE-L are limited in capturing deeper semantic or cross-modal relationships, especially in long or highly interleaved multimodal contexts. We supplement them with task-specific metrics and human-aligned evaluation.

**3. LLM-as-Judge Evaluation Criteria.** For qualitative assessment, we utilize large language models to score generated answers on five key aspects:

- **Fluency**: Assesses grammatical correctness, readability, and natural flow. High fluency indicates the response is smooth and easy to follow.
- **Citation Quality**: Evaluates the correctness and contextual appropriateness of both image and text citations, ensuring that references effectively support the narrative.
- **Text-Image Coherence**: Measures the integration and consistency between textual and visual information. The answer should present images and text in a synergistic manner.
- **Reasoning Logic**: Examines the logical structure, clarity of argument, and progression from evidence to conclusion.
- **Factuality**: Ensures the answer is factually accurate, aligning with the underlying evidence provided in the ground-truth answer.

Each criterion is scored independently to promote thorough and unbiased qualitative judgment, providing a nuanced view of answer quality beyond automated metrics. Refer to Figure 24 for the detailed prompt used for LLM-as-Judge evaluation.

# B  Supplementary Experimental Results

## B.1  Main results by using 15 quotes for Multimodal Generation

We conduct experiments on two main settings: using 15 or 20 quotes for multimodal RAG. However, due to limited pages, we include the results of 60 off-the-shelf and 5 finetuned models using 15 quotes in Figure 8.

Moreover, we report the performance difference of models by using 15 quotes as either multimodal or pure-text input sequence, as shown in Figure 9. This serves as extended experimental results to complement the comparison and analysis in Section 4.4 (Multimodal vs Pure-text Quotes). Observe that the performance difference on 15 and 20 quotes exhibit similar patterns. It is interesting to note for advanced proprietary VLMs that the degradation switching to pure-text quotes become larger in quotes-15 setting, indicating current advanced proprietary VLMs become much smarter by taking less image quotes in its inputs. Similarly for open-source and smaller properitary VLMs, the performance increase by switching to pure-text quotes become smaller in quotes-15 setting.

## B.2  Comprehensive Results and Comparison Between Thinking and Non-Thinking Modes

**Thinking mode** refers to settings in which the model performs step-by-step reasoning before generating a final answer [62], making it well-suited for complex tasks requiring deeper reasoning. In contrast, **non-thinking mode** directs the model to provide rapid, near-instant responses, which is preferable for simple questions where speed is prioritized over depth. As discussed in Section 4.3, models operating in thinking mode generally consume significantly more output tokens and often yield inferior results compared to their non-thinking counterparts. Table 10 details the performance of the models with thinking mode enabled and disabled. Table 11 further compares model performance with explicit reasoning (thinking) and direct answering (non-thinking), using either the same model or closely matched variants. Our main findings are as follows:

- **Output token efficiency.** Disabling thinking mode typically reduces output token consumption by 50% to 80%, indicating that step-by-step reasoning substantially increases both the length of generated sequences and response latency.
- **Significance for reasoning-centered models.** For models explicitly trained for reasoning (*e.g.,* the Qwen3 series), disabling thinking mode consistently degrades performance.
- **Comparison of model series.** Deepseek-R1 underperformes compared to their non-thinking counterpart (*i.e.,* Deepseek-V3). Among Qwen models, smaller Qwen3 variants (4–14B) outperform Qwen2.5 models at comparable sizes, whereas larger Qwen3 models ((>32)B) are outperformed by their Qwen2.5 counterparts (32–72B).
- **R1-style post-training strategies.** The post-training strategy adopted by Deepseek-R1, which combines Supervised Fine-Tuning (SFT) and Group Robust Policy Optimization (GRPO) [70], can be effectively applied to models such as Qwen2.5-32B and Llama3-70B to enhance performance in multimodal generation tasks.
- **Multimodal Reasoning.** Different from other thinking models, Qwen-QVQ-Max performs reasoning based on multimodal inputs. By disabling thinking mode, QVQ-Max generates almost same amount of output tokens, achieving significant performance increase on quotes selection.

## B.3  Full results by using OCR and LLM text

In section 4.5, we analyze the performance difference by using OCR-text and VLM-text. The complete results (with more fine-grained scores breakdown) of quote selection and interleaved answer generation is illustrated in Figure 12.

## B.4  Fine-grained Results by Document Domains

Beyond main results (Section 4.3) on `MMDocRAG`, we present fine-grained results breakdown by domains. As illustrated in Figure 9, different models exhibit distinct performance patterns across various document types. Our findings include:

- All models achieve the highest performance in the "Workshop" and "Others" categories. This is attributed to the typically simpler images in "Workshop" documents, which often resemble

PowerPoint presentations with single elements. In contrast, models perform worst on the "Brochure" category, due to the prevalence of complex images and non-textual information.

- Advanced VLMs consistently achieve higher and more balanced scores across document types, especially in "Brochure" and "Academic" categories. This indicates that VLMs possess a greater capacity to integrate visual content, while LLMs, limited by reliance on image descriptions, underperform in visually complex settings.

- Answer quality shows a positive correlation with the $F_1$ score of quotes selection, especially in the "Brochure" and "Workshop" categories. The $F_1$ score largely reflects image understanding and evidence selection, whereas answer quality measures the model's generation ability based on the selected evidence.

- The GPT series exhibit balanced performance across both quote selection and answer quality. Gemini and Claude models excel in quote selection but lag in answer quality, suggesting a relative strength in reasoning over generation. In the Qwen series, the LLM with 72B parameters performs well, but its VLM counterpart shows a notable drop, indicating that visual processing remains a challenge for this series.

### B.5 Fine-grained Results by Question Types

Beyond main results (Section 4.3) on `MMDocRAG`, we present fine-grained results breakdown by question types. As illustrated in Figure 10, different models exhibit distinct performance patterns across various question types. Our findings include:

- All models achieve highest performance in "Descriptive" and "Comparative" categories, attributed to their straightforward information extraction requirements. Models perform worst on "Interpretative" and "Inferential" categories due to increased reasoning complexity.

- Advanced VLMs (GPT-4.1, Gemini2.5-pro) consistently achieve higher and more balanced scores across question types, especially in complex reasoning categories. This indicates superior multi-step reasoning capacity compared to smaller models that show pronounced degradation with increased question complexity.

- Answer quality positively correlates with $F_1$ score of quote selection across all question types, with strongest correlation in "Analytical" and "Comparative" categories. $F_1$ score reflects evidence identification ability while answer quality measures generation capability from selected evidence.

- GPT-4.1 exhibits the most balanced performance across both metrics, maintaining high scores even for complex questions. Gemini2.5-pro excels in "Descriptive" tasks, Claude-3.5-sonnet shows challenges in complex reasoning, and Llama4-Mave-17Bx128E displays the most constrained performance envelope across all question types.

### B.6 Fine-grained Results by Evidence Types

Beyond main results (Section 4.3) on `MMDocRAG`, we present fine-grained results breakdown by evidence types. Figure 11 reveals how different evidence configurations impact model performance in multimodal RAG tasks. Our analysis yields several key findings:

- **Single vs. Multiple Image Evidence:** All models consistently achieve higher $F_1$ scores and answer quality when questions require evidence from a single image rather than multiple images. This pattern indicates that synthesizing information across multiple visual sources presents a significant challenge for current VLMs.

- **Single vs. Multiple Page Evidence:** Questions with evidence contained within a single page consistently outperform those requiring multi-page evidence across all models. This suggests that information gathering and consolidation across document boundaries remains a substantial bottleneck.

- **Single vs. Cross-Modal Evidence:** Unlike the previous patterns, cross-modal evidence preferences vary by model architecture. GPT-4.1 and Llama4-17Bx128 perform better with single-modal evidence, while Gemini2.5-pro and Claude-3.5-sonnet show superior performance with cross-modal evidence. This divergence reflects fundamental differences in how these models handle modality fusion and integration.

- **Overall Model Performance:** GPT-4.1 maintains the highest performance across all evidence configurations, demonstrating robust scalability as evidence complexity increases. Gemini2.5-pro

shows particularly strong gains in cross-modal settings, while Claude-3.5-sonnet and Llama4-17Bx128 exhibit more constrained performance envelopes, with Llama4 showing the most limited adaptability to evidence complexity variations.

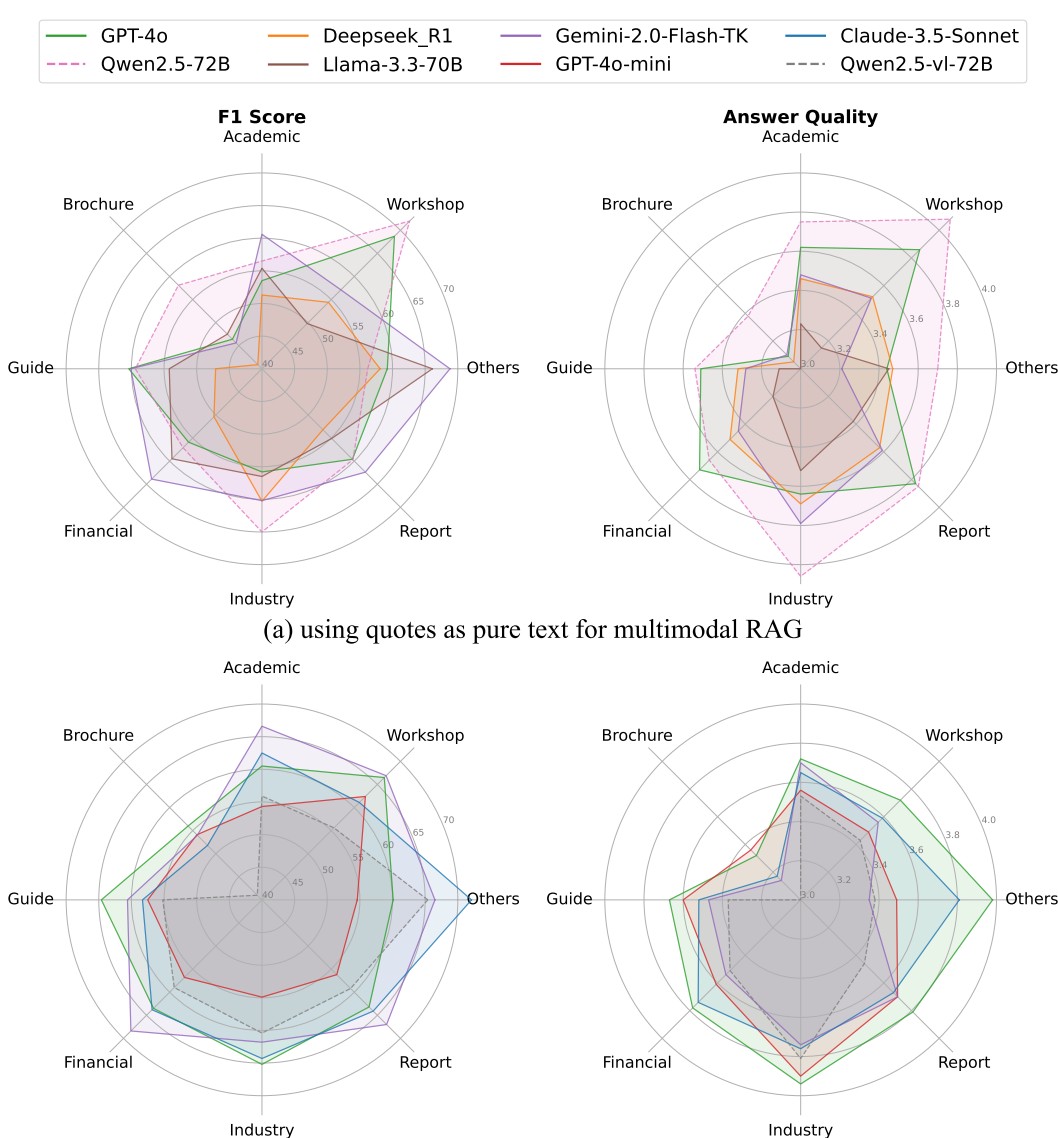

Figure 9: The fine-grained (by document domains) results of 8 representative large models in two settings: using 20 quotes as either pure-text or interleaved manner. We show the $F_1$ score of quotes selection (ranging from 40 to 70) and answer quality (ranging from 3.0 to 4.0).

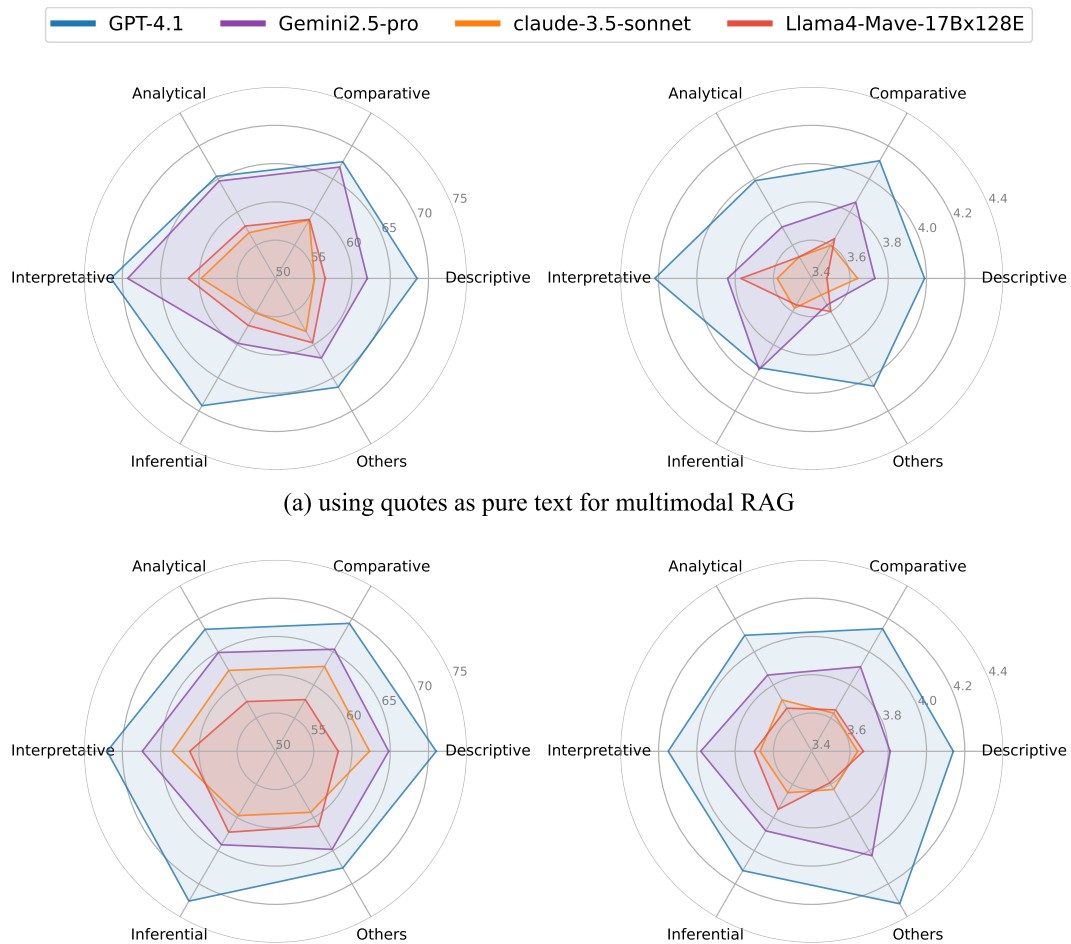

(a) using quotes as pure text for multimodal RAG

(b) using quotes as interleaved text/image for multimodal RAG

Figure 10: The fine-grained (by question types) results of 4 representative large models in two settings: using 20 quotes as either pure-text or interleaved manner. We show the $F_1$ score of quotes selection (ranging from 50 to 75) and answer quality (ranging from 3.4 to 4.4).

| Method | In | Out | Image Quotes Prec | Rec | F$_1$ | Text Quotes Prec | Rec | F$_1$ | F$_1$ | Bleu | Rou-geL | Flu-ency | Cite Qlty. | Txt-Im Coher. | Reas. Logic | Fact-uality | Avg |
|---|---|---|---|---|---|---|---|---|---|---|---|---|---|---|---|---|---|
| *Use using 15 quotes (5 images & 10 texts) as* **pure-text** *input sequence for both LLM and VLM* | | | | | | | | | | | | | | | | | |
| Qwen2.5-3B-Inst | 2.7k | 422 | 60.7 | 28.3 | 38.6 | 11.0 | 14.1 | 12.4 | 29.7 | 0.125 | 0.272 | 3.98 | 2.59 | 2.88 | 2.85 | 2.61 | 2.98 |
| *- After Fine-tuning* | 3.6k | 286 | 74.0 | 63.5 | 68.4 | 35.8 | 1.1 | 2.2 | 53.1 | 0.183 | 0.339 | 4.40 | 2.96 | 3.36 | 3.04 | 2.64 | 3.28 |
| Llama3.2-3B-Inst | 2.6k | 381 | 51.7 | 36.1 | 42.5 | 21.1 | 32.3 | 25.5 | 29.4 | 0.095 | 0.248 | 3.34 | 2.02 | 2.38 | 2.40 | 2.33 | 2.49 |
| Qwen3-4B (think) | 2.7k | 1057 | 74.1 | 67.9 | 70.9 | 37.2 | 45.5 | 40.9 | 59.8 | 0.139 | 0.301 | 4.27 | 3.16 | 3.67 | 3.50 | 3.47 | 3.61 |
| Qwen2.5-7B-Inst | 2.7k | 304 | 72.3 | 51.0 | 59.8 | 36.6 | 28.8 | 32.3 | 48.4 | 0.160 | 0.311 | 4.25 | 2.99 | 3.31 | 3.25 | 3.06 | 3.37 |
| *- After Fine-tuning* | 2.7k | 297 | 72.9 | 67.2 | 69.9 | 44.8 | 3.5 | 6.5 | 56.7 | 0.201 | 0.352 | 4.60 | 3.47 | 3.78 | 3.41 | 3.06 | 3.67 |
| Mistral-7B-Inst | 3.0k | 447 | 62.5 | 54.9 | 58.5 | 24.9 | 48.0 | 32.8 | 43.5 | 0.111 | 0.253 | 3.52 | 2.41 | 2.86 | 2.69 | 2.56 | 2.81 |
| Llama3.1-8B-Inst | 2.6k | 423 | 62.2 | 60.0 | 61.1 | 27.6 | 42.9 | 33.6 | 46.0 | 0.116 | 0.257 | 3.62 | 2.47 | 2.87 | 2.78 | 2.79 | 2.91 |
| Qwen3-8B (think) | 2.7k | 992 | 77.9 | 72.9 | 75.3 | 38.7 | 61.0 | 47.3 | 64.0 | 0.140 | 0.303 | 4.15 | 3.13 | 3.57 | 3.40 | 3.32 | 3.51 |
| InternVL3-8B | 2.7k | 379 | 68.6 | 60.7 | 64.4 | 33.1 | 36.0 | 34.5 | 52.7 | 0.152 | 0.294 | 3.93 | 2.75 | 3.17 | 3.10 | 3.00 | 3.19 |
| InternVL3-9B | 3.0k | 404 | 77.8 | 46.2 | 58.0 | 34.8 | 32.8 | 33.8 | 48.1 | 0.158 | 0.300 | 4.11 | 2.91 | 3.33 | 3.33 | 3.12 | 3.34 |
| Qwen2.5-14B-Inst | 2.7k | 356 | 77.6 | 61.9 | 68.9 | 39.1 | 48.6 | 43.4 | 59.6 | 0.151 | 0.298 | 4.28 | 3.13 | 3.47 | 3.33 | 3.29 | 3.50 |
| *- After Fine-tuning* | 2.7k | 296 | 76.8 | 73.4 | 75.1 | 55.9 | 7.2 | 12.7 | 61.5 | 0.217 | 0.370 | 4.70 | 3.70 | 4.02 | 3.69 | 3.38 | 3.90 |
| Qwen3-14B (think) | 2.7k | 891 | 77.8 | 69.9 | 73.7 | 39.3 | 58.4 | 47.0 | 62.2 | 0.143 | 0.307 | 4.29 | 3.25 | 3.66 | 3.59 | 3.47 | 3.65 |
| InternVL3-14B | 2.7k | 390 | 77.7 | 47.7 | 59.1 | 32.3 | 58.4 | 41.6 | 51.4 | 0.156 | 0.300 | 4.19 | 3.05 | 3.47 | 3.43 | 3.35 | 3.50 |
| Mistral-Small-24B-Inst | 2.8k | 383 | 57.1 | 53.6 | 55.3 | 25.5 | 50.0 | 33.8 | 42.7 | 0.092 | 0.236 | 2.34 | 1.81 | 2.16 | 1.91 | 1.93 | 2.03 |
| Qwen3-30B-A3B | 2.7k | 949 | 78.6 | 72.9 | 75.7 | 40.1 | 64.8 | 49.5 | 64.8 | 0.149 | 0.308 | 4.24 | 3.19 | 3.66 | 3.54 | 3.47 | 3.62 |
| Qwen2.5-32B-Inst | 2.7k | 316 | 76.1 | 73.8 | 75.0 | 44.8 | 33.8 | 38.5 | 63.0 | 0.162 | 0.309 | 4.41 | 3.34 | 3.67 | 3.52 | 3.44 | 3.68 |
| *- After Fine-tuning* | 2.7k | 286 | 78.6 | 74.2 | 76.3 | 62.6 | 21.7 | 32.2 | 65.5 | **0.224** | **0.376** | 4.73 | **3.71** | **4.08** | **3.77** | 3.46 | **3.95** |
| Qwen3-32B (think) | 2.7k | 884 | 78.4 | 59.8 | 67.8 | 37.4 | 67.0 | 48.0 | 56.5 | 0.137 | 0.301 | 4.30 | 3.23 | 3.63 | 3.56 | 3.46 | 3.63 |
| Mistral-8x7B-Inst | 3.0k | 286 | 64.4 | 38.6 | 48.3 | 30.2 | 26.8 | 28.4 | 34.3 | 0.103 | 0.250 | 3.27 | 2.17 | 2.50 | 2.45 | 2.33 | 2.54 |
| InternVL-38B | 2.7k | 341 | 73.3 | 56.8 | 64.0 | 35.6 | 68.2 | 46.8 | 57.3 | 0.160 | 0.307 | 4.30 | 3.22 | 3.64 | 3.53 | 3.41 | 3.62 |
| Llama3.3-70B-Inst | 2.7k | 434 | 59.8 | **89.5** | 71.8 | 32.2 | 70.4 | 44.2 | 58.5 | 0.120 | 0.263 | 3.73 | 2.72 | 3.10 | 2.98 | 3.18 | 3.14 |
| Qwen2.5-72B-Inst | 2.7k | 367 | **80.7** | 67.1 | 73.3 | 42.1 | 50.9 | 46.1 | 62.9 | 0.175 | 0.326 | 4.50 | 3.39 | 3.73 | 3.65 | 3.53 | 3.76 |
| *- After Fine-tuning* | 2.7k | 287 | 77.6 | 75.2 | 76.4 | 61.5 | 24.8 | 35.4 | 65.8 | **0.224** | **0.376** | **4.74** | 3.70 | **4.11** | **3.79** | 3.50 | **3.97** |
| InternVL3-78B | 2.7k | 373 | 72.2 | 73.7 | 73.0 | 34.3 | 69.1 | 45.8 | 59.3 | 0.158 | 0.302 | 4.23 | 3.10 | 3.56 | 3.50 | 3.42 | 3.56 |
| Qwen3-235B-A22B | 2.7k | 1068 | 77.3 | 71.8 | 74.4 | 38.2 | 64.9 | 48.1 | 62.9 | 0.137 | 0.295 | 4.33 | 3.35 | 3.79 | 3.69 | **3.61** | 3.75 |
| Deepseek-V3 | 2.7k | 239 | 76.0 | 76.9 | 76.5 | 41.5 | 63.8 | 50.3 | 64.6 | 0.173 | 0.341 | 4.54 | 3.33 | 3.74 | 3.63 | 3.54 | 3.75 |
| Deepseek-R1 | 2.6k | 953 | 72.6 | 80.8 | 76.5 | 33.8 | 70.4 | 45.7 | 62.1 | 0.116 | 0.271 | 4.16 | 3.18 | 3.57 | 3.31 | 3.30 | 3.50 |
| - Distill-Qwen-32B | 2.8k | 737 | 72.1 | 48.4 | 58.0 | 42.0 | 35.3 | 38.4 | 47.6 | 0.143 | 0.310 | 4.31 | 2.83 | 3.21 | 3.38 | 3.29 | 3.40 |
| - Distill-Llama-70B | 2.6k | 685 | 73.1 | 55.6 | 63.1 | 42.7 | 45.6 | 44.1 | 54.2 | 0.148 | 0.315 | 4.37 | 3.07 | 3.48 | 3.51 | 3.42 | 3.57 |
| Llama4-Scout-17Bx16E | 2.5k | 425 | 63.8 | 68.3 | 66.0 | 30.9 | 58.0 | 40.3 | 52.8 | 0.132 | 0.272 | 3.80 | 2.75 | 3.13 | 3.09 | 3.09 | 3.17 |
| Llama4-Mave-17Bx128E | 2.5k | 370 | 73.4 | 81.5 | **77.2** | 40.2 | 57.5 | 47.3 | 63.0 | 0.152 | 0.300 | 4.04 | 3.16 | 3.56 | 3.48 | 3.60 | 3.57 |
| Qwen-Plus | 2.7k | 306 | 74.4 | 66.4 | 70.1 | 39.8 | 56.4 | 46.6 | 59.1 | **0.172** | **0.322** | 4.35 | 3.24 | 3.56 | 3.50 | 3.46 | 3.62 |
| Qwen-Max | 2.7k | 406 | 76.9 | 72.0 | 74.4 | 41.9 | 53.7 | 47.1 | 61.9 | 0.168 | 0.319 | 4.42 | 3.46 | 3.74 | 3.64 | 3.59 | 3.77 |
| Qwen-QwQ-Plus | 2.7k | 1369 | 74.0 | 72.0 | 73.0 | 37.1 | 64.8 | 47.2 | 62.1 | 0.128 | 0.286 | 4.18 | 3.31 | 3.66 | 3.56 | 3.55 | 3.65 |
| Gemini-1.5-Pro | 2.8k | 288 | 70.3 | 73.9 | 72.1 | 33.4 | 62.4 | 43.5 | 57.8 | 0.125 | 0.261 | 3.61 | 2.61 | 3.14 | 2.82 | 2.98 | 3.03 |
| Gemini-2.0-Pro | 2.8k | 307 | 75.7 | 79.2 | 77.4 | 38.5 | 64.4 | 48.2 | 63.5 | 0.161 | 0.302 | 4.13 | 3.05 | 3.56 | 3.31 | 3.45 | 3.50 |
| Gemini-2.0-Flash | 2.8k | 282 | 67.7 | 72.2 | 69.9 | 32.5 | 68.5 | 44.1 | 56.0 | 0.132 | 0.274 | 3.85 | 2.74 | 3.22 | 3.00 | 3.15 | 3.19 |
| Gemini-2.0-Flash-Think | 2.8k | 270 | 76.5 | 73.3 | 74.9 | 38.8 | 62.3 | 47.8 | 62.2 | 0.132 | 0.270 | 4.13 | 3.07 | 3.63 | 3.30 | 3.43 | 3.51 |
| Gemini-2.5-Flash | 2.7k | 370 | 73.9 | 83.5 | 78.4 | 32.0 | **80.1** | 45.7 | 61.1 | 0.134 | 0.270 | 4.02 | 3.08 | 3.63 | 3.40 | 3.61 | 3.55 |
| Gemini-2.5-Pro | 2.7k | 380 | 77.6 | **89.5** | **83.1** | 37.0 | 79.5 | 50.5 | 66.6 | 0.145 | 0.283 | 4.27 | 3.45 | 3.91 | 3.73 | **3.86** | 3.84 |
| Claude-3.5-Sonnet | 2.9k | 344 | 71.6 | 83.3 | 77.0 | 35.7 | 78.5 | 49.1 | 61.6 | 0.122 | 0.277 | 4.31 | 3.12 | 3.63 | 3.55 | 3.54 | 3.63 |
| Grok-3-mini-beta | 2.5k | 313 | 80.1 | 83.0 | 81.5 | 40.5 | 74.4 | **52.4** | 67.2 | 0.129 | 0.263 | 4.24 | 3.23 | 3.74 | 3.44 | 3.59 | 3.65 |
| Grok-3-beta | 2.5k | 432 | 77.6 | 76.0 | 76.8 | 37.2 | 77.4 | 50.2 | 61.2 | 0.121 | 0.256 | **4.56** | 3.37 | 3.77 | 3.72 | 3.79 | 3.84 |
| GPT-4-turbo | 2.5k | 348 | 77.5 | 72.1 | 74.7 | 40.5 | 54.2 | 46.3 | 62.5 | 0.153 | 0.308 | 4.32 | 3.19 | 3.52 | 3.51 | 3.51 | 3.61 |
| GPT-4o-mini | 2.6k | 392 | 67.5 | 78.0 | 72.4 | 34.4 | 52.0 | 41.4 | 59.9 | 0.143 | 0.292 | 4.54 | 3.11 | 3.66 | 3.64 | 3.50 | 3.69 |
| GPT-4o | 2.6k | 386 | 70.9 | 80.3 | 75.3 | 40.3 | 61.4 | 48.7 | 64.1 | 0.156 | 0.307 | 4.33 | 3.41 | 3.67 | 3.60 | 3.64 | 3.73 |
| GPT-o3-mini | 2.6k | 618 | 74.3 | 69.0 | 71.5 | 36.0 | 52.2 | 42.7 | 59.3 | 0.151 | 0.306 | 3.43 | 2.77 | 3.21 | 2.97 | 3.14 | 3.11 |
| GPT-4.1-nano | 2.5k | 322 | 69.5 | 46.1 | 55.5 | 30.8 | 44.0 | 37.5 | 45.1 | 0.131 | 0.287 | 4.24 | 2.99 | 3.42 | 3.39 | 3.26 | 3.46 |
| GPT-4.1-mini | 2.5k | 400 | 73.5 | 83.3 | 78.1 | 34.7 | 71.1 | 46.6 | 63.7 | 0.139 | 0.284 | 4.48 | 3.45 | 3.98 | **3.82** | 3.78 | 3.90 |
| GPT-4.1 | 2.5k | 315 | **82.1** | 83.0 | 82.6 | **44.7** | 59.5 | 51.1 | 70.6 | 0.149 | 0.295 | 4.55 | **3.69** | **4.14** | **3.99** | **3.93** | **4.06** |
| *Use using 15 quotes (5 images & 10 texts) as* **multimodal** *input sequence for VLM* | | | | | | | | | | | | | | | | | |
| Janus-Pro-7B | - | 131 | 25.0 | 0.1 | 0.1 | 10.3 | 0.9 | 1.7 | 0.2 | 0.010 | 0.107 | 0.10 | 0.30 | 0.10 | 0.10 | 0.50 | 0.34 |
| Qwen2.5-VL-7B-Inst | 5.0k | 135 | 65.8 | 22.0 | 33.0 | 36.5 | 14.6 | 20.9 | 23.0 | 0.080 | 0.281 | 4.04 | 1.99 | 2.15 | 2.52 | 2.43 | 2.62 |
| MiniCPM-o-2.6-8B | - | 910 | 24.4 | 14.1 | 17.9 | 16.8 | 24.1 | 19.8 | 12.7 | 0.063 | 0.187 | 2.31 | 1.69 | 1.90 | 2.11 | 1.75 | 1.95 |
| InternVL2.5-8B | 11.2k | 232 | 51.5 | 46.0 | 48.6 | 26.6 | 11.7 | 16.3 | 39.7 | 0.102 | 0.279 | 3.65 | 1.96 | 2.37 | 2.17 | 2.17 | 2.48 |
| InternVL3-8B | 11.2k | 422 | 69.2 | 36.8 | 48.0 | 30.1 | 51.9 | 38.1 | 41.4 | 0.122 | 0.262 | 3.79 | 2.66 | 3.00 | 2.95 | 2.82 | 3.04 |
| InternVL3-9B | 11.2k | 304 | **78.0** | 57.3 | 66.0 | 32.3 | 24.7 | 28.0 | 53.1 | 0.153 | 0.306 | 4.04 | 2.85 | 3.25 | 3.04 | 2.78 | 3.19 |
| InternVL3-14B | 11.2k | 381 | 75.4 | 53.2 | 62.3 | 30.3 | 69.2 | 42.1 | 52.5 | 0.148 | 0.290 | 4.01 | 2.93 | 3.35 | 3.23 | 3.15 | 3.33 |
| InternVL2.5-26B | 11.2k | 218 | 67.9 | 34.3 | 45.6 | 28.3 | 7.9 | 12.4 | 32.4 | 0.105 | 0.295 | 3.70 | 1.90 | 2.23 | 2.52 | 2.28 | 2.53 |
| Qwen2.5-VL-32B-Inst | 4.9k | 774 | 64.3 | 38.6 | 47.4 | 28.6 | **76.4** | 41.7 | 39.8 | 0.087 | 0.226 | 4.23 | 3.34 | 3.71 | 3.76 | 3.75 | 3.76 |
| InternVL2.5-38B | 11.2k | 412 | 45.3 | 65.0 | 53.4 | 11.6 | 21.0 | 15.0 | 44.4 | 0.112 | 0.267 | 3.20 | 1.81 | 2.04 | 2.52 | 2.60 | 2.44 |
| InternVL3-38B | 11.2k | 356 | 72.0 | 55.0 | 62.4 | 37.4 | 68.6 | 48.4 | 56.5 | 0.159 | 0.305 | 4.13 | 3.07 | 3.49 | 3.39 | 3.33 | 3.48 |
| Qwen2.5-VL-72B-Inst | 5.0k | 325 | 71.9 | 77.9 | 74.7 | 37.7 | 56.9 | 45.4 | 60.0 | 0.151 | 0.298 | 4.16 | 3.09 | 3.45 | 3.36 | 3.37 | 3.49 |
| InternVL2.5-78B | 11.2k | 255 | 73.7 | 42.7 | 54.1 | 41.4 | 38.7 | 40.0 | 44.2 | 0.108 | 0.318 | 4.21 | 2.89 | 3.07 | 3.13 | 3.09 | 3.28 |
| InternVL3-78B | 11.2k | 312 | 75.2 | 73.4 | 74.3 | 38.5 | 59.8 | 46.8 | 62.5 | **0.167** | 0.314 | 4.11 | 3.08 | 3.52 | 3.36 | 3.25 | 3.46 |
| Llama4-Scout-17Bx16E | 7.8k | 387 | 67.2 | 60.2 | 63.5 | 30.9 | 42.3 | 35.7 | 48.5 | 0.131 | 0.287 | 3.95 | 2.67 | 3.11 | 3.14 | 3.11 | 3.20 |
| Llama4-Mave-17Bx128E | 7.8k | 325 | 72.1 | **80.0** | 75.8 | 36.4 | 44.0 | 39.8 | 61.9 | 0.154 | 0.309 | 4.22 | 3.30 | 3.62 | 3.52 | 3.58 | 3.65 |
| Qwen-VL-Plus | 5.0k | 243 | 61.8 | 22.5 | 33.0 | 27.4 | 26.0 | 26.7 | 27.2 | 0.101 | 0.278 | 3.27 | 2.09 | 2.42 | 2.26 | 2.13 | 2.43 |
| Qwen-VL-Max | 5.0k | 201 | **82.6** | 50.6 | 62.8 | 36.2 | 44.0 | 39.7 | 50.7 | 0.127 | 0.308 | 4.15 | 3.00 | 3.33 | 3.14 | 3.16 | 3.36 |
| Qwen-QVQ-Max | 4.7k | 1152 | 72.2 | 5.9 | 10.9 | 31.4 | 13.4 | 18.8 | 11.6 | 0.106 | 0.290 | 4.53 | 2.41 | 2.80 | 3.65 | 3.50 | 3.38 |
| Gemini-1.5-Pro | 2.8k | 198 | 73.2 | 79.5 | 76.2 | 40.7 | 47.3 | 43.8 | 63.3 | 0.099 | 0.265 | 3.33 | 2.58 | 2.99 | 2.51 | 2.72 | 2.83 |
| Gemini-2.0-Pro | 2.8k | 268 | 74.0 | 86.8 | 79.9 | 38.0 | 64.3 | 47.7 | 65.1 | 0.151 | 0.300 | 3.86 | 2.87 | 3.37 | 3.10 | 3.25 | 3.29 |
| Gemini-2.0-Flash | 2.8k | 222 | 77.2 | 74.8 | 76.0 | 39.8 | 65.0 | 49.3 | 62.9 | 0.132 | 0.291 | 3.66 | 2.73 | 3.15 | 2.85 | 3.02 | 3.08 |
| Gemini-2.0-Flash-Think | 2.8k | 280 | 77.9 | 83.1 | 80.4 | 43.6 | 62.9 | 51.5 | 68.9 | 0.146 | 0.298 | 4.21 | 3.26 | 3.70 | 3.40 | 3.49 | 3.61 |
| Gemini-2.5-Flash | 2.7k | 351 | 78.4 | 82.6 | 80.4 | 36.9 | 73.4 | 49.1 | 64.6 | 0.142 | 0.287 | 4.22 | 3.23 | 3.81 | 3.62 | 4.03 | 3.74 |
| Gemini-2.5-Pro | 2.7k | 429 | 78.5 | **90.4** | 84.0 | 37.9 | 76.1 | 50.6 | 68.1 | 0.144 | 0.291 | 4.35 | 3.50 | 4.03 | 3.83 | 4.03 | 3.95 |
| Claude-3.5-Sonnet | 5.5k | 313 | 72.2 | 87.6 | 79.2 | 37.1 | 74.1 | 49.5 | 65.2 | 0.121 | 0.278 | 4.27 | 3.18 | 3.63 | 3.71 | 3.55 | 3.64 |
| GPT-4o-mini | 6.8k | 356 | 69.2 | 78.6 | 73.6 | 34.4 | 50.2 | 40.8 | 60.4 | 0.147 | 0.297 | 4.53 | 3.11 | 3.61 | 3.55 | 3.33 | 3.63 |
| GPT-4o | 4.6k | 346 | 67.4 | 87.9 | 76.3 | 37.8 | 61.6 | 46.8 | 65.5 | 0.159 | 0.315 | 4.38 | 3.42 | 3.76 | 3.62 | 3.63 | 3.76 |
| GPT-4.1-nano | 9.5k | 303 | 66.3 | 27.6 | 39.0 | 34.7 | 48.9 | 40.6 | 34.9 | 0.134 | 0.303 | 4.21 | 2.74 | 3.06 | 3.21 | 2.93 | 3.23 |
| GPT-4.1-mini | 6.7k | 458 | 68.8 | 90.2 | 78.1 | 34.5 | 74.8 | 47.2 | 65.1 | 0.134 | 0.287 | 4.44 | 3.47 | 3.98 | 3.92 | 3.94 | 3.95 |
| GPT-4.1 | 4.6k | 296 | 81.8 | 87.4 | 84.5 | 45.5 | 67.2 | 54.3 | 72.6 | 0.159 | 0.315 | 4.62 | 3.75 | 4.21 | 4.12 | 4.09 | 4.16 |

Table 8: Main results (**using 15 quotes as context**) for quote selection and multimodal answer generation. The best and second best scores are in **boldface** and underlined. Two most important columns: (i) Overall F$_1$ of both image/text quotes selection, and (ii) Average Scores of fluency, cite quality, text-image coherence, reasoning logic, and factuality for answer generation, are highlighted.

| Method Multimodal (MM) | Pure-Text (PT) | In-token Usage | | | Overall Quote $F_1$ | | | Answer Avg. | | |
|---|---|---|---|---|---|---|---|---|---|---|
| | | MM | PT | Δ% | MM | PT | Δ% | MM | PT | Δ% |
| Use the **same VLM** to process both multimodal and pure-text inputs. | | | | | | | | | | |
| Gemini-1.5-Pro | | 2.8k | 2.8k | +0.0 | 63.3 | 57.8 | -8.7 | 3.03 | 2.83 | -6.6 |
| Gemini-2.0-Pro | | 2.8k | 2.8k | +0.0 | 65.1 | 63.5 | -2.5 | 3.29 | 3.50 | +6.4 |
| Gemini-2.0-Flash | | 2.8k | 2.8k | +0.0 | 62.9 | 56.0 | -11.0 | 3.08 | 3.19 | +3.6 |
| Gemini-2.0-Flash-Think | | 2.8k | 2.8k | +0.0 | 68.9 | 62.2 | -9.7 | 3.61 | 3.51 | -2.8 |
| Gemini-2.5-Pro | | 2.7k | 2.7k | +0.0 | 68.1 | 66.6 | -2.2 | 3.95 | 3.84 | -2.8 |
| Gemini-2.5-Flash | | 2.7k | 2.7k | +0.0 | 64.6 | 61.1 | -5.4 | 3.74 | 3.55 | -5.1 |
| Claude-3.5-Sonnet | | 5.5k | 2.9k | -47.3 | 65.2 | 61.6 | -5.5 | 3.64 | 3.63 | -0.3 |
| GPT-4o-mini | | 6.8k | 2.6k | -61.8 | 60.4 | 59.9 | -0.8 | 3.63 | 3.69 | +1.7 |
| GPT-4o | | 4.6k | 2.6k | -43.5 | 65.6 | 64.1 | -2.3 | 3.76 | 3.73 | -0.8 |
| GPT-4.1-nano | | 9.5k | 2.5k | 73.7 | 34.9 | 45.1 | +29.2 | 3.23 | 3.46 | +7.1 |
| GPT-4.1-mini | | 6.7k | 2.5k | -62.7 | 65.1 | 63.7 | -2.2 | 3.95 | 3.90 | -1.3 |
| GPT-4.1 | | 4.6k | 2.5k | -45.7 | 72.6 | 70.6 | -2.8 | 4.16 | 4.06 | -2.4 |
| Llama4-Scout-17Bx16E | | 7.8k | 2.5k | -67.9 | 48.5 | 52.8 | +8.9 | 3.19 | 3.17 | -0.6 |
| Llama4-Mave-17Bx128E | | 7.8k | 2.5k | -67.9 | 61.9 | 63.0 | +1.8 | 3.65 | 3.57 | -2.2 |
| InternVL3-8B | | 11.2k | 2.7k | -75.9 | 41.4 | 52.7 | +27.3 | 3.04 | 3.19 | +4.9 |
| InternVL3-9B | | 11.2k | 3.0k | -73.2 | 53.1 | 48.1 | -9.4 | 3.19 | 3.34 | +4.7 |
| InternVL3-14B | | 11.2k | 2.7k | -75.9 | 52.5 | 51.4 | -2.1 | 3.33 | 3.50 | +5.1 |
| InternVL3-38B | | 11.2k | 2.7k | -75.9 | 56.5 | 57.3 | +1.4 | 3.46 | 3.62 | +4.6 |
| InternVL3-78B | | 11.2k | 2.7k | -75.9 | 62.5 | 59.3 | -5.1 | 3.65 | 3.56 | -2.5 |
| Use **separate VLM/LLM** to process multimodal and pure-text inputs, respectively. | | | | | | | | | | |
| Qwen-VL-Plus | Qwen-Plus | 5.0k | 2.7k | -46.0 | 27.2 | 59.1 | +117.3 | 2.43 | 3.62 | +49.0 |
| Qwen-VL-Max | Qwen-Max | 5.0k | 2.7k | -46.0 | 50.7 | 61.9 | +22.1 | 3.36 | 3.77 | +12.2 |
| QVQ-Max | QwQ-32B | 4.7k | 2.7k | -42.6 | 25.8 | 52.0 | +101.6 | 2.44 | 3.64 | +49.2 |
| Qwen2.5-VL-7B | Qwen2.5-7B | 5.0k | 2.7k | -46.0 | 23.0 | 48.4 | +110.4 | 2.62 | 3.37 | +28.6 |
| Qwen2.5-VL-32B | Qwen2.5-32B | 4.9k | 2.7k | -44.9 | 39.8 | 63.0 | +58.3 | 3.76 | 3.68 | -2.1 |
| Qwen2.5-VL-72B | Qwen2.5-72B | 5.0k | 2.7k | -46.0 | 60.0 | 62.9 | +4.8 | 3.49 | 3.76 | +7.7 |

Table 9: Using **15 quotes** for multimodal generation. Δ% is calculated by values (PT-MM)/MM and displayed in percentage.

| Method | Tokens | | Quote Selection | | | | | | | Multimodal Answer Quality | | | | | | | |
|---|---|---|---|---|---|---|---|---|---|---|---|---|---|---|---|---|---|
| Metric | In | Out | Image Quotes | | | Text Quotes | | | $F_1$ | Bleu | Rou-geL | Flu-ency | Cite Qlty. | Txt-Im Coher. | Reas. Logic | Fact-uality | Avg |
| | | | Prec | Rec | $F_1$ | Prec | Rec | $F_1$ | | | | | | | | | |
| Qwen3-4B | 2.7k | 1057 | 74.1 | 67.9 | 70.9 | 37.2 | 45.5 | 40.9 | 59.8 | 0.139 | 0.301 | 4.27 | 3.16 | 3.67 | 3.50 | 3.47 | 3.61 |
| - *Disabled* | 2.7k | 271 | 67.5 | 66.6 | 67.1 | 34.9 | 38.8 | 36.8 | 55.5 | 0.147 | 0.306 | 3.91 | 2.78 | 3.08 | 2.94 | 2.90 | 3.12 |
| Qwen3-8B | 2.7k | 992 | 77.9 | 72.9 | 75.3 | 38.7 | 61.0 | 47.3 | 64.0 | 0.140 | 0.303 | 4.15 | 3.13 | 3.57 | 3.40 | 3.32 | 3.51 |
| - *Disabled* | 2.7k | 286 | 72.2 | 71.7 | 72.0 | 31.7 | 48.4 | 38.3 | 58.1 | 0.149 | 0.308 | 4.11 | 2.98 | 3.33 | 3.12 | 3.09 | 3.33 |
| Qwen3-14B | 2.7k | 891 | 77.8 | 69.9 | 73.7 | 39.3 | 58.4 | 47.0 | 62.2 | 0.143 | 0.307 | 4.29 | 3.25 | 3.66 | 3.59 | 3.47 | 3.65 |
| - *Disabled* | 2.7k | 344 | 77.2 | 67.3 | 72.0 | 33.8 | 61.2 | 43.6 | 57.9 | 0.150 | 0.296 | 4.37 | 3.21 | 3.57 | 3.45 | 3.42 | 3.60 |
| Qwen3-30B-A3B | 2.7k | 949 | 78.6 | 72.9 | 75.7 | 40.1 | 64.8 | 49.5 | 64.8 | 0.149 | 0.308 | 4.24 | 3.19 | 3.66 | 3.54 | 3.47 | 3.62 |
| - *Disabled* | 2.7k | 378 | 72.4 | 70.8 | 71.6 | 34.8 | 52.6 | 41.9 | 58.6 | 0.155 | 0.305 | 4.27 | 3.16 | 3.49 | 3.34 | 3.33 | 3.52 |
| Qwen3-4B | 3.6k | 1072 | 68.5 | 64.4 | 66.4 | 36.1 | 46.7 | 40.7 | 58.2 | 0.139 | 0.301 | 4.25 | 3.13 | 3.57 | 3.55 | 3.40 | 3.58 |
| - *Disabled* | 3.6k | 271 | 61.4 | 59.8 | 60.6 | 31.1 | 35.1 | 33.0 | 51.1 | 0.144 | 0.304 | 3.91 | 2.71 | 3.11 | 3.00 | 2.96 | 3.14 |
| Qwen3-8B | 3.6k | 1018 | 71.3 | 67.5 | 69.4 | 34.4 | 60.1 | 43.8 | 59.7 | 0.138 | 0.302 | 4.15 | 3.13 | 3.57 | 3.40 | 3.32 | 3.51 |
| - *Disabled* | 3.6k | 337 | 66.9 | 66.5 | 66.7 | 28.7 | 46.8 | 35.5 | 54.9 | 0.142 | 0.301 | 4.01 | 2.88 | 3.35 | 3.16 | 3.06 | 3.29 |
| Qwen3-14B | 3.6k | 920 | 73.0 | 64.9 | 68.7 | 36.4 | 57.3 | 44.5 | 59.9 | 0.142 | 0.305 | 4.29 | 3.25 | 3.66 | 3.59 | 3.47 | 3.65 |
| - *Disabled* | 3.6k | 352 | 72.0 | 59.9 | 65.4 | 32.2 | 61.0 | 42.1 | 54.5 | 0.147 | 0.296 | 4.31 | 3.10 | 3.56 | 3.49 | 3.38 | 3.57 |
| Qwen3-30B-A3B | 3.6k | 969 | 72.5 | 68.2 | 70.3 | 36.7 | 61.1 | 45.9 | 61.4 | 0.147 | 0.305 | 4.22 | 3.23 | 3.68 | 3.49 | 3.40 | 3.60 |
| - *Disabled* | 3.6k | 401 | 65.0 | 62.5 | 63.7 | 31.2 | 48.3 | 37.9 | 53.6 | 0.151 | 0.303 | 4.25 | 3.08 | 3.51 | 3.44 | 3.35 | 3.52 |
| Qwen-QVQ-Max | 6.8k | 1137 | 63.5 | 6.8 | 12.2 | 34.0 | 13.2 | 19.1 | 12.3 | 0.106 | 0.290 | 4.53 | 2.44 | 2.77 | 3.61 | 3.45 | 3.36 |
| - *Disabled* | 6.8k | 1129 | 57.6 | 10.6 | 17.9 | 25.3 | 45.4 | 32.4 | 23.6 | 0.064 | 0.180 | 3.42 | 2.95 | 3.23 | 3.01 | 3.31 | 3.18 |

*15 Quotes* applies to the first eight data rows; *20 Quotes* applies to the remaining rows.

Table 10: Thinking vs Non-thinking: full results on model performance by enabling and disabling thinking before final multimodal generation. The rows marked with "- *Disabled*" refer to disabling thinking mode.

| | Method | | Out-token Usage | | | Overall Quote $F_1$ | | | Answer Avg. | | |
|---|---|---|---|---|---|---|---|---|---|---|---|
| | Yes-Think | No-Think | Yes | No | Δ% | Yes | No | Δ% | Yes | No | Δ% |
| | Use the **same model** to generate thinking and non-thinking outputs. | | | | | | | | | | |
| *15 Quotes* | Qwen3-4B | | 1057 | 271 | -74.4 | 59.8 | 55.5 | -7.2 | 3.61 | 3.12 | -13.6 |
| | Qwen3-8B | | 992 | 286 | -71.2 | 64.0 | 58.1 | -9.2 | 3.51 | 3.33 | -5.1 |
| | Qwen3-14B | | 891 | 344 | -61.4 | 62.2 | 57.9 | -6.9 | 3.65 | 3.60 | -1.4 |
| | Qwen3-30B-A3B | | 949 | 378 | -60.2 | 64.8 | 58.6 | -9.6 | 3.62 | 3.52 | -2.8 |
| *20 Quotes* | Qwen3-4B | | 1072 | 271 | -74.7 | 58.2 | 51.1 | -12.2 | 3.58 | 3.14 | -12.3 |
| | Qwen3-8B | | 1018 | 337 | -66.9 | 59.7 | 54.9 | -8.0 | 3.51 | 3.29 | -6.3 |
| | Qwen3-14B | | 920 | 352 | -61.7 | 59.9 | 54.5 | -9.0 | 3.65 | 3.57 | -2.2 |
| | Qwen3-30B-A3B | | 969 | 401 | -58.6 | 61.4 | 53.6 | -12.7 | 3.60 | 3.52 | -2.2 |
| | Qwen-QVQ-Max | | 1137 | 1129 | -0.7 | 12.3 | 23.6 | +91.9 | 3.36 | 3.18 | -5.4 |
| | Use **separate models** to generate thinking and non-thinking outputs, respectively. | | | | | | | | | | |
| *15 Quotes* | Deepseek-R1 | Deepseek-V3 | 953 | 239 | -74.9 | 62.1 | 64.6 | +4.0 | 3.50 | 3.75 | +7.1 |
| | R1-Distill-Qwen-32B | Qwen2.5-32B | 737 | 316 | -57.1 | 54.2 | 63.0 | +16.2 | 3.57 | 3.68 | +3.1 |
| | R1-Distill-Llama-70B | Llama3-70B | 685 | 434 | -36.6 | 52.8 | 58.5 | +10.8 | 3.17 | 3.14 | -0.9 |
| | QwQ-Plus | Qwen-Plus | 1369 | 306 | -77.6 | 61.9 | 59.1 | -4.5 | 3.77 | 3.62 | -4.0 |
| | QVQ-Max | Qwen-VL-Max | 1152 | 201 | -82.6 | 11.6 | 50.7 | +337.1 | 3.36 | 2.43 | -27.7 |
| | GPT-o3-mini | GPT-4o-mini | 618 | 392 | -36.6 | 59.3 | 59.9 | +1.0 | 3.11 | 3.69 | +18.6 |
| | Qwen3-4B | Qwen2.5-3B | 1057 | 422 | -60.1 | 59.8 | 29.7 | -50.3 | 3.61 | 2.98 | -17.5 |
| | Qwen3-8B | Qwen2.5-7B | 992 | 304 | -69.4 | 64.0 | 48.4 | -24.4 | 3.51 | 3.37 | -4.0 |
| | Qwen3-14B | Qwen2.5-14B | 891 | 356 | -60.0 | 62.2 | 59.6 | -4.2 | 3.65 | 3.50 | -4.1 |
| | Qwen3-32B | Qwen2.5-32B | 884 | 316 | -64.3 | 56.5 | 63.0 | +11.5 | 3.63 | 3.68 | +1.4 |
| | Qwen3-235B-A22B | Qwen2.5-72B | 1068 | 367 | -65.6 | 62.9 | 62.9 | +0.0 | 3.75 | 3.76 | +0.3 |
| *20 Quotes* | Deepseek-R1 | Deepseek-V3 | 930 | 234 | -74.8 | 59.4 | 61.1 | +2.9 | 3.48 | 3.74 | +7.5 |
| | R1-Distill-Qwen-32B | Qwen2.5-32B | 731 | 320 | -56.2 | 58.9 | 44.8 | -23.9 | 3.34 | 3.63 | +8.7 |
| | R1-Distill-Llama-70B | Llama3-70B | 680 | 430 | -36.8 | 55.6 | 51.0 | -8.3 | 3.50 | 3.24 | -7.4 |
| | QwQ-Plus | Qwen-Plus | 1266 | 316 | -75.0 | 59.6 | 55.4 | -7.0 | 3.63 | 3.63 | +0.0 |
| | QVQ-Max | Qwen-VL-Max | 1137 | 206 | -81.9 | 12.3 | 46.8 | +280.5 | 3.36 | 3.35 | -0.3 |
| | GPT-o3-mini | GPT-4o-mini | 623 | 394 | -36.8 | 57.0 | 56.6 | -0.7 | 3.10 | 3.70 | +19.4 |
| | Qwen3-4B | Qwen2.5-3B | 1072 | 415 | -61.3 | 58.2 | 25.0 | -57.0 | 3.58 | 2.94 | -17.9 |
| | Qwen3-8B | Qwen2.5-7B | 1018 | 302 | -70.3 | 59.7 | 45.8 | -23.3 | 3.51 | 3.34 | -4.8 |
| | Qwen3-14B | Qwen2.5-14B | 920 | 362 | -60.7 | 59.9 | 54.7 | -8.7 | 3.65 | 3.49 | -4.4 |
| | Qwen3-32B | Qwen2.5-32B | 917 | 320 | -65.1 | 54.5 | 58.9 | +8.1 | 3.61 | 3.63 | +0.6 |
| | Qwen3-235B-A22B | Qwen2.5-72B | 1052 | 380 | -63.9 | 59.5 | 59.1 | -0.7 | 3.77 | 3.75 | -0.5 |

Table 11: Comparative results between scores achieved via thinking and non-thinking based generation. Δ% is calculated by values (No-Yes)/No and displayed in percentage.

| Method | Tokens | | Quote Selection | | | | | | | Multimodal Answer Quality | | | | | | | |
|---|---|---|---|---|---|---|---|---|---|---|---|---|---|---|---|---|---|
| | | | Image Quotes | | | Text Quotes | | | | | | | | | | | |
| Metric | In | Out | Prec | Rec | F$_1$ | Prec | Rec | F$_1$ | F$_1$ | Bleu | Rou-geL | Flu-ency | Cite Qlty. | Txt-Im Coher. | Reas. Logic | Fact-uality | Avg |
| **15 Quotes** | | | | | | | | | | | | | | | | | |
| Qwen2.5-7B-Inst | 2.7k | 304 | 72.3 | 51.0 | 59.8 | 36.6 | 28.8 | 32.3 | 48.4 | 0.160 | 0.311 | 4.25 | 2.99 | 3.31 | 3.25 | 3.06 | 3.37 |
| *- Using OCR-text* | 2.4k | 304 | 56.3 | 44.3 | 49.6 | 32.3 | 28.9 | 30.5 | 40.4 | 0.136 | 0.288 | 4.08 | 2.86 | 3.02 | 3.11 | 2.67 | 3.15 |
| Llama3.1-8B-Inst | 2.6k | 423 | 62.2 | 60.0 | 61.1 | 27.6 | 42.9 | 33.6 | 46.0 | 0.116 | 0.257 | 4.26 | 2.95 | 3.22 | 3.16 | 3.07 | 3.33 |
| *- Using OCR-text* | 2.2k | 430 | 50.9 | 54.0 | 52.4 | 25.3 | 46.4 | 32.7 | 40.4 | 0.098 | 0.238 | 4.11 | 2.63 | 3.20 | 3.07 | 2.93 | 3.19 |
| Llama3.3-70B-Inst | 2.7k | 434 | 59.8 | **89.8** | 71.8 | 32.2 | 70.4 | 44.2 | 58.5 | 0.120 | 0.263 | 3.73 | 2.72 | 3.10 | 2.98 | 3.18 | 3.14 |
| *- Using OCR-text* | 2.2k | 408 | 53.9 | 79.0 | 64.1 | 30.4 | **72.3** | 42.8 | 53.6 | 0.114 | 0.258 | 3.64 | 2.75 | 3.01 | 2.87 | 3.13 | 3.08 |
| Qwen2.5-72B-Inst | 2.7k | 367 | **80.7** | 67.1 | 73.3 | **42.1** | 50.9 | 46.1 | 62.9 | **0.175** | 0.326 | 4.50 | 3.39 | 3.73 | **3.65** | 3.53 | 3.76 |
| *- Using OCR-text* | 2.4k | 358 | 75.1 | 58.3 | 65.7 | 37.2 | 58.6 | 45.5 | 57.1 | 0.152 | 0.302 | 4.33 | 3.24 | 3.11 | 3.58 | 3.49 | 3.55 |
| Qwen-Max | 2.7k | 406 | 76.9 | 72.0 | 74.4 | 41.9 | 53.7 | 47.1 | 61.9 | 0.168 | 0.319 | 4.42 | **3.46** | **3.74** | 3.64 | 3.59 | **3.77** |
| *- Using OCR-text* | 2.4k | 380 | 71.1 | 61.1 | 65.7 | 40.2 | 58.9 | 47.8 | 57.0 | 0.150 | 0.299 | 4.29 | 3.37 | 3.55 | 3.49 | 3.48 | 3.63 |
| Deepseek-V3 | 2.7k | 239 | 76.0 | 76.9 | 76.5 | 41.5 | 63.8 | **50.3** | 64.6 | 0.173 | **0.341** | **4.54** | 3.33 | 3.74 | 3.63 | 3.54 | 3.75 |
| *- Using OCR-text* | 2.3k | 228 | 70.9 | 69.6 | 70.2 | 38.6 | 66.3 | 48.8 | 59.5 | 0.150 | 0.316 | 4.49 | 3.23 | 3.70 | 3.56 | 3.44 | 3.68 |
| Gemini-2.0-Pro | 2.8k | 307 | 75.7 | 79.2 | **77.4** | 38.5 | 64.4 | 48.2 | 63.5 | 0.161 | 0.302 | 4.13 | 3.05 | 3.56 | 3.31 | 3.45 | 3.50 |
| *- Using OCR-text* | 2.4k | 270 | 71.5 | 78.6 | 74.9 | 38.3 | 63.9 | 47.9 | 62.0 | 0.146 | 0.292 | 4.08 | 2.85 | 3.44 | 3.33 | 3.37 | 3.41 |
| Gemini-2.0-Flash-TK | 2.8k | 270 | 76.5 | 73.3 | 74.9 | 38.8 | 62.3 | 47.8 | 62.2 | 0.132 | 0.270 | 4.13 | 3.07 | 3.63 | 3.30 | 3.43 | 3.51 |
| *- Using OCR-text* | 2.4k | 252 | 73.4 | 72.5 | 73.0 | 39.7 | 62.9 | 48.7 | 61.4 | 0.124 | 0.266 | 4.10 | 2.95 | 3.57 | 3.22 | 3.36 | 3.46 |
| GPT-4o | 2.6k | 386 | 70.9 | 80.3 | 75.3 | 40.3 | 61.4 | 48.7 | 64.1 | 0.156 | 0.307 | 4.33 | 3.41 | 3.67 | 3.60 | **3.64** | 3.73 |
| *- Using OCR-text* | 2.2k | 423 | 63.8 | 76.1 | 69.4 | 35.0 | 69.2 | 46.5 | 59.4 | 0.129 | 0.274 | 4.15 | 3.37 | 3.55 | 3.58 | 3.60 | 3.65 |
| **20 Quotes** | | | | | | | | | | | | | | | | | |
| Qwen2.5-7B-Inst | 3.6k | 302 | 66.5 | 45.5 | 54.0 | 36.2 | 28.2 | 31.7 | 45.8 | 0.159 | 0.313 | 4.27 | 2.93 | 3.21 | 3.22 | 3.07 | 3.34 |
| *- Using OCR-text* | 3.1k | 302 | 50.0 | 38.5 | 43.5 | 30.5 | 26.6 | 28.4 | 37.1 | 0.134 | 0.287 | 4.16 | 2.78 | 2.94 | 3.08 | 2.77 | 3.15 |
| Llama3.1-8B-Inst | 3.4k | 435 | 54.1 | 51.8 | 52.9 | 24.1 | 38.1 | 29.5 | 41.0 | 0.112 | 0.254 | 4.17 | 2.88 | 3.15 | 3.08 | 2.99 | 3.25 |
| *- Using OCR-text* | 2.8k | 445 | 45.0 | 46.5 | 45.7 | 22.9 | 41.2 | 29.5 | 36.0 | 0.093 | 0.235 | 4.09 | 2.67 | 3.08 | 3.10 | 2.88 | 3.16 |
| Llama3.3-70B-Inst | 3.4k | 430 | 54.3 | **82.5** | 65.5 | 30.6 | 64.3 | 41.5 | 55.6 | 0.120 | 0.264 | 3.93 | 2.72 | 3.17 | 3.11 | 3.26 | 3.24 |
| *- Using OCR-text* | 2.8k | 404 | 51.1 | 74.3 | 60.6 | 29.1 | **68.9** | 40.9 | 51.7 | 0.113 | 0.257 | 3.77 | 2.80 | 3.03 | 2.93 | 3.10 | 3.13 |
| Qwen2.5-72B-Inst | 3.6k | 380 | **76.5** | 62.1 | 68.5 | **38.8** | 49.2 | 43.4 | 59.1 | 0.173 | **0.324** | 4.48 | **3.41** | **3.71** | **3.64** | 3.53 | **3.75** |
| *- Using OCR-text* | 3.1k | 364 | 68.2 | 53.0 | 59.7 | 36.0 | 57.7 | 44.3 | 53.3 | 0.151 | 0.300 | 4.27 | 3.18 | 3.06 | 3.60 | 3.41 | 3.50 |
| Qwen-Max | 3.6k | 426 | 71.7 | 66.9 | 69.3 | **39.7** | 51.5 | 44.8 | 58.9 | 0.165 | 0.315 | 4.42 | **3.47** | 3.71 | 3.64 | 3.59 | **3.77** |
| *- Using OCR-text* | 3.1k | 383 | 65.6 | 55.2 | 59.9 | 36.8 | 55.3 | 44.2 | 52.5 | 0.148 | 0.298 | 4.25 | 3.40 | 3.44 | 3.55 | 3.50 | 3.62 |
| Deepseek-V3 | 3.4k | 234 | 70.8 | 73.4 | 72.1 | 37.3 | 59.8 | 45.9 | 61.1 | 0.171 | **0.338** | **4.57** | 3.31 | **3.74** | 3.62 | 3.47 | 3.74 |
| *- Using OCR-text* | 2.9k | 228 | 65.6 | 66.0 | 65.8 | 35.8 | 63.4 | 45.7 | 56.9 | 0.149 | 0.318 | 4.40 | 3.17 | 3.55 | 3.42 | 3.40 | 3.59 |
| Gemini-2.0-Pro | 3.6k | 307 | 71.7 | 81.4 | **76.3** | 36.7 | 61.3 | 45.9 | **62.8** | 0.164 | 0.308 | 4.13 | 3.08 | 3.56 | 3.34 | 3.46 | 3.51 |
| *- Using OCR-text* | 3.1k | 276 | 66.9 | 75.3 | 70.9 | 36.5 | 61.4 | 45.8 | 59.6 | 0.144 | 0.291 | 3.99 | 2.75 | 3.28 | 3.26 | 3.30 | 3.32 |
| Gemini-2.0-Flash-TK | 3.6k | 275 | 72.0 | 73.6 | 72.8 | 37.4 | 60.5 | **46.2** | 61.0 | 0.133 | 0.272 | 4.14 | 3.04 | 3.54 | 3.27 | 3.35 | 3.47 |
| *- Using OCR-text* | 3.1k | 256 | 67.8 | 68.8 | 68.3 | 36.4 | 58.8 | 44.9 | 57.7 | 0.123 | 0.265 | 4.05 | 3.00 | 3.48 | 3.17 | 3.33 | 3.41 |
| GPT-4o | 3.4k | 353 | 66.9 | 67.1 | 67.0 | 37.0 | 57.2 | 44.9 | 57.2 | 0.160 | 0.313 | 4.29 | 3.37 | 3.65 | 3.56 | 3.59 | 3.69 |
| *- Using OCR-text* | 2.8k | 419 | 57.1 | 72.3 | 63.8 | 32.7 | 65.5 | 43.6 | 56.8 | 0.129 | 0.276 | 4.10 | 3.38 | 3.23 | 3.56 | **3.67** | 3.59 |

Table 12: Quotes as Text: full results on model performance by using OCR-text and VLM-text. The rows marked with "*- Using OCR-text*" refer to using OCR-text to represent image quotes, and otherwise VLM-text.

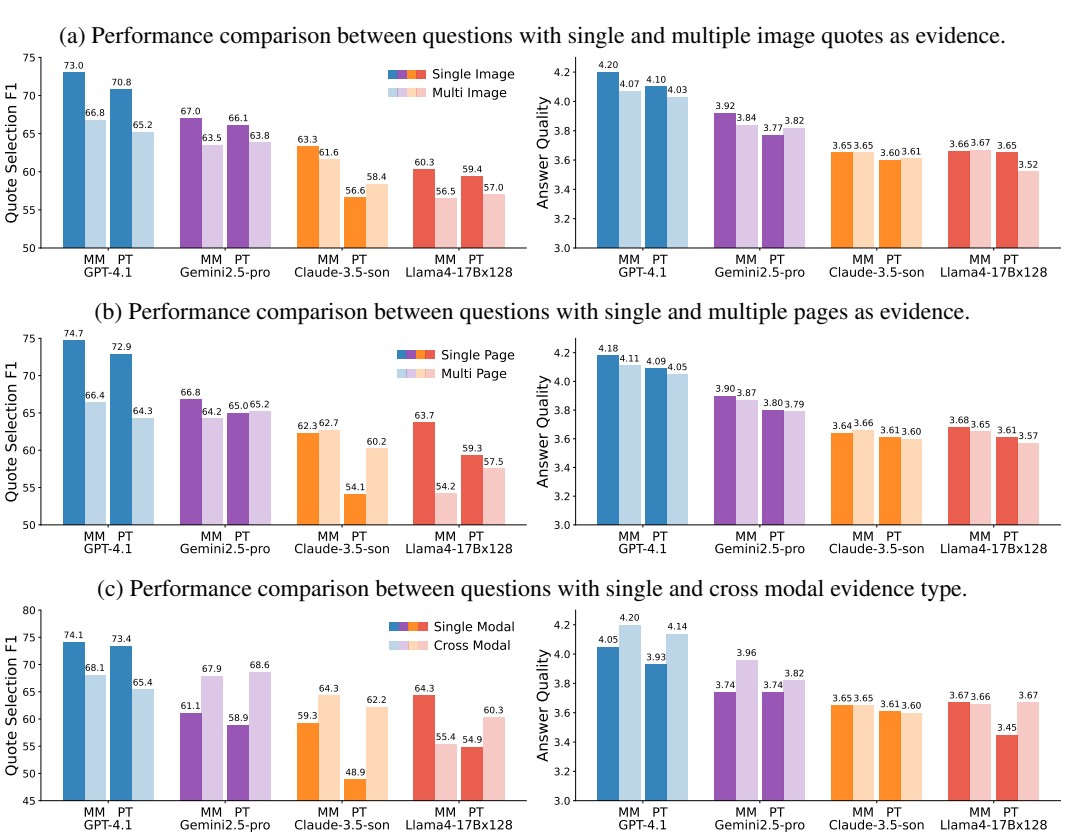

Figure 11: Fine-grained results of 4 VLMs using both 20 multimodal (MM) and pure-text (PT) quotes. The breakdown is according to questions consisting of: (a) single or multiple image quotes as evidence, (b) single or multiple pages as evidence, and (c) single or cross modal evidence type.

| Model | Parameters Total | Active | Image Support | Model Checkpoint or Identifier |
|---|---|---|---|---|
| **Open-source Models** | | | | |
| Qwen2.5-3B-Instruct [65] | 3B | - | ✗ | Qwen/Qwen2.5-3B-Instruct |
| Qwen2.5-7B-Instruct [65] | 7B | - | ✗ | Qwen/Qwen2.5-7B-Instruct |
| Qwen2.5-14B-Instruct [65] | 14B | - | ✗ | Qwen/Qwen2.5-14B-Instruct |
| Qwen2.5-32B-Instruct [65] | 32B | - | ✗ | Qwen/Qwen2.5-32B-Instruct |
| Qwen2.5-72B-Instruct [65] | 72B | - | ✗ | Qwen/Qwen2.5-72B-Instruct |
| Qwen2.5-VL-7B-Instruct [3] | 7B | - | ✓ | Qwen/Qwen2.5-VL-7B-Instruct |
| Qwen2.5-VL-32B-Instruct [3] | 32B | - | ✓ | Qwen/Qwen2.5-VL-32B-Instruct |
| Qwen2.5-VL-72B-Instruct [3] | 72B | - | ✓ | Qwen/Qwen2.5-VL-72B-Instruct |
| Qwen-QVQ-72B-Preview [64] | 72B | - | ✓ | Qwen/QVQ-72B-Preview |
| Qwen3-4B [62] | 4B | - | ✗ | Qwen/Qwen3-4B |
| Qwen3-8B [62] | 8B | - | ✗ | Qwen/Qwen3-8B |
| Qwen3-14B [62] | 14B | - | ✗ | Qwen/Qwen3-14B |
| Qwen3-32B [62] | 32B | - | ✗ | Qwen/Qwen3-32B |
| Qwen3-30B-A3B [62] | 30B | 3B | ✗ | Qwen/Qwen3-30B-A3B |
| Qwen3-235B-A22B [62] | 235B | 22B | ✗ | Qwen/Qwen3-235B-A22B |
| Llama3.2-3B-Instruct [29] | 3B | - | ✗ | meta-llama/Llama-3.2-3B-Instruct |
| Llama3.1-8B-Instruct [29] | 8B | - | ✗ | meta-llama/Llama-3.1-8B-Instruct |
| Llama3.3-70B-Instruct [29] | 70B | - | ✗ | meta-llama/Llama-3.3-70B-Instruct |
| Llama-4-Scout-17B-16E-Instruct [51] | 109B | 17B | ✓ | meta-llama/Llama-4-Scout-17B-16E |
| Llama-4-Maverick-17B-128E-Instruct [51] | 400B | 17B | ✓ | meta-llama/Llama-4-Maverick-17B-128E-Instruct |
| Mistral-7B-Instruct [34] | 7B | - | ✗ | mistralai/Mistral-7B-Instruct-v0.2 |
| Mistral-Small-24B-Instruct [53] | 24B | - | ✗ | mistralai/Mistral-Small-24B-Instruct-2501 |
| Mixtral-8x7B-Instruct [35] | 46.7B | 12.9B | ✗ | mistralai/Mixtral-8x7B-Instruct-v0.1 |
| Deepseek-V3 [12] | 671B | 37B | ✗ | deepseek-ai/DeepSeek-V3 |
| Deepseek-R1 [13] | 671B | 37B | ✗ | deepseek-ai/DeepSeek-R1 |
| DeepSeek-R1-Distill-Qwen-32B [13] | 32B | - | ✗ | deepseek-ai/DeepSeek-R1-Distill-Qwen-32B |
| DeepSeek-R1-Distill-Llama-70B [13] | 70B | - | ✗ | deepseek-ai/DeepSeek-R1-Distill-Llama-70B |
| Janus-Pro-7B [6] | 7B | - | ✓ | deepseek-ai/Janus-Pro-7B |
| MiniCPM-o-2.6-8B [85] | 8B | - | ✓ | openbmb/MiniCPM-o-2_6 |
| InternVL2.5-8B [7] | 8B | - | ✓ | OpenGVLab/InternVL2_5-8B |
| InternVL2.5-26B [7] | 26B | - | ✓ | OpenGVLab/InternVL2_5-26B |
| InternVL2.5-38B [7] | 38B | - | ✓ | OpenGVLab/InternVL2_5-38B |
| InternVL2.5-78B [7] | 78B | - | ✓ | OpenGVLab/InternVL2_5-78B |
| InternVL3-8B [90] | 8B | - | ✓ | OpenGVLab/InternVL3-8B |
| InternVL3-9B [90] | 9B | - | ✓ | OpenGVLab/InternVL3-9B |
| InternVL3-14B [90] | 14B | - | ✓ | OpenGVLab/InternVL3-14B |
| InternVL3-38B [90] | 38B | - | ✓ | OpenGVLab/InternVL3-38B |
| InternVL3-78B [90] | 78B | - | ✓ | OpenGVLab/InternVL3-78B |
| **Proprietary Models** | | | | |
| Qwen-Plus [63] | - | - | ✗ | qwen-plus-2025-01-25 |
| Qwen-Max [63] | - | - | ✗ | qwen-max-2025-01-25 |
| Qwen-VL-Plus [60] | - | - | ✓ | qwen-vl-plus-2025-01-25 |
| Qwen-VL-Max [60] | - | - | ✓ | qwen-vl-max-2025-01-25 |
| Qwen-QVQ-Max [64] | - | - | ✓ | qvq-max-2025-03-25 |
| Qwen-QwQ-Plus [61] | - | - | ✓ | qwq-plus-2025-03-05 |
| Gemini-1.5-Pro [28] | - | - | ✓ | gemini-1.5-pro |
| Gemini-2.0-Pro [27] | - | - | ✓ | gemini-2.0-pro-exp-02-05 |
| Gemini-2.0-Flash [25] | - | - | ✓ | gemini-2.0-flash-exp |
| Gemini-2.0-Flash-Thinking [26] | - | - | ✓ | gemini-2.0-flash-thinking-exp |
| Gemini-2.5-Pro [24] | - | - | ✓ | gemini-2.5-pro-preview-03-2 |
| Gemini-2.5-Flash [23] | - | - | ✓ | gemini-2.5-flash-preview-04-17 |
| Claude-3.5-Sonnet [2] | - | - | ✓ | claude-3-5-sonnet-20241022 |
| Grok-3-mini-beta [80] | - | - | ✗ | grok-3-beta-mini |
| Grok-3-beta [80] | - | - | ✗ | grok-3-beta |
| GPT-4-turbo [54] | - | - | ✗ | gpt-4-turbo-2024-04-09 |
| GPT-4o [55] | - | - | ✓ | gpt-4o-2024-08-06 |
| GPT-4o-mini [56] | - | - | ✓ | gpt-4o-mini-2024-07-18 |
| GPT-o3-mini [57] | - | - | ✗ | o3-mini-2025-01-31 |
| GPT-4.1 [58] | - | - | ✓ | gpt-4.1-2025-04-14 |
| GPT-4.1-mini [58] | - | - | ✓ | gpt-4.1-mini-2025-04-14 |
| GPT-4.1-nano [58] | - | - | ✓ | gpt-4.1-nano-2025-04-14 |

Table 13: Implementation details for Open-source and Proprietary Models

# C  Implementation Details

In this Appendix section, we details the implementation details of VLM/LLM inference (Appendix C.1), LLM finetuning (Appendix C.2), Retrievers (Appendix C.3). All related codes and datasets for training and evaluation can be access from `https://github.com/MMDocRAG/MMDocRAG`.

## C.1 Implementation Details of Large Models Inference

We evaluate 60 state-of-the-art large models, including 33 vision-language models (VLMs) that process interleaved text and image inputs, and 27 language models (LLMs) that handle text-only inputs. Specifically, our study covers 38 open-source models: Qwen-2.5 models [3, 65, 79], Qwen-3 models [62], LLama-3 models [29], Llama-4 models [51], DeepSeek models [6, 12, 13], Mistral models [34, 35, 53], InternVL-2.5 models [7], InternVL-3 models [90], and MiniCPM-o-2.6-8B [85]. Additionally, we include 22 proprietary models: Qwen models [60, 63, 64], GPT models [54–58], Gemini models [23–28], Grok3 models [80], and Claude-3.5-Sonnet [2]. We summarize the pre-trained checkpoints available on HuggingFace [3] and official model identifiers of proprietary models in Table 13. Note that Llama-3.2-11B-Vision and Llama-3.2-90B-Vision [52], which do not support taking multiple images in their input sequence, are excluded from our experiments.

**Deployment of Open-source Large Models.**   Open-source models are deployed using SWIFT[4], a scalable and lightweight fine-tuning framework. Alternatively, many open-source models can be accessed via API service providers such as Alibaba Cloud (Bailian)[5] and Deepinfra Platform[6].

**Multimodal inputs for VLM.**   For VLMs, we follow the inference setting described in Section 4.2. Multimodal quotes are provided as interleaved text and image inputs for both quote selection and multimodal answer generation. Prompts are structured using the template illustrated in Figure 22, with all images base64-encoded for input.

**Pure text inputs for LLM and VLM.**   For both LLMs and VLMs in pure-text settings, multimodal quotes are converted to textual representations following the process in Section 2.1. This includes using either OCR-derived text or VLM-generated text for images. The prompt template in Figure 23 is applied to consolidate all quotes and questions.

## C.2 Implementation Details of LLM Finetuning

As described in Section 4.2, we finetune (i) five Qwen2.5 LLMs including: Qwen2.5-3B-Instruct, Qwen2.5-7B-Instruct, Qwen2.5-14B-Instruct, Qwen2.5-32B-Instruct, and Qwen2.5-72B-Instruct, (ii) two Qwen2.5-VL VLMs namely: Qwen2.5-VL-3B-Instruct and Qwen2.5-VL-7B-Instruct, and (iii) two InternVL-3 VLMs namely: InternVL-3-8B and InternVL-3-9B.

**Data Preparation.**   Training is conducted on the `MMDocRAG` development set, comprising 2,055 questions, each annotated with both 15 and 20 quotes. As citation indices[7] differ between settings, corresponding multimodal answers also vary. Combining both settings yields 4,110 training instances, each in the format `<system instruction, user message, response>`. System instructions and user messages are generated from the prompt template in Figure 23, populated with relevant questions and multimodal quotes. The response is the corresponding multimodal answer from `MMDocRAG`.

**Supervised Finetuning.**   Supervised fine-tuning is performed with the SWIFT framework, utilizing memory-efficient methods such as LoRA [31], FlashAttention [11], and DeepSpeed [66]. We set the LoRA rank to 16 and alpha to 32. For finetuning LLMs, we set the maximum sequence length to 8k, given the average input length of 3.6k tokens (see Table 3). For finetuning VLMs, we set the maxium sequence length to 32k instead, given that images need more tokens for accurate representation. Training is performed for one epoch, using gradient accumulation to update LoRA weights every 8 training steps.

**Inference of Finetuned Model.**   Inference with finetuned models is based on the pure-text input setting, as noted in Appendix C.1. Multimodal quotes are converted to text, and the same prompt structure is used for quote selection and multimodal answer generation.

---

[3] https://huggingface.co/
[4] https://github.com/modelscope/ms-swift
[5] https://www.alibabacloud.com/
[6] https://deepinfra.com/
[7] We shuffle the indices of all quotes, and make sure the indices of gold quotes are evenly distributed.

| | Model | Dimension | Base Model | HuggingFace Checkpoint |
|---|---|---|---|---|
| Text | DPR [36] | 768 | BERT-base [14] | facebook/dpr-ctx_encoder-multiset-base
facebook/dpr-question_encoder-multiset-base |
| | ColBERT [37] | $N_{\mathrm{tok}} \times 768$ | BERT-base [14] | colbert-ir/colbertv2.0 |
| | Contriever [33] | 768 | BERT-base [14] | facebook/contriever-msmarco |
| | E5 [78] | 1,024 | BERT-large [14] | intfloat/e5-large-v2 |
| | BGE [83] | 1,024 | RetroMAE [82] | BAAI/bge-large-en-v1.5 |
| | GTE [41] | 1,024 | BERT-large [14] | thenlper/gte-large |
| Visual | $\mathrm{DSE_{wiki-ss}}$ [46] | 3,072 | Phi-3-Vision [1] | Tevatron/dse-phi3-v1.0 |
| | $\mathrm{DSE_{docmatix}}$ [46] | 3,072 | Phi-3-Vision [1] | Tevatron/dse-phi3-docmatix-v2 |
| | ColPali [21] | $N_{\mathrm{tok}} \times 1{,}024$ | PaliGemma [4] | vidore/colpali |
| | ColQwen [21] | $N_{\mathrm{tok}} \times 1{,}024$ | Qwen2-VL [60] | vidore/colqwen2-v0.1 |

Table 14: Implementation details for Text and Vision Retrieval Models

## C.3 Implementation Details of Retriever

**Text Retrieval: Introduction.** Text retrieval methods are typically categorized into sparse and dense retrieval. Sparse retrievers, such as TF-IDF [69] and BM25 [68], compute relevance based on word frequency statistics, with BM25 adding nonlinear frequency saturation and length normalization. Dense retrievers represent content as vectors: DPR [36] is a pioneering work for QA tasks; ColBERT [37] enables efficient late interaction for fine-grained question-document matching; Contriever [33] employs contrastive learning to enhance dense representations; E5 [78] and BGE [83] introduce improved training and data strategies; and GTE [41] incorporates graph-based methods for further enhancement. Despite recent progress, most text retrievers focus on textual content [40] and overlook valuable visual information that may be embedded in documents.

**Text Retriever: Implementation Details.** In our experiments (section 4), we implement 6 dense text retrievers: DPR [36], ColBERT [37], Contriever [33], E5 [78], BGE [83], and GTE [41]. All models use the BERT WordPiece tokenizer and a maximum sequence length of 512 tokens [14]. We utilize publicly available checkpoints from HuggingFace (see Table 14 for details) and the sentence-transformers library[8] for deploying E5, BGE, and GTE.

**Visual Retrieval: Introduction.** Vision Language Models (VLMs) [1, 4, 7, 65] have enabled the development of visual-driven document retrievers. Recent models such as ColPali [21] and DSE [46] leverage PaliGemma [4] and Phi3-Vision [1] to directly encode document page screenshots for multimodal retrieval. ColPali utilizes fine-grained, token-level question-document interactions similar to ColBERT, while DSE adopts a global dense embedding approach as in DPR. Visual retrievers directly exploit visual content, enabling multimodal retrieval systems to handle non-textual information natively. However, they face challenges with document pages of high resolution due to increased computational and memory requirements for visual token embedding.

**Visual Retriever: Implementation Details.** We implement four visual retrievers: $\mathrm{DSE_{wiki-ss}}$ [46], $\mathrm{DSE_{docmatix}}$ [46], ColPali [21], and ColQwen [21]. These models use image tokenizers to convert image quotes into $14 \times 14$ pixel patches, each corresponding to a visual token. We employ pre-trained checkpoints from HuggingFace, with configuration details listed in Table 14.

**Hybrid Retrieval.** For hybrid text-image retrieval, we pair top-performing text retrievers (BGE and ColBERT) with visual retrievers (ColPali and ColQwen), resulting in four combinations: ColP+ColB, ColP+BGE, ColQ+ColB, and ColQ+BGE. For each combination, we retrieve the top 10, 15, or 20 quotes, with fixed splits (*e.g.,* top 10: 3 images and 7 texts; top 15: 5 images and 10 texts; top 20: 8 images and 12 texts). This approach enables integrated retrieval from both textual and visual content.

---

[8] https://www.sbert.net/

# D Annotation Examples

In this section, we present 6 annotation examples that illustrate typical multimodal reasoning and integration patterns, which help clarify the construction and use of `MMDocRAG`. Each annotation includes the following components: question, short answer, a set of noisy image and text quotes, gold quotes, and the final multimodal answer. These examples frequently require reasoning across multiple pages and modalities. The image quotes encompass diverse formats such as figures, charts, tables, and infographics, highlighting the complexity and richness of the multimodal reasoning tasks.

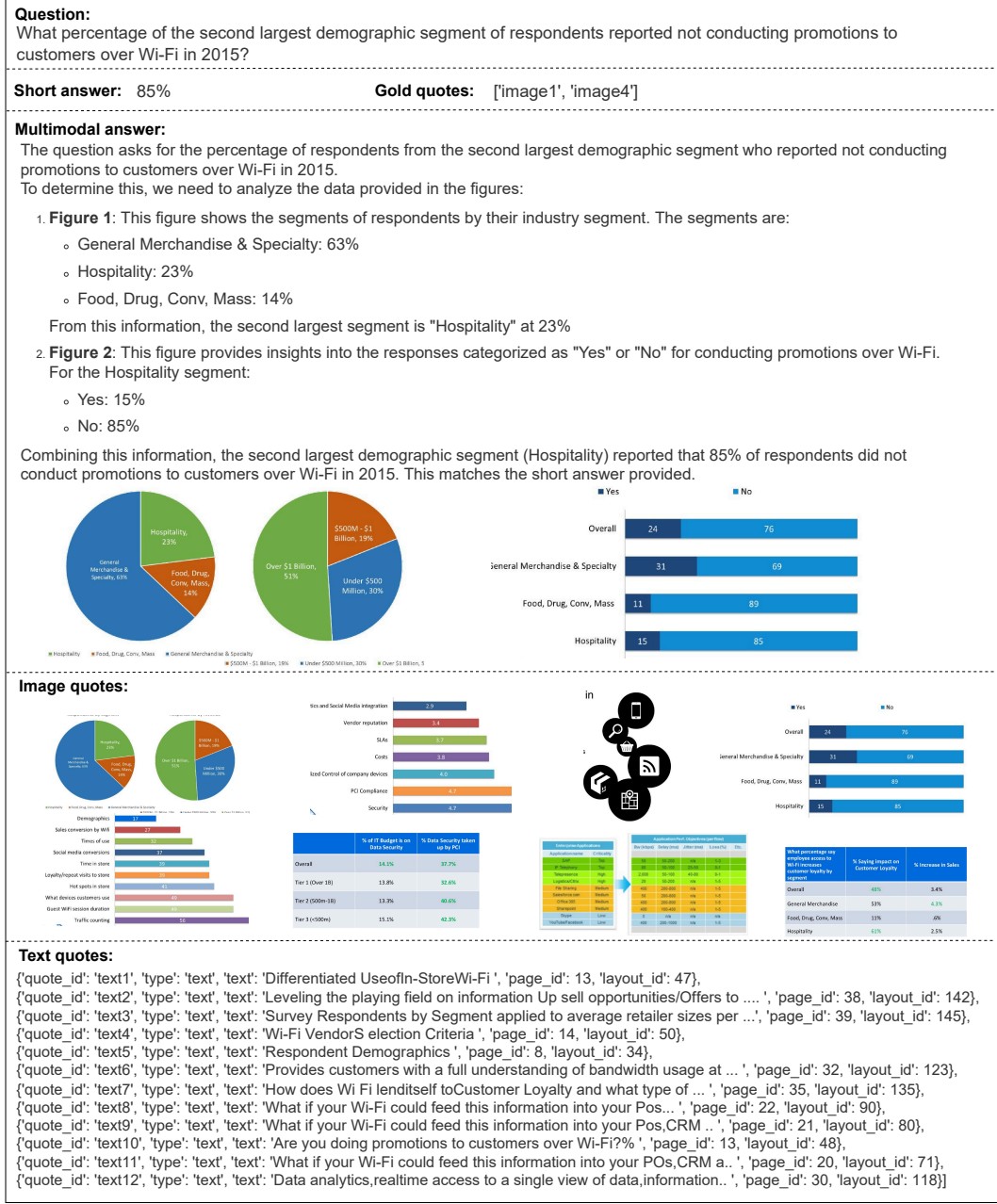

Figure 12: This example shows a typical multi-image reasoning task that requires synthesizing information from multiple image quotes. The answer is derived solely from visual evidence.

**Question:**

What is the difference in total volume between the rank 1 and rank 19 top albums?

**Short answer:** 1194000

**Gold quotes:** ['image3', 'image4']

**Multimodal answer:**

To determine the difference in total sales volume between the top-ranked album and the album ranked at number 19, we need to examine the total volume figures from the given tables.
- The top-ranked album, "1989" by Taylor Swift, has a total sales volume of 1,608,000 units.
- The album ranked at number 19, "My Everything" by Ariana Grande, has a total sales volume of 414,000 units.
The difference in total volume between these two albums is calculated as follows:
Difference = 1,608,000 - 414,000 = 1,194,000
Thus, the difference in total volume between the top-ranked album and the 19th-ranked album is 1,194,000 units.

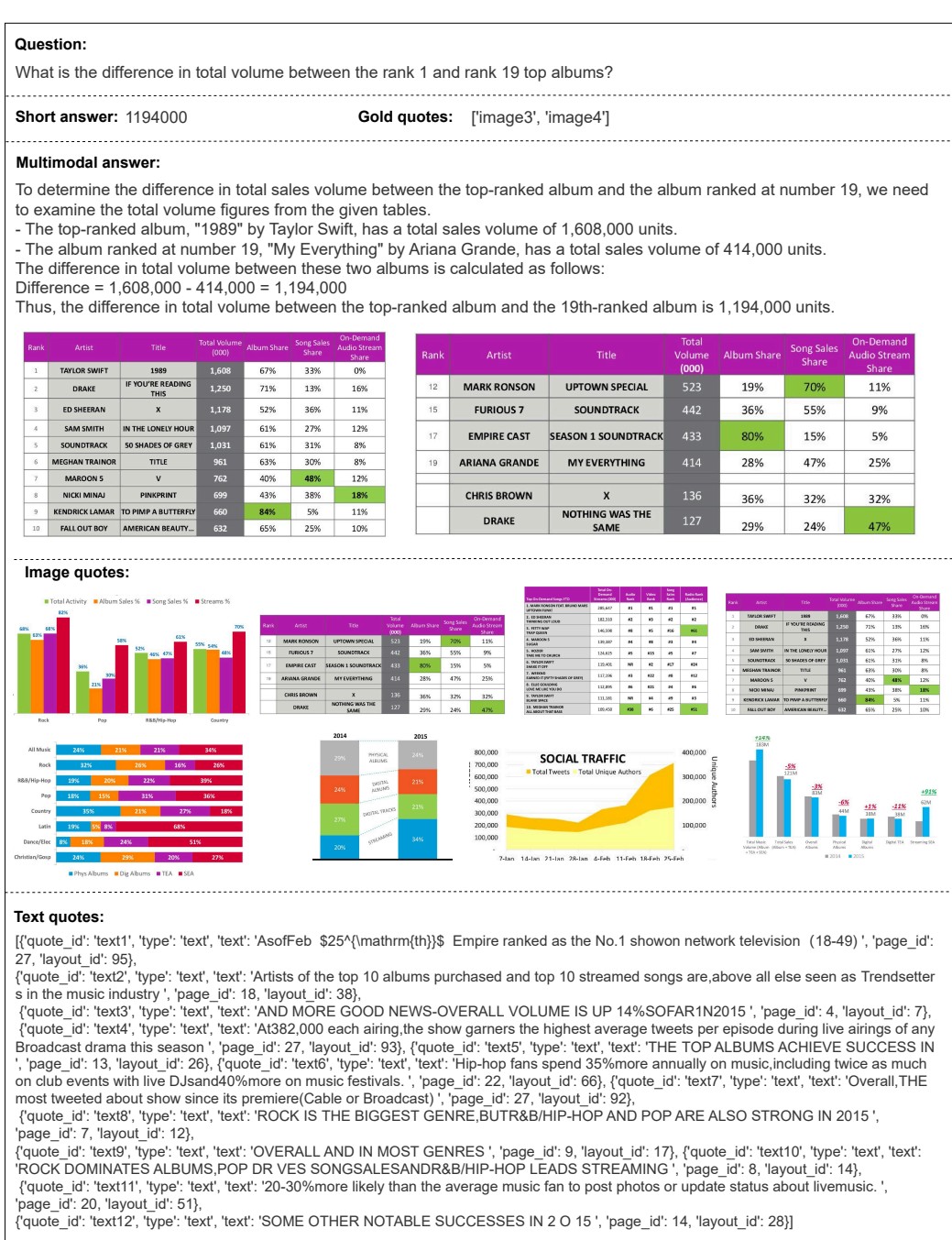

**Image quotes:**

**Text quotes:**

[{'quote_id': 'text1', 'type': 'text', 'text': 'AsofFeb  $25^{\mathrm{th}}$  Empire ranked as the No.1 showon network television  (18-49) ', 'page_id': 27, 'layout_id': 95},

{'quote_id': 'text2', 'type': 'text', 'text': 'Artists of the top 10 albums purchased and top 10 streamed songs are,above all else seen as Trendsetter s in the music industry ', 'page_id': 18, 'layout_id': 38},

 {'quote_id': 'text3', 'type': 'text', 'text': 'AND MORE GOOD NEWS-OVERALL VOLUME IS UP 14%SOFAR1N2015 ', 'page_id': 4, 'layout_id': 7},
 {'quote_id': 'text4', 'type': 'text', 'text': 'At382,000 each airing,the show garners the highest average tweets per episode during live airings of any Broadcast drama this season ', 'page_id': 27, 'layout_id': 93}, {'quote_id': 'text5', 'type': 'text', 'text': 'THE TOP ALBUMS ACHIEVE SUCCESS IN ', 'page_id': 13, 'layout_id': 26}, {'quote_id': 'text6', 'type': 'text', 'text': 'Hip-hop fans spend 35%more annually on music,including twice as much on club events with live DJsand40%more on music festivals. ', 'page_id': 22, 'layout_id': 66}, {'quote_id': 'text7', 'type': 'text', 'text': 'Overall,THE most tweeted about show since its premiere(Cable or Broadcast) ', 'page_id': 27, 'layout_id': 92},

 {'quote_id': 'text8', 'type': 'text', 'text': 'ROCK IS THE BIGGEST GENRE,BUTR&B/HIP-HOP AND POP ARE ALSO STRONG IN 2015 ', 'page_id': 7, 'layout_id': 12},

{'quote_id': 'text9', 'type': 'text', 'text': 'OVERALL AND IN MOST GENRES ', 'page_id': 9, 'layout_id': 17}, {'quote_id': 'text10', 'type': 'text', 'text': 'ROCK DOMINATES ALBUMS,POP DR VES SONGSALESANDR&B/HIP-HOP LEADS STREAMING ', 'page_id': 8, 'layout_id': 14},

 {'quote_id': 'text11', 'type': 'text', 'text': '20-30%more likely than the average music fan to post photos or update status about livemusic. ', 'page_id': 20, 'layout_id': 51},

{'quote_id': 'text12', 'type': 'text', 'text': 'SOME OTHER NOTABLE SUCCESSES IN 2 O 15 ', 'page_id': 14, 'layout_id': 28}]

Figure 13: This example depicts a multi-table quantitative reasoning task. The answer is obtained by performing precise numerical operations based on visual features extracted from multiple tables.

**Question:**

In the year in which the No of internet users was 330 million, what percentage of Indians were Debit Card users?

**Short answer:** 45

**Gold quotes:** ['image1', 'image3']

**Multimodal answer:**

In 2016, when the number of internet users in India reached 330 million, 45% of Indians were debit card users. This information is supported by the data presented in the figures.

- **Figure 1** shows the number of internet users reaching 330 million in 2016.

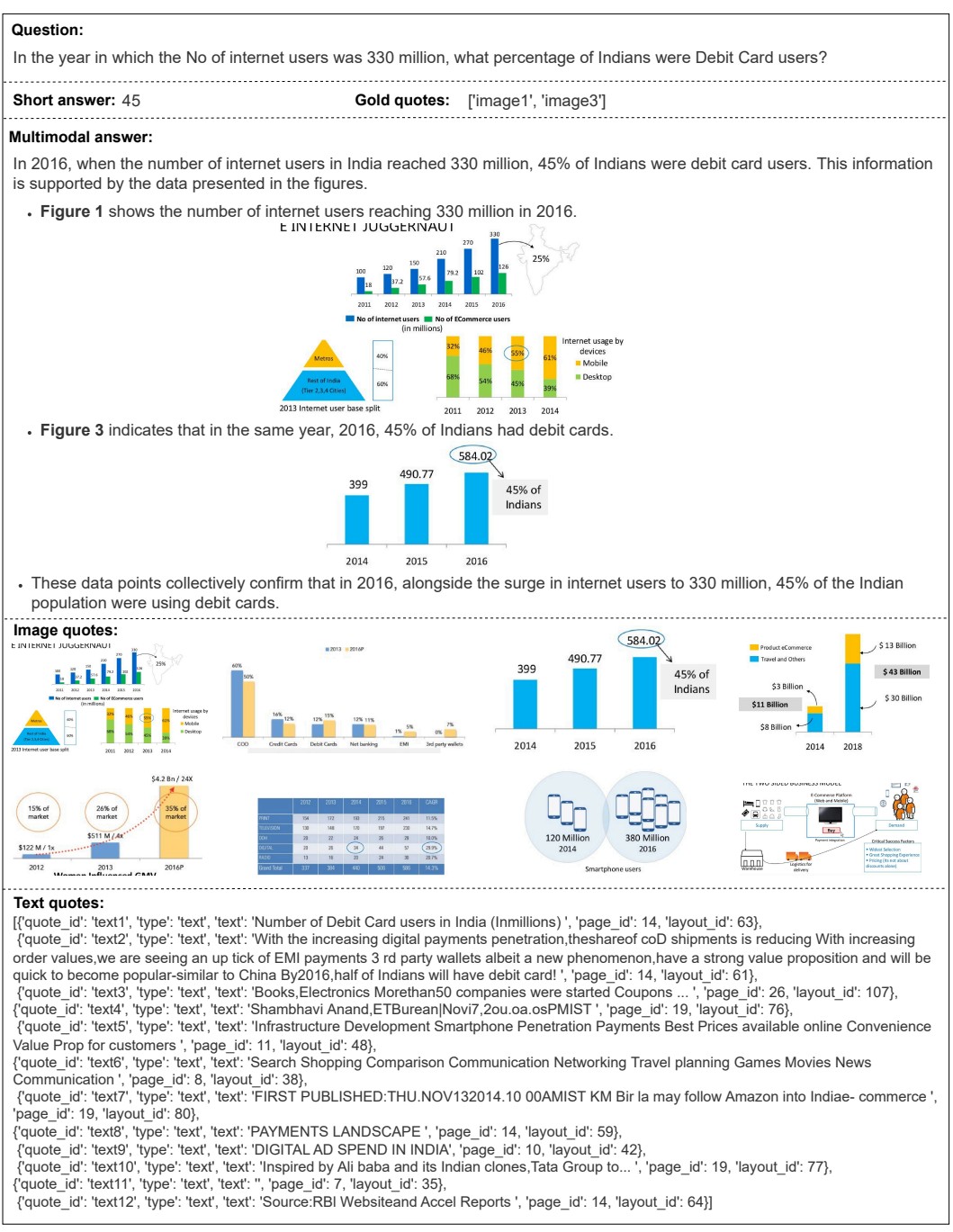

- **Figure 3** indicates that in the same year, 2016, 45% of Indians had debit cards.

- These data points collectively confirm that in 2016, alongside the surge in internet users to 330 million, 45% of the Indian population were using debit cards.

**Image quotes:**

**Text quotes:**

[{'quote_id': 'text1', 'type': 'text', 'text': 'Number of Debit Card users in India (Inmillions) ', 'page_id': 14, 'layout_id': 63},
 {'quote_id': 'text2', 'type': 'text', 'text': 'With the increasing digital payments penetration,theshareof coD shipments is reducing With increasing order values,we are seeing an up tick of EMI payments 3 rd party wallets albeit a new phenomenon,have a strong value proposition and will be quick to become popular-similar to China By2016,half of Indians will have debit card! ', 'page_id': 14, 'layout_id': 61},
 {'quote_id': 'text3', 'type': 'text', 'text': 'Books,Electronics Morethan50 companies were started Coupons ... ', 'page_id': 26, 'layout_id': 107},
 {'quote_id': 'text4', 'type': 'text', 'text': 'Shambhavi Anand,ETBurean|Novi7,2ou.oa.osPMIST ', 'page_id': 19, 'layout_id': 76},
 {'quote_id': 'text5', 'type': 'text', 'text': 'Infrastructure Development Smartphone Penetration Payments Best Prices available online Convenience Value Prop for customers ', 'page_id': 11, 'layout_id': 48},
 {'quote_id': 'text6', 'type': 'text', 'text': 'Search Shopping Comparison Communication Networking Travel planning Games Movies News Communication ', 'page_id': 8, 'layout_id': 38},
 {'quote_id': 'text7', 'type': 'text', 'text': 'FIRST PUBLISHED:THU.NOV132014.10 00AMIST KM Bir la may follow Amazon into Indiae- commerce ', 'page_id': 19, 'layout_id': 80},
 {'quote_id': 'text8', 'type': 'text', 'text': 'PAYMENTS LANDSCAPE ', 'page_id': 14, 'layout_id': 59},
 {'quote_id': 'text9', 'type': 'text', 'text': 'DIGITAL AD SPEND IN INDIA', 'page_id': 10, 'layout_id': 42},
 {'quote_id': 'text10', 'type': 'text', 'text': 'Inspired by Ali baba and its Indian clones,Tata Group to... ', 'page_id': 19, 'layout_id': 77},
 {'quote_id': 'text11', 'type': 'text', 'text': '', 'page_id': 7, 'layout_id': 35},
 {'quote_id': 'text12', 'type': 'text', 'text': 'Source:RBI Websiteand Accel Reports ', 'page_id': 14, 'layout_id': 64}]

Figure 14: This example demonstrates a multimodal alignment task involving both numerical and categorical reasoning. The solution requires aligning and synthesizing temporal and quantitative information across multiple image quotes.

**Question:**
According to this report, which group is greater in population in 2018? Republican who are hispanic, or no leans who are male?

- - -

**Short answer:** no leans who are male       **Gold quotes:**   ['text1', 'text5', 'image4', 'image3']

- - -

**Multimodal answer:**

Based on the information provided in the figures, we can determine the population distribution among different political affiliations and demographic groups in 2018.
In Figure 2, the breakdown is as follows:

- **Hispanic Republicans**: The table shows that 7% of Republicans are Hispanic.
- **No Leans who are Male**: The table indicates that 55% of individuals who have no political lean are male.

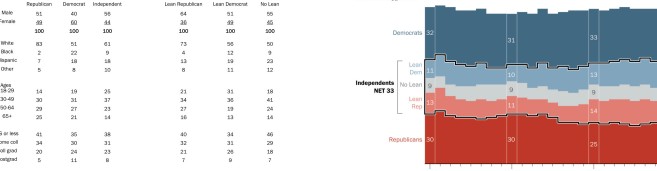

To compare these two groups in terms of population size, we can refer to Figure 1 to understand the overall population proportions:

- **Republicans** accounted for 26% of the total population in 2018**[1]**.
- **No Lean** individuals made up 7% of the total population in 2018**[5]**.

Given these percentages, we can calculate the approximate population shares:

1. **Hispanic Republicans**:
   - Total Republicans in the population: 26%
   - Hispanic Republicans = 7% of 26% = 1.82% of the total population.
2. **No Leans who are Male**:
   - Total No Leans in the population: 7%
   - Male No Leans = 55% of 7% = 3.85% of the total population.

Therefore, the group of "no leans who are male" is larger in population than "Hispanic Republicans" in 2018.

- - -

**Image quotes:**

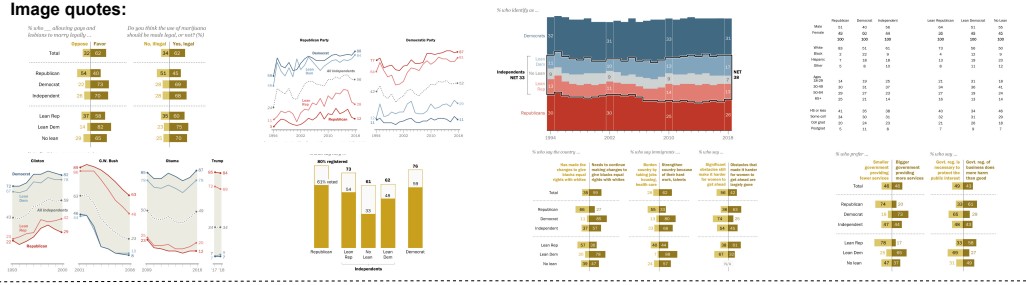

- - -

**Text quotes:**

[{'quote_id': 'text1', 'type': 'text', 'text': 'Among the public overall, $38\%$ describe themselves as independents, while $31\%$ are Democrats and $_{26\%}$ call themselves Republicans, according to Pew Research Center surveys... ', 'page_id': 2, 'layout_id': 10},
{'quote_id': 'text2', 'type': 'text', 'text': 'Currently, $87\%$ of those who identify with the Republican Party view the Democratic Party unfavorably; Republican-leaning independents are almost as likely to view the Democratic Party .... ', 'page_id': 19, 'layout_id': 97},
{'quote_id': 'text3', 'type': 'text', 'text': 'Since 2000, there have been sizable increases in the shares of both Republicans and Republican- leaning independents who identify as conservative. Today, more Republican-leaning independents.... ', 'page_id': 15, 'layout_id': 79},
{'quote_id': 'text4', 'type': 'text', 'text': 'In a survey conducted last fall, shortly after the midterm elections, partisan leaners were less likely than partisans to say they registered to vote and voted in the congressional elections. About half ... ', 'page_id': 4, 'layout_id': 21},
{'quote_id': 'text5', 'type': 'text', 'text': 'An overwhelming majority of independents $(81\%)$ continue to "lean" toward either the Republican Party or the Democratic Party. Among the public overall, $17\%$ are Democratic-leaning independents... ', 'page_id': 3, 'layout_id': 14},
...........................................
{'quote_id': 'text12', 'type': 'text', 'text': 'Compared with partisans, independents are younger and more likely to be men ', 'page_id': 21, 'layout_id': 101}]

Figure 15: This example presents a a typical structure-aware reasoning task. It requires interpreting visual tabular data concerning variables such as political affiliation, ethnicity, and gender, and performing numerical comparisons across multiple image quotes.

**Question:**

Which model achieves the highest F1 score across multiple datasets?

**Short answer:** BERT+DSC achieves the highest F1 score across multiple datasets.    **Gold quotes:** ['text8', 'text6', 'image3', 'image6']

**Multimodal answer:**

BERT+DSC consistently achieves the highest F1 scores across various datasets, both in Chinese and English**[6]**.

**Chinese Datasets**

- **CTB5 Dataset:** BERT+DSC achieves an F1 score of 97.92, the highest among the evaluated models.
- **CTB6 Dataset:** BERT+DSC also records the highest F1 score of 96.57.
- **UD1.4 Dataset:** Again, BERT+DSC leads with an F1 score of 96.98.

| Model | CTB5 Prec. | CTB5 Rec. | CTB5 F1 | CTB6 Prec. | CTB6 Rec. | CTB6 F1 | UD1.4 Prec. | UD1.4 Rec. | UD1.4 F1 |
|---|---|---|---|---|---|---|---|---|---|
| Joint-POS(Sig)(Shao et al., 2017) | 93.68 | 94.47 | 94.07 | - | - | 90.81 | 89.28 | 89.54 | 89.41 |
| Joint-POS(Ens)(Shao et al., 2017) | 93.95 | 94.81 | 94.38 | - | - | - | 89.67 | 89.86 | 89.75 |
| Lattice-LSTM(Zhang and Yang, 2018) | 94.77 | 95.51 | 95.14 | 92.00 | 90.86 | 91.43 | 90.47 | 89.70 | 90.09 |
| BERT-Tagger(Devlin et al., 2018) | 95.86 | 96.26 | **96.06** | 94.91 | 94.63 | **94.77** | 95.42 | 94.17 | **94.79** |
| BERT+FL | 96.11 | 97.42 | 96.76 (+0.70) | 95.80 | 95.08 | 95.44 (+0.67) | 96.33 | 95.85 | 96.81 (+2.02) |
| BERT+DL | 96.77 | 98.87 | 97.81 (+1.75) | 94.08 | 96.12 | 95.09 (+0.32) | 96.10 | 97.79 | 96.94 (+2.15) |
| BERT+DSC | 97.10 | 98.75 | **97.92 (+1.86)** | 96.29 | 96.85 | **96.57 (+1.80)** | 96.24 | 97.73 | **96.98 (+2.19)** |

**English Datasets**

- **English WSJ:** BERT-Tagger+DSC achieves the highest F1 score of 99.38.
- **English Tweets:** BERT-Tagger+DSC achieves the highest F1 score of 92.58.

| English WSJ Model | Prec. | Rec. | F1 |
|---|---|---|---|
| Meta BiLSTM(Bohnet et al., 2018) | - | - | 98.23 |
| BERT-Tagger (Devlin et al., 2018) | 99.21 | 98.36 | **98.86** |
| BERT-Tagger+FL | 98.36 | 98.97 | 98.88 (+0.02) |
| BERT-Tagger+DL | 99.34 | 98.22 | 98.91 (+0.05) |
| BERT-Tagger+DSC | 99.41 | 98.93 | **99.38 (+0.52)** |

| English Tweets Model | Prec. | Rec. | F1 |
|---|---|---|---|
| FastText+CNN+CRF(Godin, 2019) | - | - | 91.78 |
| BERT-Tagger (Devlin et al., 2018) | 92.33 | 91.98 | **92.34** |
| BERT-Tagger+FL | 91.24 | 93.22 | 92.47 (+0.13) |
| BERT-Tagger+DL | 91.44 | 92.88 | 92.52 (+0.18) |
| BERT-Tagger+DSC | 92.87 | 93.54 | **92.58 (+0.24)** |

These results demonstrate that BERT+DSC is a robust model that consistently performs well across different datasets and languages, achieving superior precision, recall, and F1 scores compared to other models. This consistency suggests it effectively handles various linguistic challenges, making it a strong choice for part-of-speech tagging tasks**[8]**.

**Image quotes:**

**Text quotes:**

[{'quote_id': 'text1', 'type': 'text', 'text': '• ELMo: a tagging model with pretraining from  Peters et al.  ( 2018 ). • Lattice-LSTM:  Zhang and Yang ( 2018 ) con- structs a word-character lattice, only used in Chinese datasets. • CVT: Clark ....', 'page_id': 5, 'layout_id': 78}

{'quote_id': 'text6', 'type': 'text', 'text': 'DSC achieves the highest F1 score across all datasets. Specially, for $^+$ positive , DSC achieves minor improvements  $(+0.05\,\mathrm{F}1)$  ) over DL. In con- trast, it significantly outperforms DL ...', 'page_id': 7, 'layout_id': 100},

{'quote_id': 'text7', 'type': 'text', 'text': 'In Table  2 , we summarize all the aforementioned losses. Figure  1  gives an explanation from the per- spective in derivative: The derivative of DSC ap- proaches zero right after   $p$   exceeds 0.5, which sug...', 'page_id': 4, 'layout_id': 63},

{'quote_id': 'text8', 'type': 'text', 'text': 'Results Table  3  presents the experimental results on Chinese datasets. As can be seen, the proposed DSC loss outperforms the best baseline results by a large margin, i.e., outperforming BERT-tagger by  ... ', 'page_id': 4, 'layout_id': 70},

{'quote_id': 'text12', 'type': 'text', 'text': 'Results Table  5  shows experimental results on NER datasets. DSC outperforms BERT-MRC( Li et al. , 2019 ) by  $+0.29$  ,  $+0.96$  ,  $+0.97$  and $+2.36$  re- spectively on CoNLL2003, OntoNotes5.0, MSRA and OntoNotes4.0. As far as we are concerned, we are setting new SOTA performances on all of the four NER datasets. ', 'page_id': 5, 'layout_id': 79}]

Figure 16: This example illustrates a comparative reasoning task, which requires scanning multiple structured tables, applying numerical reasoning, achieving visual alignment, and making global comparisons among multiple tables.

**Question:**

What was the change in the total fair value of marketable securities from January 31, 2019, to January 31, 2020?

**Short answer:** The total fair value increased by $2,129 million.  **Gold quotes:** ['text10', 'text11', 'image8', 'image3']

**Multimodal answer:**

The total fair value of marketable securities on January 31, 2020, was $3,802 million**[10]**, compared to $1,673 million on January 31, 2019**[11]**. This indicates an increase of $2,129 million in the total fair value over the year.

| Investments classified as Marketable Securities | Amortized Cost | Unrealized Gains | Unrealized Losses | Fair Value | | Investments classified as Marketable Securities | Amortized Cost | Unrealized Gains | Unrealized Losses | Fair Value |
|---|---|---|---|---|---|---|---|---|---|---|
| Corporate notes and obligations | $2,199 | $ 9 | $(1) | $2,207 | | Corporate notes and obligations | $1,027 | $0 | $ (8) | $1,019 |
| U.S. treasury securities | 182 | 1 | 0 | 183 | | U.S. treasury securities | 89 | 0 | (1) | 88 |
| Mortgage backed obligations | 225 | 1 | 0 | 226 | | Mortgage backed obligations | 79 | 0 | (1) | 78 |
| Asset backed securities | 779 | 2 | 0 | 781 | | Asset backed securities | 245 | 0 | (1) | 244 |
| Municipal securities | 157 | 1 | 0 | 158 | | Municipal securities | 104 | 0 | 0 | 104 |
| Foreign government obligations | 69 | 0 | 0 | 69 | | Foreign government obligations | 58 | 0 | (1) | 57 |
| U.S. agency obligations | 12 | 0 | 0 | 12 | | U.S. agency obligations | 4 | 0 | 0 | 4 |
| Time deposits | 1 | 0 | 0 | 1 | | Time deposits | 4 | 0 | 0 | 4 |
| Covered bonds | 165 | 0 | 0 | 165 | | Covered bonds | 75 | 0 | 0 | 75 |
| Total marketable securities | $3,789 | $14 | $(1) | $3,802 | | Total marketable securities | $1,685 | $0 | $(12) | $1,673 |

The tables above show the detailed breakdown of different types of marketable securities and their respective fair values for the years 2020 and 2019. The increase in the total fair value could be attributed to changes in the market value of these securities, as well as possibly changes in the composition or volume of the securities held.

**Image quotes:**

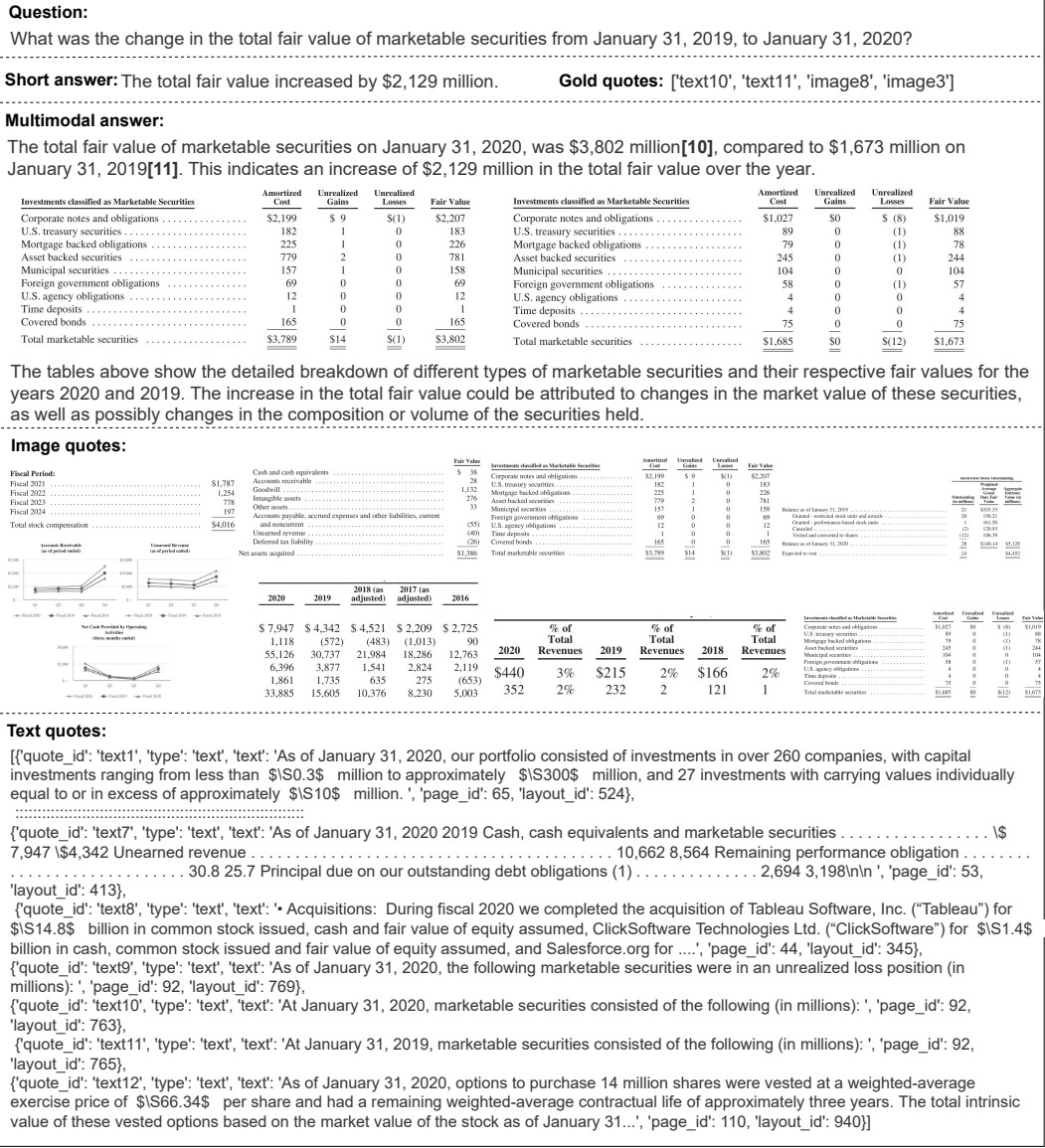

**Text quotes:**

[{'quote_id': 'text1', 'type': 'text', 'text': 'As of January 31, 2020, our portfolio consisted of investments in over 260 companies, with capital investments ranging from less than $\S0.3$ million to approximately $\S300$ million, and 27 investments with carrying values individually equal to or in excess of approximately $\S10$ million. ', 'page_id': 65, 'layout_id': 524},
:::::::::::::::::::::::::::::::::::::::::::::::::::::::::::

{'quote_id': 'text7', 'type': 'text', 'text': 'As of January 31, 2020 2019 Cash, cash equivalents and marketable securities . . . . . . . . . . . . . . . . . \$ 7,947 \$4,342 Unearned revenue . . . . . . . . . . . . . . . . . . . . . . . . . . . . . . . . . . . . . . . . . 10,662 8,564 Remaining performance obligation . . . . . . . . . . . . . . . . . . . . . . . . . 30.8 25.7 Principal due on our outstanding debt obligations (1) . . . . . . . . . . . . . 2,694 3,198\n\n ', 'page_id': 53, 'layout_id': 413},

{'quote_id': 'text8', 'type': 'text', 'text': '• Acquisitions: During fiscal 2020 we completed the acquisition of Tableau Software, Inc. ("Tableau") for $\S14.8$ billion in common stock issued, cash and fair value of equity assumed, ClickSoftware Technologies Ltd. ("ClickSoftware") for $\S1.4$ billion in cash, common stock issued and fair value of equity assumed, and Salesforce.org for ....', 'page_id': 44, 'layout_id': 345},

{'quote_id': 'text9', 'type': 'text', 'text': 'As of January 31, 2020, the following marketable securities were in an unrealized loss position (in millions): ', 'page_id': 92, 'layout_id': 769},

{'quote_id': 'text10', 'type': 'text', 'text': 'At January 31, 2020, marketable securities consisted of the following (in millions): ', 'page_id': 92, 'layout_id': 763},

{'quote_id': 'text11', 'type': 'text', 'text': 'At January 31, 2019, marketable securities consisted of the following (in millions): ', 'page_id': 92, 'layout_id': 765},

{'quote_id': 'text12', 'type': 'text', 'text': 'As of January 31, 2020, options to purchase 14 million shares were vested at a weighted-average exercise price of $\S66.34$ per share and had a remaining weighted-average contractual life of approximately three years. The total intrinsic value of these vested options based on the market value of the stock as of January 31...', 'page_id': 110, 'layout_id': 940}]

Figure 17: This example displays a table-based numerical reasoning task, which requires extracting structured financial values from visually similar but distinct tables. This can also reflect model's ability to perform numerical reasoning over extracted values.

# E   Prompt Instructions

## E.1   Dataset Creation

According to Section 2.1, we generate the initial multimodal answer based on the question, document page screenshots, cropped images, and text snippets, using the prompt template specified in Figure 18. We then explicit cite the gold quotes in the generated multimodal answer using the prompt template illustrated in Figure 19.

## E.2   Dataset Quality Assurance

According to Section 2.3, we leverage on automated validation on our initial multimodal answer. Specifically, we use VLMs to examine the generated multimodal answer on whether it selects and inserts relevant visual content coherently, via the prompt shown in Figure 20. Meanwhile, we use LLM to check the accuracy and coherence of integrated text, via the prompt shown in Figure 21.

## E.3   Inference using Pure-text/Multimodal Quotes

According to Section 4.2 and Appendix C.1, we formulate multimodal answer generation by representing multimodal quotes in two formats: (i) multimodal (interleaved text-image) sequence for VLM, and (ii) pure-text sequence for both VLM and LLM. For multimodal answer generation using multimodal inputs, we use the prompt template illustrated in Figure 22. For multimodal answer generation using pure-text inputs, we use the prompt template illustrated in Figure 23.

## E.4   LLM Evaluation

According to Section 4.1 and Appendix A.2, we adopt LLM-as-Judge as evaluation criteria for multimodal answer generation. Specifically, we use the prompt template shown in Figure 24, which scores the generated answer from five key aspects: fluency, citation quality, text-image coherence, reasoning logic, and factuality.

# Task description
You are good at understanding multi-modal documents/pages and generating comprehensive multi-modal answer.

**Task**: You are given a question and its short answer, along with its supporting evidence. You need to generate a more comprehensive answer. The answer should contain multimodal information extracted from the supporting evidence.

## 1. Understand Evidence
1.1 The given evidence can be multiple screenshot pages of a document/webpage.
- the screenshots contain rich multimodal information, including text, images, and tables.
- understand the number of screenshots pages: if there is only one screenshot, the question pertains to a single page; if there are multiple screenshots, the question involves multiple pages.
- Determine the type of multimodal data present and detect the quantity of images or tables within the screenshots.
1.2 The given evidence can also be texts, the texts can contain useful information for you to understand the question and answer.
- the texts is extracted from screenshots and contain useful information that can help you understand the evidence
1.3 The given evidence can also be cropped figures
- the figures is extracted from screenshots and contain very useful information for you
- the number of figures is not specific, if there is only one figure, you need understand and generate the comprehensive answer through this figure; if there are many figures, you need understand and generate the comprehensive answer through all figures.
- you need understand the figures carefully, include the name of the figures, the content of the figure and the detail number of the figures if it contain specific quantitative information. For example, for tables, describe each row and column, highlighting important figures related to the question, for images, describe the content, focusing on elements related to the question, such as colours, quantities, people, etc.
- summarise key information related to the questions and answers, explaining how the given answer is generated based on this information.

## 2. Question Understanding
- understand the given question, the short answer is used to facilitate your understanding.
- extract the supporting text/multi-modal information (e.g., figures/tables in the given evidence), if cropped pictures is provided, you can directly use cropped pictures to understand.

## 3. Comprehensive Answer Generation:
3.1 **Answer Output Format**:
- the response must be presented in Markdown format, the answer need to be interleaved image/text. Note that do not need too much title or other information.
3.2 **Figure insert**
- you only need to insert the useful figure, and the figures must be chosen from the cropped figures instead of screenshots.
- figure insert format, when inserting multimodal information, use the format `` where "figure" is the specific cropped figure sequence, for example, if you insert the first given cropped figure, use ``; if you insert the second given cropped figure, use ``, the sequence is very important, please do not make error.
- figure insert position, you have flexibility in placement: it can be above or below the analysis, and if there are multiple insertions, they can be grouped together or interspersed between analyses, based on the understanding and clarity of your response.
3.3 **Answer styles:** based on different question types, you can have flexibility in answer type.
- if the question is an exam question or seeks a direct answer, we encourage providing the conclusion first, followed by an explanation or detailed description.
- if the answer involves multiple steps in a specific order, we encourage a step-by-step format, with one step per line.
- if the answer involves multiple aspects or requires listing several points, we encourage a bullet-point format with detailed descriptions for each point.
- if the answer relates to causes, processes, or circumstances, we encourage using appropriate paragraphing to provide detailed explanations.
- for multiple-choice, true/false, or fill-in-the-blank questions, directly provide the corresponding answer first, followed by an explanation or detailed description.
- for complex questions or when the answer covers a broad scope, we encourage combining different response formats.

Figure 18: Prompt template for generating the initial multimodal answer based on the question, document page screenshots, cropped images, and text snippets.

# System Prompt:
You are good at question answering. You are given the question, short answer, interleaved text-image long answer. You need to understand the provided text passages, and decide if any text passages are relevant to the answers. Finally, you need to quote relevant text passages in the correct place.

# 1. Understanding the Question and Answer
- The short answer is provided to you to facilitate your understanding;
- The interleaved long answer is provided to you for fine-grained understanding;

# 2. Selecting Evidence from Text Passages
- You need to decide if the provided text passages are relevant to the question and answer;
- Relevant text passage is helpful for question understanding and can be quoted by the long answer;
- Irrelevant text passage provides no useful information for question understanding and cannot be quoted by the long answer;
- Relevant here refers to content that includes necessary fragments of information from the interleaved long answer. Since the interleaved long answer is quite long, some text fragments, such as paragraph titles, table names, sheet names, or image captions, although they may exactly match parts of the long answer, should not be selected because they are too short and do not contribute significantly to the answer;
- Useful information refers to the essential content needed to derive the short answer from long answer, such as key numbers, important definitions, crucial comparisons, etc. Without these, the answer cannot be properly deduced. On the other hand, broad or vague descriptions cannot be selected as useful information;
- The selected evidence must contain the key elements, which refer to the necessary components required in the steps to derive the short answer from the long answer. It should not be a simple semantic match based on the long answer;
- Some entries that merely describe definitions or detailed explanations of certain text fragments in the long answer should not be selected;

- Entries that describe situations identical to those in the long answer but lack critical keys should also not be selected;
- If there is no relevant text passages, set "need_text"=False;
- If there are any relevant text passages, set "need_text"=True;
- If "need_text"==True, please select the relevant text by choosing the text passage indices;

- Note: Do not forcibly select evidence, only select evidence that is fully or strongly relevant. If there is no such evidence, then it should be considered as having no evidence, avoid making forced associations just to select evidence;

# 3. Citing/Quoting Text Passage Indices in Long Answer
- This step is only applicable when "need_text"==True;
- If "need_text"==True, you need to insert the text passage indices into the long answer;
- Make sure answer text at the insertion positions is relevant to the text passage;
- You need to re-evaluate the evidence you have chosen. If you cannot find a suitable position to insert it, you should abandon that piece of evidence;
- Every piece of evidence selected must correspond to the keys in the long answer, meaning it must be eligible for annotation insertion;
- Do not change the content of long answer; you must insert only the index in the format of "[index]";
- Under no circumstances should you add or remove any other words from the original answer. This task strictly involves adding annotations in the form of "[index]" without altering the original text in any other way;
- All evidence in evidence_indices must be inserted into the answer. If you cannot find a suitable insertion position, you must discard that piece of evidence;

# Output Instructions
Return the (1) the status of "need_text=True/False" (2) evidence indices, and (3) modified long answer in the following json format, and the long answer text need to be Markdown format:
"need-text": Boolean, "evidence-indices": [...], "long-answer": "..."

Figure 19: Prompt template to support fine-grained text passage selection and citation in multimodal question answering.

# System Prompt:
You are a robust vision-language evaluator. Your task is to automatically assess whether a given multimodal answer (with text interleaved with figures/images) correctly and coherently selects and inserts the most relevant visual content as supporting evidence.

**You will be provided with:**
- The original question and its short answer;
- The full set of available cropped figures (named, sequenced, and described in the prompt);
- The generated multimodal answer, formatted in Markdown, with  syntax for image insertion;

**Your assessment process:**
1. **Relevance of Figure Selection:**
- Examine whether the answer selects only those figures relevant to the question and the answer;
- Check if any crucial/required visual evidence has been ignored or omitted;
2. **Accuracy and Clarity of Figure Insertions:**
- Verify the figures are inserted correctly by referencing the right sequence (i.e., figure1, figure2, etc.) and that the associated description (name) matches the actual content;
- Check that figures are placed in a way that makes sense, aiding interpretation rather than confusing the reader;

3. **Coherence and Support:**
- Determine if the inserted figures clearly support, elaborate, or justify the accompanying text at appropriate narrative points;
- Evaluate whether the integration of images enhances understanding and directly relates to the explanation or answer, maintaining logical and coherent flow;

**Scoring & Output**
For each of the following, rate on a scale from 0 (not at all) to 5 (perfect):
- **Figure Relevance**: Are all inserted figures relevant and necessary, with no missing or irrelevant ones;
- **Insertion Accuracy**: Are all figures referenced and inserted in the right sequence and with correct names/descriptions;
- **Image-Text Coherence**: Does the placement and use of figures improve understanding and logically connect with the accompanying explanation/text;

Report results as a JSON object with this format:
{"Figure Relevance": <score>, "Insertion Accuracy": <score>, "Image-Text Coherence": <score> }

Assign only integer scores. Do not include explanations, comments, or any text outside the above JSON.

Figure 20: Prompt template for using VLMs to examine the generated multimodal answer on whether it selects and inserts relevant visual content coherently.

# System Prompt:
You are an expert answer validation assistant specializing in language comprehension and content evaluation. Your task is to automatically assess a generated multimodal answer, focusing exclusively on the accuracy and coherence of the integrated textual explanation.

**You will be provided with:**
- The original question and its short answer;
- he full supporting evidence (including any extracted texts, descriptions of images/tables, figure captions, etc.);
- The initial multimodal answer, with text and figure placeholders (e.g., );

**Your assessment process:**
1. **Comprehension & Alignment:**
- Fully understand the question and required information;
- Review the provided supporting evidence, including any relevant extracted texts or descriptions;
2. **Accuracy of Integrated Text:**
- Examine whether the text portions of the multimodal answer accurately address the question, are factually correct, and are clearly derived from the supporting evidence;
- Check logical consistency and factuality between the cited evidence and the short answer;
- Assess if any essential information from the evidence is omitted or incorrectly incorporated;

3. **Coherence of Explanation:**
- Determine whether the explanation flows logically and is easy to read;
- Evaluate whether the textual content is well-structured, connects naturally with cited visual content (even if you do not evaluate the visuals themselves), and supports the main answer;
- Ensure that the explanation has no serious redundancy or ambiguity

**Scoring & Output**
For each of the following, rate on a scale from 0 (not at all) to 5 (perfect):
- **Textual Accuracy**: Does the answer's text correctly reflect the question and evidence, with no significant factual errors or gaps;
- **Textual Coherence**: Is the textual explanation clear, well-organized, and logically connected to the overall answer;

Report results as a JSON object with this format:
{"Textual Accuracy": <score>, "Textual Coherence": <score> }

Assign only integer scores. Do not include explanations, comments, or any text outside the above JSON.

Figure 21: Prompt template for using LLMs to check the accuracy and coherence of integrated text in the generated multimodal answer.

**# System Prompt:**
You are a helpful question-answering assistant. Your task is to generate an interleaved text and image response based on provided questions and quotes.
- Note that 'interleaved text and image response' refers to a format where both text and images are presented together in an alternating manner.

**1. Evidence Selection**
- Carefully read and understand the question, identifying the key evidence it requires;
- Carefully analyze and comprehend text and image quotes, accurately identifying the key information they contain;
- From both text and image quotes, pinpoint those that are really relevant for answering the question. Focus on significance and direct relevance;

**2. Answer Construction**
- Use Markdown to embed text and images in your response;
- Depending on the question type:
  • Employ a sequential format for procedural queries;
  • Use bullet points for questions needing a list-based response;
  • Write in paragraphs for detailed explorations of causes or processes;
  • Merge response styles for complex queries to ensure complete coverage;

  • Conclude with a direct and concise answer to the question in a simple and clear sentence;

**3. Quote Citation**
- Cite text by adding [text index]; for example, quote from the first text should be [1];
- Use `![{conclusion}](image index)` format for the first image, use `` to cite images; The conclusion should be a concise one-sentence summary of the image's content;
- Flexibly place image citations dependent on their contribution to text explanation—either above or below the related analysis, or group multiple images as needed;
- - - - - - - - - - - - - - - - - - - - - - - - - - - -
**# User Message:**
**1. Text Quotes** are:
   - [1] {text quote 1}
   ...
   - [12] {text quote 12}

**2. Image Quotes** are:
   - image1 is: data:image/jpeg;base64,{base64 encoding of image quote 1}
   ...
   - image8 is: data:image/jpeg;base64,{base64 encoding of image quote 8}

**3. User question** is: {question}

Figure 22: Prompt template for inputting multimodal (interleaved text-image) sequence to VLM for multimodal answer generation.

**# System Prompt:**
You are a helpful question-answering assistant. Your task is to generate an interleaved text and image response based on provided questions and quotes.

**Note:** 'Interleaved text and image response' refers to a format where both text and images are presented together in an alternating manner.

**1. Evidence Selection**
- Carefully read and understand the question, identifying the key evidence it requires.
- Carefully read and understand all the quotes, identifying the key information they contain.
- From both text and image quotes, pinpoint those really relevant for answering the question. Focus on significance and direct relevance.
- Each image quote is the description of the image.

**2. Answer Construction**
- Use Markdown to embed text and images in your response.
- Depending on the question type:
  • Employ a sequential format for procedural queries;
  • Use bullet points for questions needing a list-based response;
  • Write in paragraphs for detailed explorations of causes or processes;
  • Merge response styles for complex queries to ensure complete coverage;

  • Conclude with a direct and concise answer to the question in a simple and clear sentence.

**3. Quote Citation**
- Cite text by adding [text index]; for example, quote from the first text should be [1].
- Use `![{conclusion}](image index)` format to cite images; for the first image, use ``. The `{conclusion}` should be a concise one-sentence summary of the image's content.
- Flexibly place image citations based on their contribution to text explanation—either above or below the related analysis, or group multiple images as needed.
- - - - - - - - - - - - - - - - - - - - - - - - - - - -
**# User Message:**
**1. Text Quotes** are:
   - [1] {text quote 1}
   ...
   - [12] {text quote 12}

**2. Image Quotes** are:
   - image1 is described as: {VLM-text or OCR-text of image quote 1}
   ...
   - image8 is described as: {VLM-text or OCR-text of image quote 8}

**3. User question** is: {question}

Figure 23: Prompt template for inputting multimodal quotes as pure-text sequence to both LLM and VLM for multimodal answer generation.

# System Prompt:
You are a helpful content evaluation assistant. You will receive a question, a short answer, a perfect answer, and an interleaved answer. Your task is to evaluate the quality of the interleaved answer with scores.

# 1. Understand Evidence
- Analyze and comprehend the question and short answer, identifying the key evidence it requires;
- Analyze and comprehend the perfect answer, accurately identifying the key information it contains;
- Analyze and comprehend the interleaved answer, identifying the information it contains.
- In the interleaved answer, images are cited using the format `![{summary}](image index)`, where `summary` corresponds to a short summary of the image; texts are cited using the `[text{quote\_id}]` format.

# 2. Scoring Criteria
Evaluate the quality of the interleaved answer based on the following scoring criteria, assigning a specific score for each aspect:
- 0: The answer completely fails to meet the requirement, or is entirely irrelevant.
- 1: The answer completely fails to meet the requirement, with significant errors, missing information, or weak justification that severely impact the overall quality.
- 2: The answer partly meets the requirement but contains noticeable gaps, minor inaccuracies, or readability issues.
- 3: The answer moderately meets the requirement, but small inconsistencies, lack of clarity, or minor justification issues remain.
- 4: The answer largely meets the requirement with minor imperfections.
- 5: The answer perfectly meets the requirement, is flawless, well-structured, and highly relevant.

# 3. Scoring Aspects
The following scoring criteria are independent of each other. When scoring, make sure each item is evaluated independently, objectively, and fairly. One option should not influence the scores of other options.
- 1. **Fluency**: Is the interleaved answer grammatically correct, coherent, and easy to read? Does it flow naturally?
- 2. **Citation Quality**: Is the placement of the citation positioned appropriately? Does the citation appear at a key point in the response where it is necessary for supporting the answer, or is its placement illogical or irrelevant?
- 3. **Text-Image Coherence**: Through image summary, do the text and image complement each other seamlessly? Is each image integrated into the narrative in a way that enhances the overall understanding?
- 4. **Reasoning Logic**: Does the interleaved answer follow a logical, well-structured, and clear reasoning process? Check if the steps taken are rational and systematic.
- 5. **Factuality**: Does the interleaved answer's overall reasoning and framework align with the perfect answer? Are there any major factual inaccuracies or misleading information?

# 4. Response
The response should be structured as a JSON object following this fixed format:
`{'Aspect': score}`
For example, the response should be:
'Fluency': score, 'Citation Quality': score, 'Text-Image Coherence': score, 'Reasoning Logic': score, 'Factuality': score
Provide only the integer scores in the specified format. Do not include additional details beyond the score.

Figure 24: Prompt template for adopting LLM-as-Judge as evaluation criteria for multimodal answer generation. It scores the generated answer from five key aspects: fluency, citation quality, text-image coherence, reasoning logic, and factuality.

# F Qualitative Study

In this section, we present a qualitative study on the quality of multimodal answer generation for existing and finetuned large models, comprising (F.1) error analysis for four typical errors, (F.2) performance comparison of VLM by using multimodal and pure-text quotes for multimodal generation, and (F.3) assessment of finetuning effectiveness.

## F.1 Error Analysis: Qualitative Study on 4 Common Errors

To gain a comprehensive understanding of model competence beyond quantitative scores, we conduct a detailed error analysis of multimodal (interleaved text-image) answers generated by GPT-4o [55] compared to gold answers in MMDocRAG. We manually analyzed 200 cases to identify recurrent issues.

For **citing quality**, we identify the following primary errors:

- **Excessive Citation**: The model often over-cites irrelevant images or fails to select the most relevant ones. Confusion among similar images frequently leads to incorrect selections, and repeated citation of the same image is common. For text, the model sometimes cites irrelevant or duplicate passages. This issue was present in approximately 34.5% of cases.
- **Inadequate Citation**: The model occasionally cites only one primary image or omits relevant images needed for a complete answer. Similarly, for text, it sometimes fails to cite the most pertinent excerpts, indicating challenges in extracting meaningful information. This occurred in about 30.0% of cases.
- **Citation Position**: Citations are sometimes placed out of alignment with the relevant sentences, observed in approximately 16.5% of cases.

Regarding **reasoning and factual consistency**, the model sometimes fails to fully comprehend visual content, omitting crucial information or selecting incorrect but similar images. This results in inaccurate or incomplete answers, highlighting the need for improved image discrimination and logical reasoning.

In terms of **text-image coherence**, we frequently observe mismatches between the model's citation placement and the gold standard. While this does not significantly impact answer correctness, it affects answer coherence and highlights subjective aspects of evaluating multimodal integration. Nonetheless, image placement is generally satisfactory and reflects flexible interleaving rather than a rigid order.

For **fluency**, most generated answers are linguistically coherent, as large language models typically produce fluent, high-quality sentences with few grammatical errors.

In summary, although the model demonstrates strong language fluency, there remains considerable room for improvement in visual understanding, multimodal integration, and citation coherence. Enhancing logical reasoning and the alignment of cited evidence is essential for further improving overall model performance.

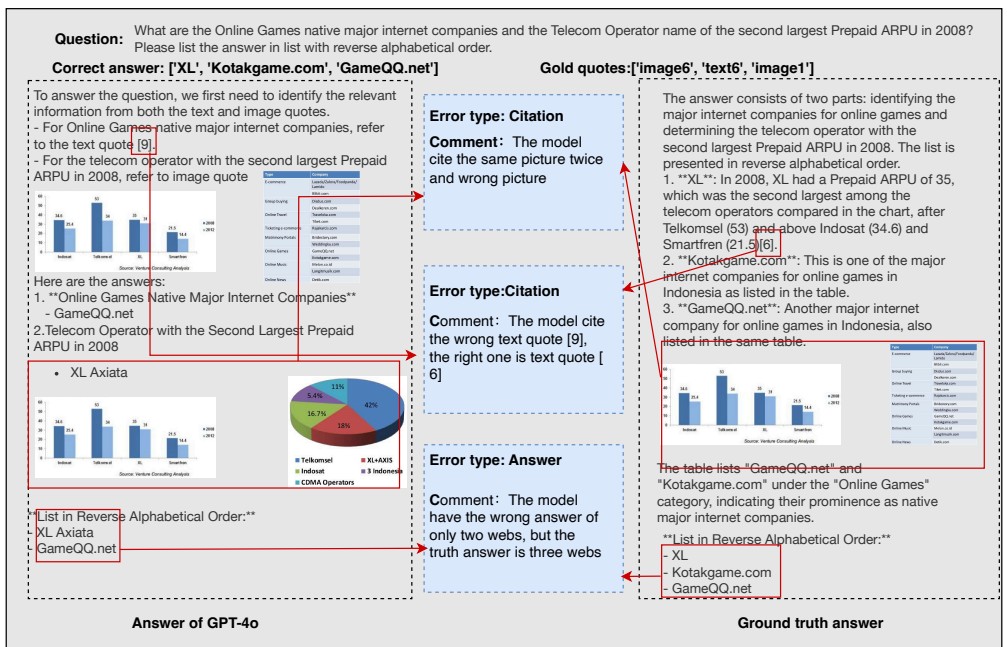

Figure 25: This example demonstrates that the model repeatedly cited the same figure and referenced incorrect textual passages, resulting in an incorrect final answer.

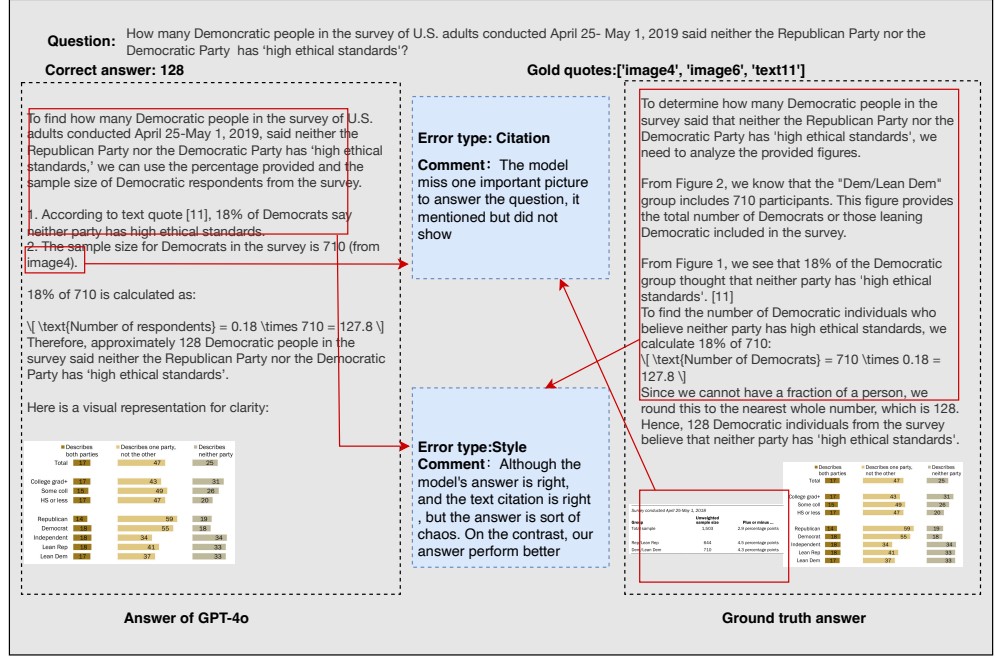

Figure 26: This example demonstrates that the model failed to cite the key figure as a reference. As a result, the answer is poorly organized and lacks logical coherence, making it difficult to follow.

## F.2 Multimodal vs Pure-text Quotes: Qualitative Analysis

As discussed in Section 4.4, we compare model performance when quotes are provided as either pure-text or multimodal (interleaved text-image) inputs. The quantitative results are presented in Table 9 and Table 4. To further illustrate the differences beyond quantitative scores, we perform a detailed qualitative analysis contrasting interleaved text-image inputs with pure-text inputs.

GPT-4o demonstrates moderate advantages in multimodal reasoning when provided with original images. The model accurately interprets and integrates visual details, enabling the identification and extraction of key information that is often missed when relying solely on text descriptions.

In terms of **citation quality**, pure-text input increases the likelihood of incorrect or missed image citations. The model is more prone to confusing visually similar but semantically different images, which leads to citation errors and, ultimately, incorrect answers. In contrast, directly providing original images enables the model to achieve higher citation precision and stronger evidence grounding.

Regarding **answer quality**, text-only inputs sometimes result in hallucinations or factual inaccuracies, as the model fails to capture critical visual information. Nevertheless, GPT-4o still maintains comparable logical coherence and, to some extent, factuality in its text-based responses, suggesting that advanced VLMs can leverage textual context effectively, but substantial advantages are realized when visual content is directly accessible.

In summary, for advanced VLMs like GPT-4o, providing original images substantially improves citation accuracy, factual grounding, and multimodal reasoning. While these models exhibit strong language-based reasoning, integrating visual inputs is essential for achieving optimal performance on multimodal tasks. In contrast, VLMs with smaller model sizes struggle to interpret and integrate information from multiple images within an input sequence, resulting in decreased performance on multimodal tasks (see Figure 29). For these less advanced models, it is generally preferable to use pure-text inputs, as they process textual information more reliably than complex multimodal content.

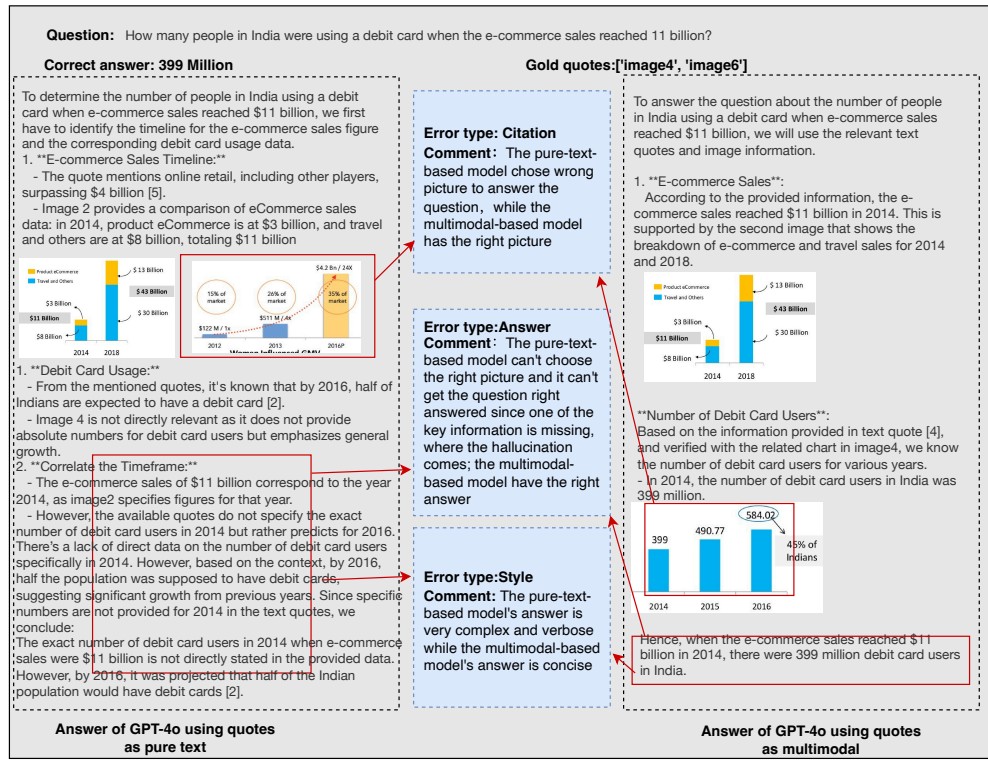

Figure 27: This example shows that the pure-text-based GPT-4o failed to select a key figure. While the answer is correct, it is more verbose compared to that of multimodal-based GPT-4o.

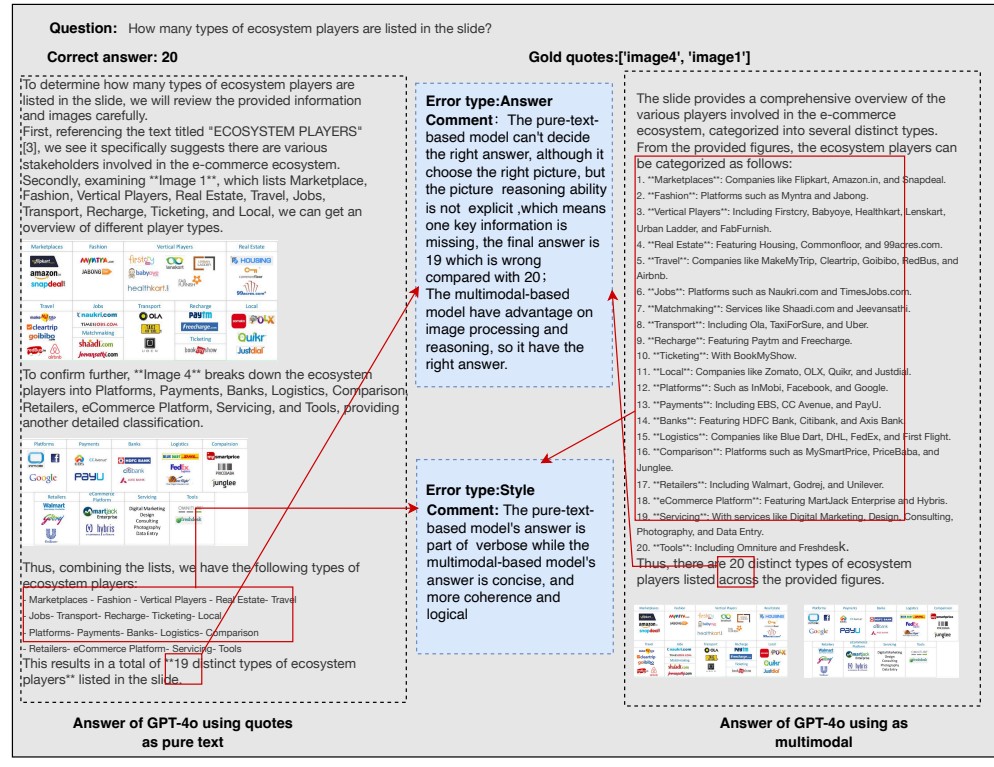

Figure 28: This example shows that although the pure-text-based GPT-4o selected the correct image, its multimodal reasoning was incorrect and not concise, resulting in an incoherent and verbose answer.

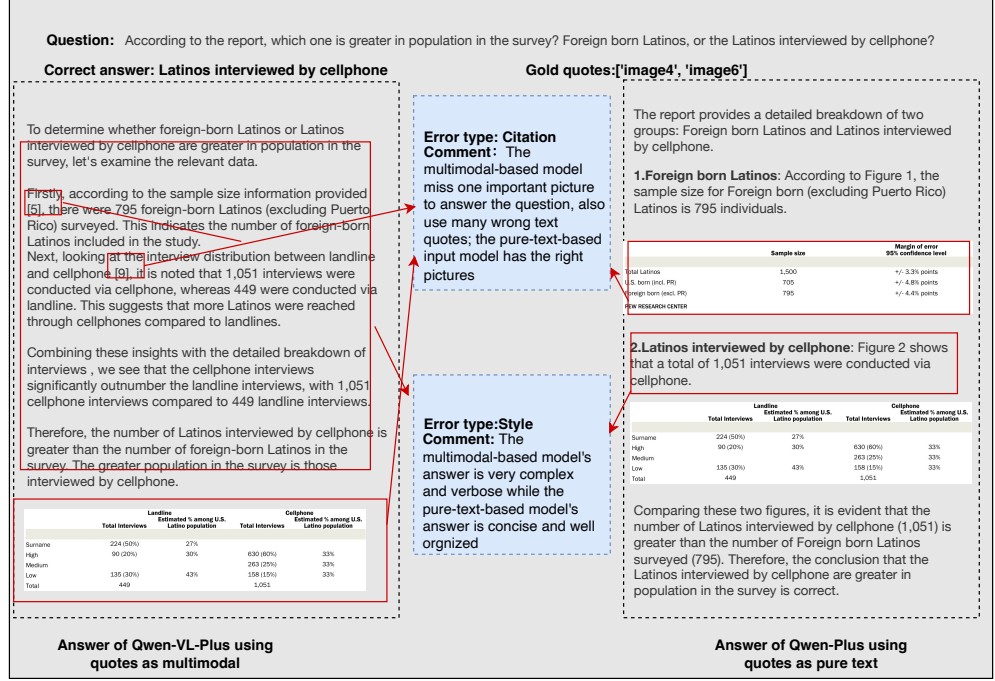

Figure 29: The examples shows that in Qwen-VL-Plus failed to one key image evidence. In contrast, Qwen-Plus which relies on pure-text inputs, correctly selected the evidence and led to correct answer.

### F.3 Finetuning Effectiveness: Qualitative Analysis

As discussed in Section 4.3 and illustrated in Figure 6, fine-tuning significantly enhances the model's ability to select and generate multimodal information. To further investigate this effect, we conduct a qualitative analysis of Qwen2.5-14B-Instruct [65] before and after fine-tuning, manually reviewing 100 cases to assess performance changes.

Our analysis reveals substantial improvements across multiple evaluation dimensions. Fine-tuning markedly strengthens the model's citation capabilities for both textual and visual evidence. Prior to fine-tuning, the model frequently selected incorrect images or failed to present relevant visual information. After fine-tuning, it consistently select images that closely align with gold-standard answers. For text citation, the base model often chose irrelevant passages or produced redundant references, whereas the fine-tuned model reliably identified appropriate textual segments, resulting in more accurate and relevant support.

Furthermore, the overall answer quality improves, with fine-tuned responses exhibiting higher factual accuracy and stronger reasoning consistency, which primarily due to improved evidence selection. The logical integration and positioning of cited images also become more coherent. Additionally, the fine-tuned model generates answers that are more concise, explicit, and faithful to the ground truth, demonstrating increased clarity, relevance, and structured reasoning.

In summary, these findings underscore that fine-tuning greatly improves citation precision, factual grounding, logical coherence, and answer fluency, leading to comprehensive performance gains on multimodal RAG tasks.

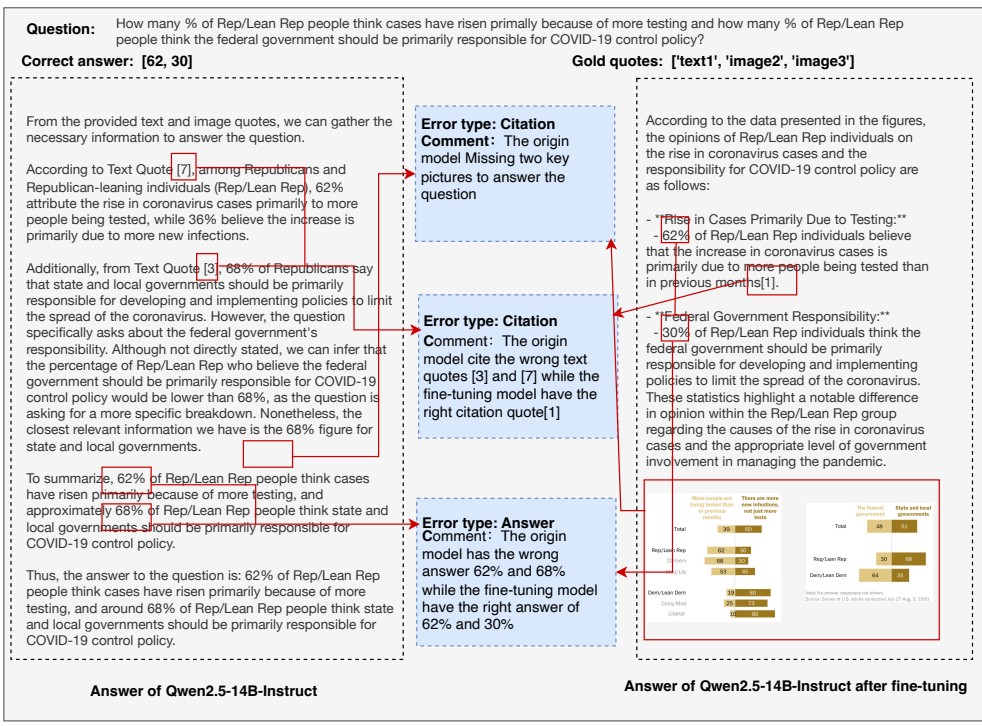

Figure 30: This example demonstrates that the base model failed to cite two key images and referenced incorrect textual passages, resulting in incorrect answer. In contrast, the fine-tuned model successfully cited the relevant images and text, leading to a correct and well-supported response.

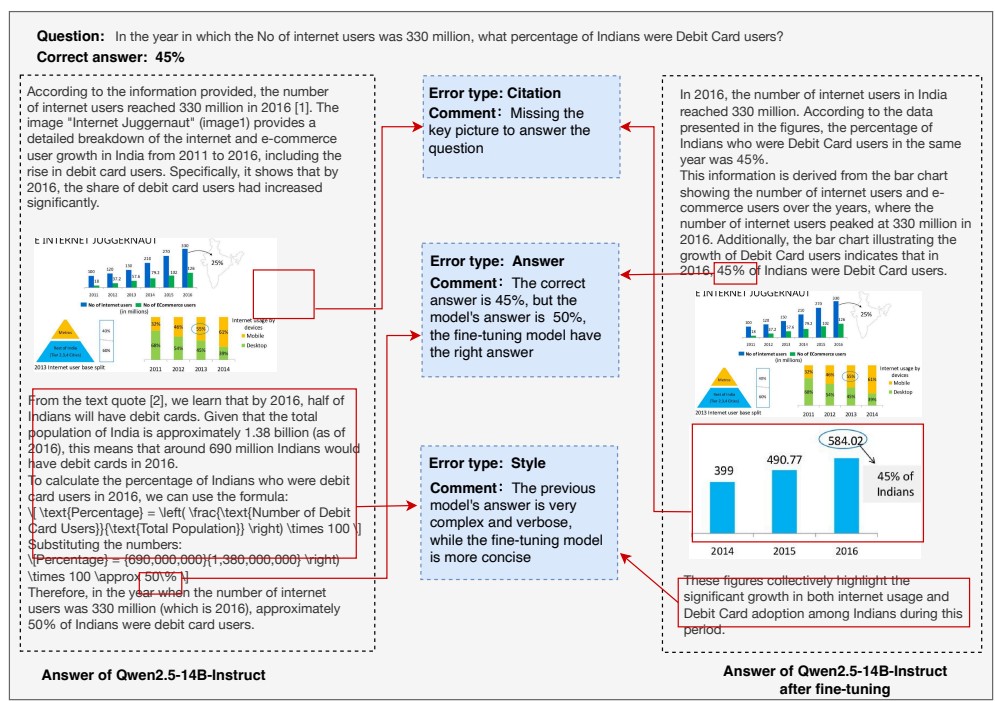

Figure 31: This example demonstrates that the base model failed to cite the key image and produced an overly verbose and lengthy reasoning chain, resulting in an incorrect answer. In contrast, the fine-tuned model successfully cited the relevant image and provided a more concise reasoning process, leading to a correct response.

## G    License Agreement

`MMDocRAG` reuses document data and select annotations from the MMDocIR dataset [19], which is distributed under the terms of the Apache License 2.0. The Apache License 2.0 permits use, reproduction, and distribution for research purposes, provided that compliance with its terms is maintained. For the new annotations contributed in this work, including but not limited to the questions, evidence annotations, and multimodal answers, we make them available solely for research purposes. Users are permitted to use, modify, and share these annotations for academic and non-commercial research activities. Any other use, including commercial exploitation, is not permitted without explicit written permission from the authors.

## H    Limitations

- **Limited Multimodal Retrieval Capability**: `MMDocRAG` primarily focuses on multimodal answer generation by evaluating the ability of LLMs and VLMs to select and integrate relevant multimodal quotes from a noisy set. In our proposed `MMDocRAG`, each question is associated with an average of 2.7 gold quotes out of 15/20 candidates, resulting in only 18.0/13.5% relevant quotes for answer generation. However, real-world RAG scenarios might present cases where no relevant quotes are retrieved, especially with imperfect multimodal retrieval modules. Our current setup does not adequately capture such rare unanswerable cases.

- **Fine-tuning Limited to LLMs**: Our experiments fine-tune five Qwen2.5 models (ranging from 3B to 72B parameters) where multimodal quotes are linearized as pure text sequences. We do not report fine-tuning results on Qwen2.5-VL models due to computational constraints: multimodal input sequences are significantly longer (7.1k vs.3.6k tokens on average), and, in extreme cases with high-resolution images, can exceed 20k tokens. While we attempted to fine-tune Qwen2.5-VL

models, training was unstable and frequently interrupted. As a result, we do not present VLM fine-tuning results in this work.

# I   Societal Impacts

The introduction and broader adoption of `MMDocRAG` may have potential societal impacts spanning both positive and negative dimensions. Below, we outline possible negative consequences and discuss potential mitigation strategies:

**Privacy Risks:** `MMDocRAG` enables models to process and reason over lengthy, multimodal documents, which may include sensitive personal, financial, or health information. There is a risk that such technologies could be leveraged for large-scale automated surveillance, unauthorized extraction of personal data, or other privacy violations.

**Fairness and Bias:** If benchmarked models are trained or evaluated on data that does not reflect diverse demographic, linguistic, and organizational backgrounds, outputs may exhibit biases. This may lead to unfair decision-making or perpetuating harmful stereotypes.

**Content Safety:** Given that the benchmark includes real-world documents with complex multimodal content, there is a risk that generative models might output inappropriate, offensive, or harmful content if such material is present in source documents.

**Mitigation Strategies:** To mitigate these risks, we recommend that: (i) Benchmark development uses only publicly available, carefully vetted datasets, with sensitive information anonymized or removed; (ii) Model outputs are monitored for bias and fairness, and evaluation includes checks for disparate impact; (iii) Access to fine-tuned models and annotations is controlled, and usage is governed by strict ethical guidelines and data use agreements; (iv) Content moderation tools are integrated where appropriate, especially for public-facing applications.

We encourage researchers and practitioners employing `MMDocRAG` to be mindful of these factors and to actively work toward responsible development and deployment, including transparency about limitations and proactive safeguards where needed. We welcome community feedback and collaboration on best practices to further reduce risks as this technology evolves.

