# OpenReview forum: "Benchmarking Retrieval-Augmented Multimomal Generation for Document Question Answering"
_NeurIPS.cc/2025/Datasets_and_Benchmarks_Track — NeurIPS 2025 Datasets and Benchmarks Track poster_

### Official Review · Reviewer_ddPK · 2025-07-01

**Rating:** 5
**Confidence:** 4

**Summary:**

This paper introduces a comprehensive benchmark, MMDocRAG, for multimodal document question answering. It includes 4,055 expert-annotated QA pairs with multi-page and cross-modal evidence chains. Among them, 821 QA pairs are selected fromthe  existing dataset, and the remaining are generated by VLMs. The authors conduct extensive experiments, evaluating the performance of over 60 existing open-source, closed-source, and fine-tuned large models on the MMDocRAG dataset, and find that fine-tuning can significantly improve the model performance.

**Dataset Code Accessibility:**

Yes

**Ethical Considerations:**

No, there are no or only very minor ethics concerns

**Final Justification:**

The authors have carefully addressed my concerns, and thus I have raised my rating.

**Limitations Weaknesses:**

- Most QA pairs in the dataset are autonomously generated by VLMs. Consequently, such large-scale synthetic data inevitably exhibits stylistic biases and artifacts introduced by generative models, making it challenging to represent the true distribution of real user queries.

- A more detailed experimental analysis is necessary. The proposed MMDocRAG includes eight distinct question types; therefore, it is critical to investigate whether existing models exhibit significant performance variations across these types. Furthermore, are there noticeable differences in performance between cross-page versus single-page questions, multi-image versus single-image questions, or cross-modal versus single-modal questions?

- Human evaluations are needed, rather than just employing LLM as a judge.

- The dataset appears to contain only English-language content.

- There are several writing errors, such as "LVM."

**Strengths Contributions:**

- The proposed MMDocRAG is a comprehensive multimodal document question answering benchmark. Multimodal evidences, including both text and image quotes, are annotated.

- Extensive evaluations are conducted. The experiments cover 6 text, 4 visual, and 4 hybrid retrievers; 37 open-source models, 23 proprietary models, and 5 finetuned models.

- The paper is well-written and clearly organized.

---

> ### Author Rebuttal · Authors · 2025-07-30
>
> Dear Reviewer ddPK,
>
> We sincerely thank you for your thoughtful review and valuable feedback on our work. Below, we address each of your concerns with additional experimental evidence and clarifications.
>
>
>
> ## **1. Data Distribution**
>
> Thank you for this valuable feedback. We acknowledge that synthetic data may exhibit stylistic biases and artifacts from generative models. To address this concern, we implemented a comprehensive multi-stage pipeline involving manual selection, filtering, refinement, and validation to calibrate the synthetic data and ensure quality. While we recognize that synthetic data limitations cannot be entirely eliminated, our annotation process incorporates substantial human oversight to minimize these biases and better approximate real user query distributions.
>
>
>
> ## **2. Fine-grained Performance Analysis**
>
> Thank you for this valuable feedback. We provide a comprehensive breakdown of model performance across the requested dimensions. While Figure 8 shows domain-specific results via radar charts, rebuttal constraints prevent new images, so we present detailed tabular analysis (for multimodal quote selection F1) below.
>
> ### **2.1 Single vs Multi-Page Performance**
>
> | Model                 | Multimodal Quotes |             | Pure-text Quotes |             |
> | --------------------- | ----------------- | ----------- | ---------------- | ----------- |
> |                       | Single            | Multi (Δ)   | Single           | Multi (Δ)   |
> | **GPT-4.1**           | 74.7              | 66.4 (-8.3) | 72.9             | 64.3 (-8.6) |
> | **Gemini-2.5-Pro**    | 66.8              | 64.2 (-2.6) | 65.0             | 65.2 (+0.2) |
> | **Claude-3.5-Sonnet** | 62.3              | 62.7 (+0.4) | 54.1             | 60.2 (+6.1) |
> | **Llama-4-Maverick**  | 63.7              | 54.2 (-9.5) | 59.3             | 57.5 (-1.8) |
>
> **Key findings**: Multi-page questions pose significant challenges for most models, with GPT-4.1 and Llama-4-Maverick showing substantial performance drops (-8.3 to -9.5 F1). Claude-3.5-Sonnet demonstrates resilience, particularly improving on pure-text multi-page tasks (+6.1 F1).
>
> ### **2.2 Single vs Multi-Image Performance**
>
> | Model                 | Multimodal Quotes |             | Pure-text Quotes |             |
> | --------------------- | ----------------- | ----------- | ---------------- | ----------- |
> |                       | Single            | Multi (Δ)   | Single           | Multi (Δ)   |
> | **GPT-4.1**           | 73.0              | 66.8 (-6.2) | 70.8             | 65.2 (-5.6) |
> | **Gemini-2.5-Pro**    | 67.0              | 63.5 (-3.5) | 66.1             | 63.8 (-2.3) |
> | **Claude-3.5-Sonnet** | 63.3              | 61.6 (-1.7) | 56.6             | 58.4 (+1.8) |
> | **Llama-4-Maverick**  | 60.3              | 56.5 (-3.8) | 59.4             | 57.0 (-2.4) |
>
> **Key findings**: Multi-image processing consistently degrades performance across models, with GPT-4.1 showing the largest drops (-6.2 F1). Claude-3.5-Sonnet exhibits the most stable performance and even improves on pure-text multi-image tasks.
>
> ### **2.3 Single vs Cross-Modal Performance**
>
> | Model                 | Multimodal Quotes |             | Pure-text Quotes |              |
> | --------------------- | ----------------- | ----------- | ---------------- | ------------ |
> |                       | Single            | Cross (Δ)   | Single           | Cross (Δ)    |
> | **GPT-4.1**           | 74.1              | 68.1 (-6.0) | 73.4             | 65.4 (-8.0)  |
> | **Gemini-2.5-Pro**    | 61.1              | 67.9 (+6.8) | 58.9             | 68.6 (+9.7)  |
> | **Claude-3.5-Sonnet** | 59.3              | 64.3 (+5.0) | 48.9             | 62.2 (+13.3) |
> | **Llama-4-Maverick**  | 64.3              | 55.4 (-8.9) | 54.9             | 60.3 (+5.4)  |
>
> **Key findings**: Cross-modal questions reveal divergent model behaviors. While GPT-4.1 and Llama-4-Maverick struggle with cross-modal integration, Gemini-2.5-Pro and Claude-3.5-Sonnet show substantial improvements, suggesting better cross-modal reasoning capabilities.
>
> ### **2.4 Performance by Question Type**
>
> | Model                 | Descriptive | Comparative | Analytical | Interpretative | Inferential | Others | σ    |
> | --------------------- | ----------- | ----------- | ---------- | -------------- | ----------- | ------ | ---- |
> | **Multimodal Quotes** |             |             |            |                |             |        |      |
> | GPT-4.1               | 71.0        | 69.3        | 68.4       | 72.0           | 72.6        | 67.6   | 1.96 |
> | Gemini-2.5-Pro        | 64.8        | 65.4        | 64.9       | 67.4           | 64.1        | 64.8   | 1.19 |
> | Claude-3.5-Sonnet     | 62.3        | 62.8        | 62.2       | 63.5           | 59.7        | 59.2   | 1.66 |
> | Llama-4-Maverick      | 58.2        | 57.8        | 57.5       | 61.2           | 62.2        | 61.3   | 2.09 |
> | **Pure-text Quotes**  |             |             |            |                |             |        |      |
> | GPT-4.1               | 68.5        | 67.6        | 65.4       | 71.5           | 69.2        | 66.4   | 2.07 |
> | Gemini-2.5-Pro        | 62.0        | 66.8        | 64.7       | 69.3           | 59.8        | 62.0   | 3.42 |
> | Claude-3.5-Sonnet     | 55.1        | 58.8        | 56.9       | 59.7           | 55.2        | 58.0   | 1.86 |
> | Llama-4-Maverick      | 56.5        | 58.9        | 57.9       | 61.4           | 57.1        | 59.7   | 1.76 |
>
> **Key findings**: Question type analysis reveals varying model strengths. Gemini-2.5-Pro shows the most consistent performance (lowest σ=1.19 for multimodal), while GPT-4.1 excels at interpretative and inferential questions. All models perform comparably across question types, with standard deviations below 3.5, indicating robust generalization.
>
> ### **2.5 Summary**
>
> This analysis reveals three key patterns: (1) Multi-page and multi-image questions consistently challenge most models, (2) Cross-modal reasoning capabilities vary significantly between models, with newer architectures showing advantages, and (3) Performance across question types remains relatively stable, suggesting good generalization across reasoning tasks.
>
>
>
> ## **3. Human Evaluation**
>
> We appreciate this important concern. While conducting manual evaluation across all models and 2,000 questions would be resource-prohibitive, we performed comprehensive manual error analysis on a representative sample of 200 questions to validate our LLM-judge methodology (detailed in Appendix F.1).
>
> **3.1 Manual Error Analysis Results**
>
> | Model             | Excessive Citation | Inadequate Citation | Irrelevant Cite Position | Reason/Fact Inconsistency | Text/Image Incoherence |
> | ----------------- | ------------------ | ------------------- | ------------------------ | ------------------------- | ---------------------- |
> | GPT-4o            | 34.5%              | 30.0%               | 16.5%                    | 17.7%                     | 16.3%                  |
> | Claude-3.5-Sonnet | 36.9%              | 26.2%               | 20.7%                    | 18.5%                     | 15.5%                  |
> | Gemini-2.5-Pro    | 32.3%              | 18.6%               | 17.4%                    | 13.8%                     | 12.5%                  |
>
> **3.2 Corresponding LLM-Judge Scores:**
>
> | Model             | Selection F1 | Cite Quality | Reason Logic | Factuality | Text/Image Coherence |
> | ----------------- | ------------ | ------------ | ------------ | ---------- | -------------------- |
> | GPT-4o            | 62.6         | 3.42         | 3.58         | 3.58       | 3.74                 |
> | Claude-3.5-Sonnet | 62.5         | 3.22         | 3.54         | 3.53       | 3.71                 |
> | Gemini-2.5-Pro    | 65.4         | 3.40         | 3.78         | 3.94       | 3.97                 |
>
> *Quotes Selection F1 is computed automatically via heuristic regex extraction (Section 4.1).*
>
> **3.3 Key Validation Findings:**
>
> - **Strong manual-LLM alignment:** Models with higher manual error rates receive correspondingly lower LLM-judge scores (e.g., Claude-3.5-Sonnet: 36.9% citation errors → 3.22 citation quality)
> - **Consistent ranking:** Gemini-2.5-Pro shows lowest error rates and highest LLM-judge scores across dimensions
> - **Automatic metric correlation:** Selection F1 correlates well with LLM-judge citation scores, validating our evaluation framework
>
> This concordance supports the reliability of our LLM-judge methodology for the full dataset.
>
>
>
> ## **4. Multilingual Setting**
>
> Thank you for this valuable feedback. We acknowledge that our benchmark is limited to English-language content. Given resource constraints, we strategically focused on addressing the research gap in multimodal RAG for English to ensure comprehensive coverage within this scope. Extending the benchmark to multilingual settings represents an important direction for future work that would significantly broaden the applicability of multimodal RAG.
>
>
>
> ## **5. Typos**
>
> Thank you for the reminder on typos. We will fix them in our revision to ensure good presentation of our work.
>
>
>
>
>
> We hope these responses address your concerns comprehensively. Feel free to let us know if you have any additional questions, and we are more than happy to provide further clarification on any aspect of our work.
>
> Best regards,
>
> The Authors

---

> > ### Author Response · Authors · 2025-08-05
> >
> > Dear Reviewer ddPK,
> >
> > Thank you for your initial feedback on our manuscript. We have submitted a comprehensive response with extensive additional experiments to address the concerns you raised.
> >
> > As we approach the end of the discussion period, we would like to check if our responses and new experimental results have adequately addressed your questions. If our clarifications have resolved your concerns, we would be grateful if you could by any chance, consider adjusting your evaluation accordingly.
> >
> > Should any points require further clarification or additional experiments, we remain ready to provide prompt responses.
> >
> > Thank you for your time and consideration.
> > Best regards,
> > Authors

---

### Official Review · Reviewer_Cb3Y · 2025-07-02

**Rating:** 5
**Confidence:** 3

**Summary:**

This paper introduces MMDocRAG, a novel benchmark for multimodal Document Visual Question Answering (DocVQA) featuring 4,055 expert-annotated QA pairs with multi-page, cross-modal evidence chains, addressing current limitations in evaluating multimodal evidence integration. The authors propose innovative metrics for assessing multimodal quote selection and introduce a framework enabling answers that combine text with relevant visual elements, demonstrating through extensive experiments with 60+ models that proprietary LVMs outperform open-source alternatives in handling multimodal inputs. The benchmark provides rigorous evaluation standards and actionable insights for advancing multimodal document understanding systems, particularly highlighting the performance benefits of fine-tuned LLMs using detailed image descriptions.

**Dataset Code Accessibility:**

Yes

**Ethical Considerations:**

No, there are no or only very minor ethics concerns

**Final Justification:**

My concerns have been addressed, so I will keep the "Accept" score.

**Limitations Weaknesses:**

The reviewers suggest that the authors summarize the key challenges of this benchmark and provide insights for future researchers working on this task.

**Strengths Contributions:**

This paper makes three key contributions: (1) introduces MMDocRAG, the first comprehensive benchmark for multimodal DocVQA with 4,055 expert-annotated QA pairs and multi-page evidence chains; (2) develops novel evaluation metrics for multimodal quote selection and visual-textual answer generation; (3) provides extensive empirical analysis of 60+ models, revealing performance gaps between proprietary/open-source systems and benefits of fine-tuning with image descriptions. The benchmark enables rigorous evaluation of cross-modal document understanding.

---

> ### Author Rebuttal · Authors · 2025-07-30
>
> Dear Reviewer Cb3Y,
>
> We sincerely thank you for your thoughtful review and valuable feedback on our work. Below, we address each of your concerns with additional experimental evidence and clarifications.
>
>
>
> ## 1. **Key Challenges and Future Insights**
>
> We appreciate your suggestion to summarize the key challenges of this benchmark and provide insights for future researchers. Based on our extensive evaluation of 60+ models and additional experiments, we identify the following key challenges that MMDocRAG presents:
>
> ### **1.1 Core Challenges:**
>
> - **End-to-End RAG Pipeline Complexity**: Our end-to-end RAG evaluation reveals that imperfect retrieval significantly impacts answer quality, creating a cascading effect where retrieval errors compound reasoning difficulties
> - **Cross-modal Evidence Integration**: Successfully combining textual and visual information across multiple document pages to form coherent evidence chains
> - **Multi-page and Multi-image Reasoning**: Our fine-grained performance analysis shows substantial difficulty in maintaining context across lengthy documents with multiple images while identifying relevant evidence spans
> - **Multimodal Quote Selection**: Accurately extracting and presenting both textual quotes and relevant visual elements as supporting evidence, particularly challenging when evidence spans multiple modalities
>
> ### **1.2 Insights for Future Research:**
>
> - **End-to-End System Design**: Our end-to-end RAG experiments highlight the critical need for robust retrieval mechanisms, as retrieval quality creates a performance bottleneck for the entire pipeline
>
> - **VLM Fine-tuning Strategies**: Our fine-tuning experiments on VLMs demonstrate promising directions for improving multimodal document understanding, particularly for cross-modal reasoning tasks
>
> - **Performance Scaling Patterns**: Fine-grained analysis reveals that performance degrades significantly with increased document complexity (multi-page, multi-image scenarios), suggesting specific architectural improvements needed for handling complex documents
>
> - **Cross-modal Performance Gaps**: Our analysis identifies specific cross-modal reasoning patterns where current models struggle, providing targeted directions for future model development
>
> - **Generative Multimodal Answering**: Beyond selecting and quoting existing document images, future methods could advance toward editing or creating new images, charts, or visualizations as part of multimodal answers, enabling more sophisticated and tailored responses
>
>
>
> **Moreover, we provide additional experiments and analysis related to aforementioned key challenges and future insights.**
>
> ## **2. Additional Experiments: End-to-end RAG**
>
> To address this limitation and simulate more realistic RAG environments, we conducted comprehensive end-to-end experiments that evaluate system performance under imperfect retrieval conditions.
>
> ### **2.1 Experimental Setup**
>
>
> We tested different retrieval configurations to assess multimodal generation quality when retrieval is imperfect:
>
> - **Perfect retriever** (baseline): All gold quotes included with noisy quotes
> - **Imperfect retrievers**:
>   - Single BGE retriever with original questions
>   - Single BGE retriever with expanded multi-clause queries
>   - Multiple retrievers (BGE, Qwen3-0.6B, BM25, E5) with expanded queries
>   - Reranking applied using Qwen3-0.6B-reranker (top 8 image + 12 text quotes)
>
>
>
> ### **2.2 Experimental Results**
>
> |Model|Retriever|Query|Txt Rec.|Img Rec.|All Rec.|Txt Sel.|Img Sel.|All Sel.|Bleu|RougeL|LLM judge: Avg |
> |-|--|-|-|-|-|-|-|-|-|--|- |
> |GPT-4.1|perfect|-|-|-|-|52.0|80.7|70.2|0.157|0.313|4.14 |
> |GPT-4.1|multiple|clauses|49.5|86.8|84.9|41.4|65.6|59.9|0.141|0.303|3.79 |
> |GPT-4.1|BGE|clauses|42.1|83.6|78.9|37.9|64.0|57.5|0.141|0.302|3.71 |
> |GPT-4.1|BGE|original|34.6|77.8|71.0|34.2|60.8|54.4|0.137|0.299|3.53 |
> |Gemini2.5-Flash|perfect|-|-|-|-|46.1|76.2|62.4|0.139|0.284|3.76 |
> |Gemini2.5-Flash|multiple|clauses|49.5|86.8|84.9|34.3|60.3|51.8|0.124|0.281|3.42 |
> |Gemini2.5-Flash|BGE|clauses|42.1|83.6|78.9|30.9|59.2|50.4|0.125|0.281|3.39 |
> |Gemini2.5-Flash|BGE|original|34.6|77.8|71.0|27.5|55.6|47.7|0.124|0.280|3.21 |
>
> *Note: (i) Rec. is for Recall (ii) Sel. is for selection F1 (iii) Avg for the average of 5 metrics of LLM-Judge*
>
> ### **2.3 Key Findings**
>
> 1. **Clear correlation between retrieval quality and downstream performance**: Results demonstrate that higher retrieval recall consistently leads to improved quote selection accuracy and answer generation quality.
>
> 2. **Realistic performance degradation**: Even state-of-the-art models (GPT4 and Gemini2.5) show approximately 10% performance degradation when using imperfect retrievers compared to the perfect retriever baseline.
>
> 3. **Importance of retrieval strategy**: Multi-retriever approaches with query expansion achieve significantly better recall (84.9%) compared to naive single-retriever methods (71.0%), demonstrating the importance of retrieval strategy in real-world scenarios.
>
> These experiments validate that our benchmark captures the challenges of end-to-end RAG systems, extending our benchmark capable of measuring both retrieval and generation components for practical multimodal RAG applications.
>
>
>
> ## **3. Additional Experiments: VLM Fine-tuning**
>
> Our submmitted version contains only fine-tuning on LLMs. To bridge the gap, we conducted additional experiments fine-tuning Qwen2.5-VL models (3B and 7B), InternVL3 models (2B, 8B, and 9B) on our multimodal RAG task using interleaved multimodal inputs.
>
> ### **3.1 Experimental Results**
>
>
> |Model|Txt Sel.|Img Sel.|All Sel.|Bleu|RougeL|LLM Judge: Avg |
> |-|-|-|-|-|--|- |
> |Qwen2.5-VL-3B|5.3|1.0|1.2|0.105|0.283|2.25 |
> |- After fine-tuning|9.8|68.6|55.5|0.186|0.341|3.36 |
> |Qwen2.5-VL-7B|16.3|23.2|16.6|0.069|0.273|2.47 |
> |- After fine-tuning|16.5|72.4|58.6|0.199|0.355|3.64 |
> |InternVL3-2B|10.7|9.5|6.8|0.097|0.250|2.10 |
> |- After fine-tuning|8.9|63.6|51.4|0.180|0.333|3.15 |
> |InternVL3-8B|34.4|40.7|37.0|0.119|0.260|2.98 |
> |- After fine-tuning|30.4|71.7|58.8|0.205|0.356|3.69 |
> |InternVL3-9B|29.3|60.8|50.9|0.146|0.303|3.12 |
> |- After fine-tuning|28.4|72.9|60.3|0.210|0.362|3.87 |
>
> ### **3.2 Key Findings**
>
> 1. **Dramatic improvement in quote selection**: Fine-tuning helps models much better at selecting relevant quotes, especially images. The Qwen2.5-VL-3B model's performance jumps from 1.2 to 55.5, InternVL3-2B model improves from 6.8 to 51.4, and the Qwen2.5-VL-7B model improves from 16.6 to 58.6. InternVL3-8B and 9B model also experience substantial improvements.
>
> 2. **Enhanced answer quality across all metrics**: Fine-tuning consistently improves BLEU scores, RougeL scores, and LLM judge ratings.
>
> 3. **Successful adaptation of VLMs to multimodal RAG**: These results show that VLMs can effectively learn multimodal RAG tasks when properly trained. However, inferring and training VLMs typically incur significant more resources due to prolonged input sequence length (QwenVL: 3.6k vs. 8.7k, InternVL: 3.6K vs. 17.2k)
>
> The dramatic improvements after fine-tuning validate both our high-quality training dataset and comprehensive benchmark design.
>
>
>
> ## **4. Additional Analysis: Fine-grained Performance**
>
> We provide a comprehensive breakdown of model performance accross different evidence types. While Figure 8 shows domain-specific results via radar charts, rebuttal constraints prevent new images, so we present detailed tabular analysis (for multimodal quote selection F1) below.
>
> ### **4.1 Single vs Multi-Page Performance**
>
> |Model|Multimodal Quotes||Pure-text Quotes||
> |--|--|-|-|- |
> ||Single|Multi (Δ)|Single|Multi (Δ) |
> |**GPT-4.1**|74.7|66.4 (-8.3)|72.9|64.3 (-8.6) |
> |**Gemini-2.5-Pro**|66.8|64.2 (-2.6)|65.0|65.2 (+0.2) |
> |**Claude-3.5-Sonnet**|62.3|62.7 (+0.4)|54.1|60.2 (+6.1) |
> |**Llama-4-Maverick**|63.7|54.2 (-9.5)|59.3|57.5 (-1.8) |
>
> **Key findings**: Multi-page questions pose significant challenges for most models, with GPT-4.1 and Llama-4-Maverick showing substantial performance drops (-8.3 to -9.5 F1). Claude-3.5-Sonnet demonstrates resilience, particularly improving on pure-text multi-page tasks (+6.1 F1).
>
> ### **4.2 Single vs Multi-Image Performance**
>
> |Model|Multimodal Quotes||Pure-text Quotes||
> |--|--|-|-|- |
> ||Single|Multi (Δ)|Single|Multi (Δ) |
> |**GPT-4.1**|73.0|66.8 (-6.2)|70.8|65.2 (-5.6) |
> |**Gemini-2.5-Pro**|67.0|63.5 (-3.5)|66.1|63.8 (-2.3) |
> |**Claude-3.5-Sonnet**|63.3|61.6 (-1.7)|56.6|58.4 (+1.8) |
> |**Llama-4-Maverick**|60.3|56.5 (-3.8)|59.4|57.0 (-2.4) |
>
> **Key findings**: Multi-image processing consistently degrades performance across models, with GPT-4.1 showing the largest drops (-6.2 F1). Claude-3.5-Sonnet exhibits the most stable performance and even improves on pure-text multi-image tasks.
>
> ### **4.3 Single vs Cross-Modal Performance**
>
> |Model|Multimodal Quotes||Pure-text Quotes ||
> |--|--|-|-|- |
> ||Single|Cross (Δ)|Single|Cross (Δ)|
> |**GPT-4.1**|74.1|68.1 (-6.0)|73.4|65.4 (-8.0)|
> |**Gemini-2.5-Pro**|61.1|67.9 (+6.8)|58.9|68.6 (+9.7)|
> |**Claude-3.5-Sonnet**|59.3|64.3 (+5.0)|48.9|62.2 (+13.3) |
> |**Llama-4-Maverick**|64.3|55.4 (-8.9)|54.9|60.3 (+5.4)|
>
> **Key findings**: Cross-modal questions reveal divergent model behaviors. While GPT-4.1 and Llama-4-Maverick struggle with cross-modal integration, Gemini-2.5-Pro and Claude-3.5-Sonnet show substantial improvements, suggesting better cross-modal reasoning capabilities.
>
>
>
> We hope these responses address your concerns comprehensively. Feel free to let us know if you have any additional questions, and we are more than happy to provide further clarification on any aspect of our work.
>
> Best regards,
>
> The Authors

---

> > ### Comment · Reviewer_Cb3Y · 2025-08-04
> > **Thanks for author's rebuttals**
> >
> > Thanks for your responses. I think my initial score is appropriate, and I will maintain it

---

> > > ### Author Response · Authors · 2025-08-05
> > >
> > > Dear Reviewer Cb3Y,
> > >
> > > We sincerely thank you for your constructive review and your feedback is invaluable in helping us address the key issues and improve the quality of our work. We appreciate the time and effort you invested in this review process.
> > >
> > > Best regards,
> > > The Authors

---

### Official Review · Reviewer_GErz · 2025-07-02

**Rating:** 4
**Confidence:** 4

**Summary:**

This paper introduces MMDocRAG, a comprehensive benchmark aimed at evaluating retrieval-augmented, multimodal answer generation for document question answering (DocVQA). The proposed benchmark addresses two major challenges in real-world multimodal document QA: (1) integrating information from lengthy, complex documents containing both text and images (including tables and figures), and (2) assessing a model’s capabilities to identify, select, and coherently integrate supporting evidence from both textual and visual modalities. Extensive experiments on 60 large models (including 33 VLMs and 27 LLMs, both open-source and proprietary) and 14 different retrieval systems demonstrate the difficulties current models face in multimodal evidence identification and integration, especially under noisy conditions.

**Dataset Code Accessibility:**

No

**Ethical Considerations:**

No, there are no or only very minor ethics concerns

**Final Justification:**

The rebuttal fully addressed my concern about calibration of ground truth and task design. Only the response on end-to-end RAG framework didn't completely convince me. So overall, I give a borderline accept.

**Limitations Weaknesses:**

- While the benchmark primarily evaluates multimodal answer generation given a fixed set of quotes (evidence snippets), the retrieval stage itself is not the main focus, and limitations remain in modeling the true end-to-end RAG scenario where retrieval errors can lead to unanswerable cases. Real-world applications may thus find the fixed-quote setting somewhat idealized.
- Fine-tuning is only reported for LLMs using pure-text representations rather than for VLMs using interleaved multimodal input.
- The annotations and evaluations are highly detailed but the task design primarily focuses on answer generation with evidence selection. Other aspects of document understanding (e.g., open-ended generation, summarization, or conversational multihop QA) are not directly covered and may require additional benchmarks.
- How did the author calibrate the ground truth, especially the quotes? I think it is difficult to get the ground truth completely accurate for some questions involving multiple pages.
- Missing some recent paper [1,2].

[1] ViDoRAG: Visual Document Retrieval-Augmented Generation via Dynamic Iterative Reasoning Agents

[2] MDocAgent: A Multi-Modal Multi-Agent Framework for Document Understanding

**Strengths Contributions:**

- MMDocRAG fills a key gap by enabling systematic evaluation of multimodal retrieval-augmented generation on realistic, noisy, long-form document scenarios.
- The benchmark provides detailed annotations at multiple levels—page, quote, evidence type (text, table, figure), and gold/noisy distractors—and introduces practical evaluation metrics for multimodal quote selection, surface similarity, and answer quality from various aspects.
- The authors perform large-scale experiments across a wide range of models (60 LLMs/VLMs), fine-tuned and off-the-shelf, as well as multiple retrievers and input paradigms (pure-text vs interleaved).

---

> ### Author Rebuttal · Authors · 2025-07-30
>
> Dear Reviewer GErz,
>
> We sincerely thank you for your thoughtful review and valuable feedback on our work. Below, we address each of your concerns with additional experimental evidence and clarifications.
>
>
>
> ## **1. End-to-end RAG Experimental Results**
>
> We acknowledge the concern about the idealized nature of fixed-quote settings. To address this limitation and simulate more realistic RAG environments, we conducted comprehensive end-to-end experiments that evaluate system performance under imperfect retrieval conditions.
>
> ### **1.1 Experimental Setup**
>
>
> We tested different retrieval configurations to assess multimodal generation quality when retrieval is imperfect:
>
> - **Perfect retriever** (baseline): All gold quotes included with noisy quotes (8 image + 12 text quotes)
> - **Imperfect retrievers**:
>   - Single BGE retriever with original questions
>   - Single BGE retriever with expanded multi-clause queries
>   - Multiple retrievers (BGE, Qwen3-0.6B, BM25, E5) with expanded queries
>   - Reranking applied using Qwen3-0.6B-reranker (top 8 image + 12 text quotes)
>
> *Note: We focused on single-vector embedding retrievers (BGE, Qwen3-0.6B, E5) for compatibility with vector databases (Milvus DB), excluding multi-vector token-level embeddings (e.g., ColBERT, ColQwen).*
>
> ### **1.2 Experimental Results**
>
> | Model           | Retriever | Query    | Txt Rec. | Img Rec. | All Rec. | Txt Sel. | Img Sel. | All Sel. | Bleu  | RougeL | LLM judge: Avg |
> | --------------- | --------- | -------- | -------- | -------- | -------- | -------- | -------- | -------- | ----- | ------ | -------------- |
> | GPT-4.1         | perfect   | -        | -        | -        | -        | 52.0     | 80.7     | 70.2     | 0.157 | 0.313  | 4.14           |
> | GPT-4.1         | multiple  | clauses  | 49.5     | 86.8     | 84.9     | 41.4     | 65.6     | 59.9     | 0.141 | 0.303  | 3.79           |
> | GPT-4.1         | BGE       | clauses  | 42.1     | 83.6     | 78.9     | 37.9     | 64.0     | 57.5     | 0.141 | 0.302  | 3.71           |
> | GPT-4.1         | BGE       | original | 34.6     | 77.8     | 71.0     | 34.2     | 60.8     | 54.4     | 0.137 | 0.299  | 3.53           |
> | Gemini2.5-Flash | perfect   | -        | -        | -        | -        | 46.1     | 76.2     | 62.4     | 0.139 | 0.284  | 3.76           |
> | Gemini2.5-Flash | multiple  | clauses  | 49.5     | 86.8     | 84.9     | 34.3     | 60.3     | 51.8     | 0.124 | 0.281  | 3.42           |
> | Gemini2.5-Flash | BGE       | clauses  | 42.1     | 83.6     | 78.9     | 30.9     | 59.2     | 50.4     | 0.125 | 0.281  | 3.39           |
> | Gemini2.5-Flash | BGE       | original | 34.6     | 77.8     | 71.0     | 27.5     | 55.6     | 47.7     | 0.124 | 0.280  | 3.21           |
>
> *Note: (i) Rec. is for Recall (ii) Sel. is for selection F1 (iii) Avg for the average of 5 metrics of LLM-Judge*
>
> ### **1.3 Key Findings**
>
> 1. **Clear correlation between retrieval quality and downstream performance**: Results demonstrate that higher retrieval recall consistently leads to improved quote selection accuracy and answer generation quality.
>
> 2. **Realistic performance degradation**: Even state-of-the-art models (GPT4 and Gemini2.5) show approximately 10% performance degradation when using imperfect retrievers compared to the perfect retriever baseline.
>
> 3. **Importance of retrieval strategy**: Multi-retriever approaches with query expansion achieve significantly better recall (84.9%) compared to naive single-retriever methods (71.0%), demonstrating the importance of retrieval strategy in real-world scenarios.
>
> We appreciate the valuable feedback. These experiments validate that our benchmark captures the challenges of end-to-end RAG systems, extending our benchmark capable of measuring both retrieval and generation components for practical multimodal RAG applications.
>
>
>
> ## **2. VLM Fine-tuning Results**
>
> We appreciate the valuable feedback regarding the limitation of fine-tuning only LLMs. To address this concern, we conducted additional experiments fine-tuning Qwen2.5-VL models (3B and 7B), InternVL3 models (2B, 8B, and 9B) on our multimodal RAG task using interleaved multimodal inputs.
>
> ### **2.1 Experimental Results**
>
>
> | Model               | Txt Sel. | Img Sel. | All Sel. | Bleu  | RougeL | LLM Judge: Avg |
> | ------------------- | -------- | -------- | -------- | ----- | ------ | -------------- |
> | Qwen2.5-VL-3B       | 5.3      | 1.0      | 1.2      | 0.105 | 0.283  | 2.25           |
> | - After fine-tuning | 9.8      | 68.6     | 55.5     | 0.186 | 0.341  | 3.36           |
> | Qwen2.5-VL-7B       | 16.3     | 23.2     | 16.6     | 0.069 | 0.273  | 2.47           |
> | - After fine-tuning | 16.5     | 72.4     | 58.6     | 0.199 | 0.355  | 3.64           |
> | InternVL3-2B        | 10.7     | 9.5      | 6.8      | 0.097 | 0.250  | 2.10           |
> | - After fine-tuning | 8.9      | 63.6     | 51.4     | 0.180 | 0.333  | 3.15           |
> | InternVL3-8B        | 34.4     | 40.7     | 37.0     | 0.119 | 0.260  | 2.98           |
> | - After fine-tuning | 30.4     | 71.7     | 58.8     | 0.205 | 0.356  | 3.69           |
> | InternVL3-9B        | 29.3     | 60.8     | 50.9     | 0.146 | 0.303  | 3.12           |
> | - After fine-tuning | 28.4     | 72.9     | 60.3     | 0.210 | 0.362  | 3.87           |
>
> ###  **2.2 Key Findings**
>
> 1. **Dramatic improvement in quote selection**: Fine-tuning helps models much better at selecting relevant quotes, especially images. The Qwen2.5-VL-3B model's performance jumps from 1.2 to 55.5, InternVL3-2B model improves from 6.8 to 51.4, and the Qwen2.5-VL-7B model improves from 16.6 to 58.6. InternVL3-8B and 9B model also experience substantial improvements.
>
> 2. **Enhanced answer quality across all metrics**: Fine-tuning consistently improves BLEU scores, RougeL scores, and LLM judge ratings.
>
> 3. **Successful adaptation of VLMs to multimodal RAG**: These results show that VLMs can effectively learn multimodal RAG tasks when properly trained. However, inferring and training VLMs typically incur significant more resources due to prolonged input sequence length (QwenVL: 3.6k vs. 8.7k, InternVL: 3.6K vs. 17.2k)
>
> We thank for the suggestion of finetuning VLM, and the dramatic improvements after fine-tuning validate both our high-quality training dataset and comprehensive benchmark design. Our evaluation effectively captures VLM capabilities across quote selection, answer generation, and multimodal understanding. These findings establish our benchmark as a valuable resource for training and evaluating VLMs on realistic multimodal RAG tasks.
>
>
>
> ## **3. Task Design and Scope of MMDocRAG**
>
> We acknowledge that our benchmark does not cover all document understanding tasks such as open-ended generation, summarization, or conversational multihop QA. However, these tasks are already well-addressed by existing benchmarks including MS MARCO, Natural Questions, QuAC, CoQA, and various summarization datasets like CNN/DailyMail and XSum.
>
> In contrast, multimodal RAG on long documents remains significantly underexplored. Given resource constraints, we strategically prioritized filling this critical research gap by focusing specifically on multimodal RAG challenges. Our benchmark provides comprehensive evaluation across three key dimensions: (i) retrieval at both page and quote levels, (ii) evidence selection from multimodal sources, and (iii) multimodal answer generation.
>
> This focused design allows for thorough assessment of multimodal RAG capabilities while addressing a distinct and underserved area of document understanding research.
>
>
>
> ## **4. Ground Truth Calibration**
>
> We fully agree that calibrating ground truth for multimodal answers and quotes is challenging, particularly for questions spanning multiple pages (39.2% of our dataset). To address this concern, we implemented a rigorous multi-stage validation process (in Section 2.3):
>
> - **Semi-automated validation**: VLMs performed initial accuracy checks for quote selection and answer coherence, followed by expert annotator review. This process retained 90.2% of answers and 93.5% of gold quotes after validation.
>
> - **Manual cross-validation**: We employed a dual-group annotation strategy with 500 overlapping QA pairs serving as validation checkpoints. Each group independently annotated separate portions while both groups labeled the overlapping set for quality comparison. This approach achieved high inter-annotator agreement with F1 scores of 89.7 and 91.4 for quote selection, and answer quality scores of 4.23 and 4.17 respectively.
>
> **Manual refinement**: Annotators systematically corrected logical and factual errors while optimizing multimodal citation placement for better text-image coherence. This comprehensive validation framework and manual efforts ensure reliable ground truth despite the inherent complexity of multimodal long-document QA.
>
>
>
> ## **5. Recent Related Work**
>
> Thank you for the reminder. We will cite ViDoRAG and MDocAgent in our revision to ensure comprehensive coverage of recent related work.
>
>
>
> ## **6. Dataset Code Accessibility**
>
> We believe there may have been a misunderstanding regarding dataset code accessibility (a 'No' has been ticked on it). We would like to clarify that all artifacts (data, code, etc.) in MMDocRAG are publicly available, as acknowledged by Reviewer bkr6:
>
> > "The submission provides both the dataset and code in their final and executable forms, with clear documentation that supports ease of use and reproducibility".
>
>
>
> We hope these responses address your concerns comprehensively. Feel free to let us know if you have any additional questions, and we are more than happy to provide further clarification on any aspect of our work.
>
> Best regards,
>
> The Authors

---

> > ### Comment · Reviewer_GErz · 2025-08-01
> >
> > Thanks for your responses. It addresses most of my concerns. I will raise my score.

---

> > > ### Author Response · Authors · 2025-08-03
> > >
> > > Dear Reviewer GErz,
> > >
> > > We sincerely thank you for your constructive review and for raising your score after considering our responses. Your feedback is invaluable in helping us address the key issues and improve the quality of our work. We appreciate the time and effort you invested in this review process.
> > >
> > > Best regards,
> > > The Authors

---

### Official Review · Reviewer_bkr6 · 2025-07-03

**Rating:** 4
**Confidence:** 4

**Summary:**

This paper introduces MMDocRAG, a comprehensive benchmark designed for multimodal document question answering and retrieval-augmented generation (RAG). The benchmark comprises 4,055 expert-annotated QA pairs with multimodal evidence chains, along with novel evaluation metrics for quote selection and interleaved multimodal answer generation. Extensive experiments involving 60 state-of-the-art large language models (LLMs) and vision-language models (VLMs) reveal persistent challenges in multimodal evidence retrieval, selection, and integration. The results further underscore the superior performance of proprietary models compared to open-source counterparts, as well as the advantages of fine-tuning for improved task performance.

**Additional Feedback:**

A more detailed error analysis of proprietary models, along with the inclusion of fine-tuning results for Qwen2.5-VL models, will further strengthen the contribution of this work and better demonstrate its practical value.

**Dataset Code Accessibility:**

Yes

**Dataset Code Comments:**

The submission provides both the dataset and code in their final and executable forms, with clear documentation that supports ease of use and reproducibility.

**Ethical Comments:**

The paper does not raise any substantial ethical issues. The proposed benchmark focuses on multimodal document understanding and retrieval-augmented generation, and the dataset is constructed using expert-annotated QA pairs without involving sensitive personal data or content.

**Ethical Considerations:**

No, there are no or only very minor ethics concerns

**Limitations Weaknesses:**

1. Limited Scope of Multimodal Retrieval: MMDocRAG primarily evaluates models' capabilities in selecting and integrating multimodal quotes from a pre-retrieved, noisy candidate set, rather than assessing the full retrieval pipeline. However, in real-world RAG applications, systems often face situations where no relevant quotes are retrieved, particularly when dealing with imperfect or underperforming multimodal retrieval modules. The current benchmark setup does not sufficiently account for such unanswerable or retrieval-failure cases, which limits its applicability in evaluating end-to-end RAG systems under realistic conditions.

2. Insufficient Error Analysis of Proprietary Models: While the paper presents some error analysis for open-source models, it lacks a detailed examination of the specific failure modes exhibited by proprietary models. A more comprehensive error analysis for these models could yield valuable insights into their limitations and areas for improvement. Such analysis is an essential component of a benchmark study, as it can provide guidance for both model development and future benchmarking efforts.

3. Limitations of Fine-Tuning: The experiments fine-tune only five Qwen2.5 models by converting multimodal quotes into pure text sequences. Due to the computational constraints, this paper does not report fine-tuning results for the Qwen2.5-VL models. This is a notable limitation. While the computational burden is understandable, including fine-tuning results for at least one or two smaller-sized Qwen2.5-VL models would have offered deeper insights into the capabilities and limitations of vision-language models (VLMs) in multimodal RAG tasks. This can be achieved with limited resources and will strengthen the empirical evaluation.

**Strengths Contributions:**

1. The paper makes substantial contributions to the fields of Document Visual Question Answering (DocVQA) and Retrieval-Augmented Generation (RAG) by introducing MMDocRAG, a high-quality benchmark specifically designed to address the limitations of existing benchmarks in evaluating multimodal evidence selection and integration.

2. MMDocRAG is a comprehensive benchmark that includes a large number of expert-annotated QA pairs, covering diverse forms of evidence and incorporating a realistic mix of gold-standard and noisy quotes to better reflect real-world retrieval scenarios.

3. Novel Evaluation Metrics and Multimodal Answer Support: The paper proposes innovative evaluation metrics tailored to multimodal evidence selection and introduces support for multimodal answer generation. These design choices enhance both the interpretability and cognitive plausibility of model outputs. Additionally, the results of large-scale experiments offer valuable insights into the current capabilities and limitations of multimodal RAG systems, while providing actionable guidance for future research directions.

4. The paper is clearly written and well-structured.

---

> ### Author Rebuttal · Authors · 2025-07-30
>
> Dear Reviewer bkr6,
>
> We sincerely thank you for your thoughtful review and valuable feedback on our work. Below, we address each of your concerns with additional experimental evidence and clarifications.
>
>
>
> ## **1. End-to-end RAG Experimental Results**
>
> We acknowledge the concern about the idealized nature of fixed-quote settings. To address this limitation and simulate more realistic RAG environments, we conducted comprehensive end-to-end experiments that evaluate system performance under imperfect retrieval conditions.
>
> ###  **1.1 Experimental Setup**
>
>
> We tested different retrieval configurations to assess multimodal generation quality when retrieval is imperfect:
>
> - **Perfect retriever** (baseline): All gold quotes included with noisy quotes (8 image + 12 text quotes)
> - **Imperfect retrievers**:
>   - Single BGE retriever with original questions
>   - Single BGE retriever with expanded multi-clause queries
>   - Multiple retrievers (BGE, Qwen3-0.6B, BM25, E5) with expanded queries
>   - Reranking applied using Qwen3-0.6B-reranker (top 8 image + 12 text quotes)
>
> *Note: We focused on single-vector embedding retrievers (BGE, Qwen3-0.6B, E5) for compatibility with vector databases (Milvus DB), excluding multi-vector token-level embeddings (e.g., ColBERT, ColQwen).*
>
> ### **1.2 Experimental Results**
>
> | Model           | Retriever | Query    | Txt Rec. | Img Rec. | All Rec. | Txt Sel. | Img Sel. | All Sel. | Bleu  | RougeL | LLM judge: Avg |
> | --------------- | --------- | -------- | -------- | -------- | -------- | -------- | -------- | -------- | ----- | ------ | -------------- |
> | GPT-4.1         | perfect   | -        | -        | -        | -        | 52.0     | 80.7     | 70.2     | 0.157 | 0.313  | 4.14           |
> | GPT-4.1         | multiple  | clauses  | 49.5     | 86.8     | 84.9     | 41.4     | 65.6     | 59.9     | 0.141 | 0.303  | 3.79           |
> | GPT-4.1         | BGE       | clauses  | 42.1     | 83.6     | 78.9     | 37.9     | 64.0     | 57.5     | 0.141 | 0.302  | 3.71           |
> | GPT-4.1         | BGE       | original | 34.6     | 77.8     | 71.0     | 34.2     | 60.8     | 54.4     | 0.137 | 0.299  | 3.53           |
> | Gemini2.5-Flash | perfect   | -        | -        | -        | -        | 46.1     | 76.2     | 62.4     | 0.139 | 0.284  | 3.76           |
> | Gemini2.5-Flash | multiple  | clauses  | 49.5     | 86.8     | 84.9     | 34.3     | 60.3     | 51.8     | 0.124 | 0.281  | 3.42           |
> | Gemini2.5-Flash | BGE       | clauses  | 42.1     | 83.6     | 78.9     | 30.9     | 59.2     | 50.4     | 0.125 | 0.281  | 3.39           |
> | Gemini2.5-Flash | BGE       | original | 34.6     | 77.8     | 71.0     | 27.5     | 55.6     | 47.7     | 0.124 | 0.280  | 3.21           |
>
> *Note: (i) Rec. is for Recall (ii) Sel. is for selection F1 (iii) Avg for the average of 5 metrics of LLM-Judge*
>
> ###  **1.3 Key Findings**
>
> 1. **Clear correlation between retrieval quality and downstream performance**: Results demonstrate that higher retrieval recall consistently leads to improved quote selection accuracy and answer generation quality.
>
> 2. **Realistic performance degradation**: Even state-of-the-art models (GPT4 and Gemini2.5) show approximately 10% performance degradation when using imperfect retrievers compared to the perfect retriever baseline.
>
> 3. **Importance of retrieval strategy**: Multi-retriever approaches with query expansion achieve significantly better recall (84.9%) compared to naive single-retriever methods (71.0%), demonstrating the importance of retrieval strategy in real-world scenarios.
>
> We appreciate the valuable feedback. These experiments validate that our benchmark captures the challenges of end-to-end RAG systems, extending our benchmark capable of measuring both retrieval and generation components for practical multimodal RAG applications.
>
>
>
> ## **2. Error Analysis**
>
> Thank you for highlighting the need for more comprehensive error analysis of proprietary models. We acknowledge that our initial analysis focused solely on GPT-4o due to resource constraints. To address this concern, we have conducted additional error analysis on the same 200 cases for Claude-3.5-Sonnet and Gemini-2.5-Pro, examining the same error categories detailed in Appendix F.1.
>
> ### **2.1 Expanded Error Analysis Results:**
>
> | Model             | Excessive Citation | Inadequate Citation | Citation Misplacement | Reasoning/Factual Issues | Text-Image Incoherence |
> | ----------------- | ------------------ | ------------------- | --------------------- | ------------------------ | ---------------------- |
> | GPT-4o            | 34.5%              | 30.0%               | 16.5%                 | 17.7%                    | 16.3%                  |
> | Claude-3.5-Sonnet | 36.9%              | 26.2%               | 20.7%                 | 18.5%                    | 15.5%                  |
> | Gemini-2.5-Pro    | 32.3%              | 18.6%               | 17.4%                 | 13.8%                    | 12.5%                  |
>
> ### **2.2 Key Findings:**
>
> - **Citation patterns vary significantly** across models: while all models struggle with excessive citation (32-37%), Gemini-2.5-Pro shows notably better performance in adequate citation (18.6% vs. 26-30% for others)
> - **Reasoning consistency** follows model performance rankings, with Gemini-2.5-Pro demonstrating the lowest error rate (13.8%) in factual inconsistencies
> - **Cross-model consistency** in citation positioning errors (16-21%) suggests this remains a fundamental challenge in multimodal document understanding
>
> We hope this expanded analysis offers valuable additional insight into the comparative limitations and failure modes of current proprietary models. We appreciate your suggestion, which has helped improve the comprehensiveness of our benchmark study.
>
>
>
> ## **3. VLM Fine-tuning Results**
>
> We appreciate the valuable feedback regarding the limitation of fine-tuning only LLMs. To address this concern, we conducted additional experiments fine-tuning Qwen2.5-VL models (3B and 7B), InternVL3 models (2B, 8B, and 9B) on our multimodal RAG task using interleaved multimodal inputs.
>
> ### **3.1 Experimental Results**
>
>
> | Model               | Txt Sel. | Img Sel. | All Sel. | Bleu  | RougeL | LLM Judge: Avg |
> | ------------------- | -------- | -------- | -------- | ----- | ------ | -------------- |
> | Qwen2.5-VL-3B       | 5.3      | 1.0      | 1.2      | 0.105 | 0.283  | 2.25           |
> | - After fine-tuning | 9.8      | 68.6     | 55.5     | 0.186 | 0.341  | 3.36           |
> | Qwen2.5-VL-7B       | 16.3     | 23.2     | 16.6     | 0.069 | 0.273  | 2.47           |
> | - After fine-tuning | 16.5     | 72.4     | 58.6     | 0.199 | 0.355  | 3.64           |
> | InternVL3-2B        | 10.7     | 9.5      | 6.8      | 0.097 | 0.250  | 2.10           |
> | - After fine-tuning | 8.9      | 63.6     | 51.4     | 0.180 | 0.333  | 3.15           |
> | InternVL3-8B        | 34.4     | 40.7     | 37.0     | 0.119 | 0.260  | 2.98           |
> | - After fine-tuning | 30.4     | 71.7     | 58.8     | 0.205 | 0.356  | 3.69           |
> | InternVL3-9B        | 29.3     | 60.8     | 50.9     | 0.146 | 0.303  | 3.12           |
> | - After fine-tuning | 28.4     | 72.9     | 60.3     | 0.210 | 0.362  | 3.87           |
>
> ###  **3.2 Key Findings**
>
> 1. **Dramatic improvement in quote selection**: Fine-tuning helps models much better at selecting relevant quotes, especially images. The Qwen2.5-VL-3B model's performance jumps from 1.2 to 55.5, InternVL3-2B model improves from 6.8 to 51.4, and the Qwen2.5-VL-7B model improves from 16.6 to 58.6. InternVL3-8B and 9B model also experience substantial improvements.
>
> 2. **Enhanced answer quality across all metrics**: Fine-tuning consistently improves BLEU scores, RougeL scores, and LLM judge ratings.
>
> 3. **Successful adaptation of VLMs to multimodal RAG**: These results show that VLMs can effectively learn multimodal RAG tasks when properly trained. However, inferring and training VLMs typically incur significant more resources due to prolonged input sequence length (QwenVL: 3.6k vs. 8.7k, InternVL: 3.6K vs. 17.2k)
>
> We thank for the suggestion of finetuning VLM, and the dramatic improvements after fine-tuning validate both our high-quality training dataset and comprehensive benchmark design. Our evaluation effectively captures VLM capabilities across quote selection, answer generation, and multimodal understanding. These findings establish our benchmark as a valuable resource for training and evaluating VLMs on realistic multimodal RAG tasks.
>
>
>
> We hope these responses address your concerns comprehensively. Feel free to let us know if you have any additional questions, and we are more than happy to provide further clarification on any aspect of our work.
>
> Best regards,
>
> The Authors

---

> > ### Author Response · Authors · 2025-08-05
> >
> > Dear Reviewer bkr6,
> >
> > Thank you for your initial feedback on our manuscript. We have submitted a comprehensive response with extensive additional experiments to address the concerns you raised.
> >
> > As we approach the end of the discussion period, we would like to check if our responses and new experimental results have adequately addressed your questions. If our clarifications have resolved your concerns, we would be grateful if you could by any chance, consider adjusting your evaluation accordingly.
> >
> > Should any points require further clarification or additional experiments, we remain ready to provide prompt responses.
> >
> > Thank you for your time and consideration.
> > Best regards,
> > Authors

---

### Note · Authors · 2025-08-15

We sincerely thank all reviewers for their constructive feedback and recognition of MMDocRAG's contributions.

**Core Contributions Acknowledged by Reviewers:**

- **Comprehensive benchmark filling critical gap** (bkr6, GErz, Cb3Y, ddPK): First multimodal Doc-VQA/RAG benchmark with 4,055 expert-annotated QA pairs, multi-page evidence chains, and realistic gold/noisy quote mixture for systematic evaluation on realistic, noisy documents.
- **Novel evaluation framework** (bkr6, GErz, Cb3Y): Innovative metrics for multimodal quote selection and interleaved multimodal answer generation with multi-level annotations (page/quote/evidence-type), enhancing interpretability and cognitive plausibility.
- **Extensive empirical analysis** (bkr6, GErz, Cb3Y, ddPK): Large-scale experiments across 60+ models (37 open-source, 23 proprietary, 5 fine-tuned) with 14 retrievers, providing actionable research guidance.
- **Clear presentation** (bkr6, ddPK): Well-written and structured paper facilitating understanding and application of MMDocRAG.

**Addressing Reviewer Concerns with Extended Results:**

We conducted comprehensive additional experiments to strengthen our work:

- **1. End-to-end RAG experiments** (bkr6, GErz): Implemented 4 retrieval strategies with single/multiple retrievers, query expansion and reranking, demonstrating strong correlation between retrieval quality and downstream performance, validating practical applicability.
- **2. VLM fine-tuning** (bkr6, GErz): Fine-tuned 5 VLMs (Qwen2.5-VL 3B/7B, InternVL3 2B/8B/9B) showing dramatic improvements across all metrics, establishing MMDocRAG as valuable training resource.
- **3. Fine-grained analysis** (Cb3Y, ddPK): Analyzed performance across single/multi-page, single/multi-image, cross-modal evidence, and 8 question types, revealing key challenges in cross-modal integration and multi-page reasoning for future research.
- **4. Extended error analysis/evaluation** (bkr6, ddPK): Human evaluation on GPT-4o, Claude-3.5 and Gemini-2.5 outputs confirms strong manual-LLM alignment; Selection F1 correlates well with LLM-judge citation scores, validating our automatic metrics.

These comprehensive additions along with other clarifications directly address all reviewer concerns and significantly strengthen our contribution. All results will be incorporated in our revised version. We are grateful for the reviewers' valuable insights that help improve our work substantially.

Best regards,

Authors

---

### Decision · Program_Chairs · 2025-09-18

**Decision:**

Accept (poster)

**Comment:**

This paper presents MMDocRAG, a benchmark addressing the limitations of prior DocVQA methods that often miss critical visual information. It includes 4,055 expert-annotated QA pairs requiring reasoning across text, images, and tables spanning multiple pages. In addition, the MMDocRAG benchmark reveals major challenges in multimodal evidence retrieval, selection, and integration.

Strengths:
1. This paper demonstrates a clear and well-structured organization, which is easy to read.
2. MMDocRAG is the first comprehensive benchmark for multimodal DocVQA, which features 4,055 expert-annotated QA pairs. It has the potential to accelerate research of this community.
3. This paper introduces new evaluation metrics specifically tailored for assessing multimodal evidence selection and the quality of answers that interleave text and visual elements.
4. This paper conducts extensive experiments involving a wide range of models (60+ LLMs/VLMs).

Weaknesses:
1. The fine-tuning experiments are limited, as they are conducted only on text-based LLMs. While the authors also report results for fine-tuned VLMs in author responses, insufficient details are provided regarding the fine-tuning procedure—for example, whether full-parameter or partial-parameter fine-tuning was applied. More fine-tuning experiments should be included and discussed.
2. The experimental analysis is broad in scope but somewhat superficial. It primarily describes the experimental results without delving into the underlying mechanisms explaining them.
3. Additional details regarding the human evaluation process should be provided. For instance, the authors should specify the annotators’ background, the number of annotators involved, and the inter-annotator agreement rate.